# Diagnosing and Fixing Manifold Overfitting in Deep Generative Models

**Gabriel Loaiza-Ganem**                                             *gabriel@layer6.ai*
*Layer 6 AI*

**Brendan Leigh Ross**                                               *brendan@layer6.ai*
*Layer 6 AI*

**Jesse C. Cresswell**                                               *jesse@layer6.ai*
*Layer 6 AI*

**Anthony L. Caterini**                                              *anthony@layer6.ai*
*Layer 6 AI*

**Reviewed on OpenReview:** *https://openreview.net/forum?id=0nEZCVshxS*

## Abstract

Likelihood-based, or *explicit*, deep generative models use neural networks to construct flexible high-dimensional densities. This formulation directly contradicts the manifold hypothesis, which states that observed data lies on a low-dimensional manifold embedded in high-dimensional ambient space. In this paper we investigate the pathologies of maximum-likelihood training in the presence of this dimensionality mismatch. We formally prove that degenerate optima are achieved wherein the manifold itself is learned but not the distribution on it, a phenomenon we call *manifold overfitting*. We propose a class of two-step procedures consisting of a dimensionality reduction step followed by maximum-likelihood density estimation, and prove that they recover the data-generating distribution in the nonparametric regime, thus avoiding manifold overfitting. We also show that these procedures enable density estimation on the manifolds learned by *implicit* models, such as generative adversarial networks, hence addressing a major shortcoming of these models. Several recently proposed methods are instances of our two-step procedures; we thus unify, extend, and theoretically justify a large class of models.

## 1  Introduction

We consider the standard setting for generative modelling, where samples $\{x_n\}_{n=1}^N \subset \mathbb{R}^D$ of high-dimensional data from some unknown distribution $\mathbb{P}^*$ are observed, and the task is to estimate $\mathbb{P}^*$. Many deep generative models (DGMs) (Bond-Taylor et al., 2021), including variational autoencoders (VAEs) (Kingma & Welling, 2014; Rezende et al., 2014; Ho et al., 2020; Kingma et al., 2021) and variants such as adversarial variational Bayes (AVB) (Mescheder et al., 2017), normalizing flows (NFs) (Dinh et al., 2017; Kingma & Dhariwal, 2018; Behrmann et al., 2019; Chen et al., 2019; Durkan et al., 2019; Cornish et al., 2020), energy-based models (EBMs) (Du & Mordatch, 2019), and continuous autoregressive models (ARMs) (Uria et al., 2013; Theis & Bethge, 2015), use neural networks to construct a flexible density trained to match $\mathbb{P}^*$ by maximizing either the likelihood or a lower bound of it. This modelling choice implies the model has $D$-dimensional support,[1] thus directly contradicting the manifold hypothesis (Bengio et al., 2013), which states that high-dimensional data is supported on $\mathcal{M}$, an unknown $d$-dimensional embedded submanifold of $\mathbb{R}^D$, where $d < D$.

---

[1]This is indeed true of VAEs and AVB, even though both use low-dimensional latent variables, as the observational model being fully dimensional implies every point in $\mathbb{R}^D$ is assigned strictly positive density, regardless of what the latent dimension is.

There is strong evidence supporting the manifold hypothesis. Theoretically, the sample complexity of kernel density estimation is known to scale exponentially with ambient dimension $D$ when no low-dimensional structure exists (Cacoullos, 1966), and with intrinsic dimension $d$ when it does (Ozakin & Gray, 2009). These results suggest the complexity of learning distributions scales exponentially with the intrinsic dimension of their support, and the same applies for manifold learning (Narayanan & Mitter, 2010). Yet, if estimating distributions or manifolds required exponentially many samples in $D$, these problems would be impossible to solve in practice. The success itself of deep-learning-based methods on these tasks thus supports the manifold hypothesis. Empirically, Pope et al. (2021) estimate $d$ for commonly-used image datasets and find that, indeed, it is much smaller than $D$.

A natural question arises: *how relevant is the aforementioned modelling mismatch?* We answer this question by proving that when $\mathbb{P}^*$ is supported on $\mathcal{M}$, maximum-likelihood training of a flexible $D$-dimensional density results in $\mathcal{M}$ itself being learned, but not $\mathbb{P}^*$. Our result extends that of Dai & Wipf (2019) beyond VAEs to all likelihood-based models and drops the empirically unrealistic assumption that $\mathcal{M}$ is homeomorphic to $\mathbb{R}^d$ (e.g. one can imagine the MNIST (LeCun, 1998) manifold as having 10 connected components, one per digit). This phenomenon – which we call *manifold overfitting* – has profound consequences for generative modelling. Maximum-likelihood is indisputably one of the most important concepts in statistics, and enjoys well-studied theoretical properties such as consistency and asymptotic efficiency under seemingly mild regularity conditions (Lehmann & Casella, 2006). These conditions can indeed be reasonably expected to hold in the setting of "classical statistics" under which they were first considered, where models were simpler and available data was of much lower ambient dimension than by modern standards. However, in the presence of $d$-dimensional manifold structure, the previously innocuous assumption that there exists a ground truth $D$-dimensional density cannot possibly hold. Manifold overfitting thus shows that DGMs do not enjoy the supposed theoretical benefits of maximum-likelihood, which is often regarded as a principled objective for training DGMs, because they will recover the manifold but not the distribution on it. We highlight that manifold overfitting is a problem with maximum-likelihood itself, and thus universally affects all explicit DGMs.

In order to address manifold overfitting, we propose a class of two-step procedures, depicted in Fig. 1. The first step, which we call *generalized autoencoding*, [2] reduces the dimension of the data through an encoder $g : \mathbb{R}^D \to \mathbb{R}^d$ while also learning how to map back to $\mathcal{M}$ through a decoder $G : \mathbb{R}^d \to \mathbb{R}^D$. In the second step, maximum-likelihood estimation is performed on the low-dimensional representations $\{g(x_n)\}_{n=1}^N$ using a DGM. Intuitively, the first step removes the dimensionality mismatch in order to avoid manifold overfitting in the second step. This intuition is confirmed in a second theoretical result where we prove that, given enough capacity, our two-step procedures indeed recover $\mathbb{P}^*$ in the infinite data limit while retaining density evaluation. Just as manifold overfitting pervasively affects likelihood-based DGMs, our proposed two-step procedures address this issue in an equally broad manner. We also iden-

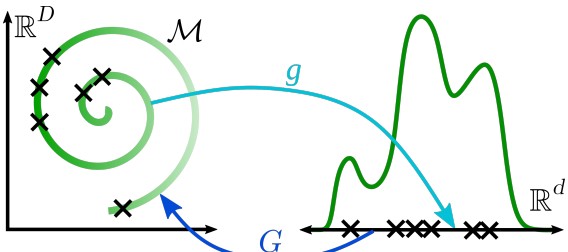

Figure 1: Depiction of our two-step procedures. In the first step, we learn to map from $\mathcal{M}$ to $\mathbb{R}^d$ through $g$, and to invert this mapping through $G$. In the second step, we perform density estimation (green density on the right) on the dataset encoded through $g$. Our learned distribution on $\mathcal{M}$ (shades of green on the spiral) is given by pushing forward the density from the second step through $G$.

tify DGMs that are instances of our procedure class. Our methodology thus results in novel models, and provides a unifying perspective and theoretical justification for all these related works.

We also show that some implicit models (Mohamed & Lakshminarayanan, 2016), e.g. generative adversarial networks (GANs) (Goodfellow et al., 2014), can be made into generalized autoencoders. Consequently, in addition to preventing manifold overfitting on explicit models, our two-step procedures enable density evaluation for implicit models, thus addressing one of their main limitations. We show that this newly obtained ability of implicit models to perform density estimation can be used empirically to perform out-of-distribution

---

[2]Our generalized autoencoders are unrelated to those of Wang et al. (2014).

(OOD) detection, and we obtain very promising results. To the best of our knowledge principled density estimation with implicit models was previously considered impossible.

Finally, we achieve significant empirical improvements in sample quality over maximum-likelihood, strongly supporting our theoretical findings. We show these improvements persist even when accounting for the additional parameters of the second-step model, or when adding Gaussian noise to the data as an attempt to remove the dimensionality mismatch that causes manifold overfitting.

## 2 Related Work and Motivation

**Manifold mismatch**  It has been observed in the literature that $\mathbb{R}^D$-supported models exhibit undesirable behaviour when the support of the target distribution has complicated topological structure. For example, Cornish et al. (2020) show that the bi-Lipschitz constant of topologically-misspecified NFs must go to infinity, even without dimensionality mismatch, explaining phenomena like the numerical instabilities observed by Behrmann et al. (2021). Mattei & Frellsen (2018) observe VAEs can have unbounded likelihoods and are thus susceptible to similar instabilities. Dai & Wipf (2019) study dimensionality mismatch in VAEs and its effects on posterior collapse. These works motivate the development of models with low-dimensional support. Goodfellow et al. (2014) and Nowozin et al. (2016) model the data as the pushforward of a low-dimensional Gaussian through a neural network, thus making it possible to properly account for the dimension of the support. However, in addition to requiring adversarial training – which is more unstable than maximum-likelihood (Chu et al., 2020) – these models minimize the Jensen-Shannon divergence or $f$-divergences, respectively, in the *nonparametric* setting (i.e. infinite data limit with sufficient capacity), which are ill-defined due to dimensionality mismatch. Attempting to minimize Wasserstein distance has also been proposed (Arjovsky et al., 2017; Tolstikhin et al., 2018) as a way to remedy this issue, although estimating this distance is hard in practice (Arora et al., 2017) and unbiased gradient estimators are not available. In addition to having a more challenging training objective than maximum-likelihood, these *implicit* models lose a key advantage of *explicit* models: density evaluation. Our work aims to both properly account for the manifold hypothesis in likelihood-based DGMs while retaining density evaluation, and endow implicit models with density evaluation.

**NFs on manifolds**  Several recent flow-based methods properly account for the manifold structure of the data. Gemici et al. (2016), Rezende et al. (2020), and Mathieu & Nickel (2020) construct flow models for prespecified manifolds, with the obvious disadvantage that the manifold is unknown for most data of interest. Brehmer & Cranmer (2020) propose injective NFs, which model the data-generating distribution as the pushforward of a $d$-dimensional Gaussian through an injective function $G : \mathbb{R}^d \rightarrow \mathbb{R}^D$, and avoid the change-of-variable computation through a two-step training procedure; we will see in Sec. 5 that this procedure is an instance of our methodology. Caterini et al. (2021) and Ross & Cresswell (2021) endow injective flows with tractable change-of-variable computations, the former through automatic differentiation and numerical linear algebra methods, and the latter with a specific construction of injective NFs admitting closed-form evaluation. We build a general framework encompassing a broader class of DGMs than NFs alone, giving them low-dimensional support without requiring injective transformations over $\mathbb{R}^d$.

**Adding noise**  Denoising approaches add Gaussian noise to the data, making the $D$-dimensional model appropriate at the cost of recovering a noisy version of $\mathbb{P}^*$ (Vincent et al., 2008; Vincent, 2011; Alain & Bengio, 2014; Meng et al., 2021; Chae et al., 2021; Horvat & Pfister, 2021a;b; Cunningham & Fiterau, 2021). In particular, Horvat & Pfister (2021b) show that recovering the true manifold structure in this case is only guaranteed when adding noise orthogonally to the tangent space of the manifold, which cannot be achieved in practice when the manifold itself is unknown. In the context of score-matching (Hyvärinen, 2005), denoising has led to empirical success (Song & Ermon, 2019; Song et al., 2021). In Sec. 3.2 we show that adding small amounts of Gaussian noise to a distribution supported on a manifold results in highly peaked densities, which can be hard to learn. Zhang et al. (2020b) also make this observation, and propose to add the same amount of noise to the model itself. However, their method requires access to the density of the model after having added noise, which in practice requires a variational approximation and is thus only applicable to VAEs. Our first theoretical result can be seen as a motivation for any method based on adding noise to the data (as

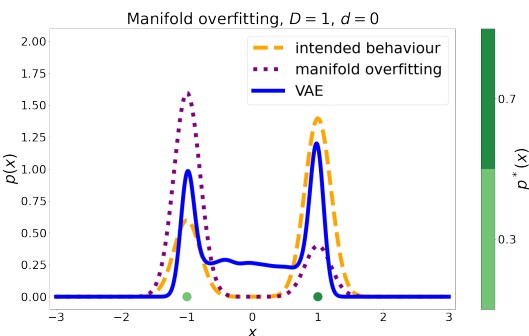
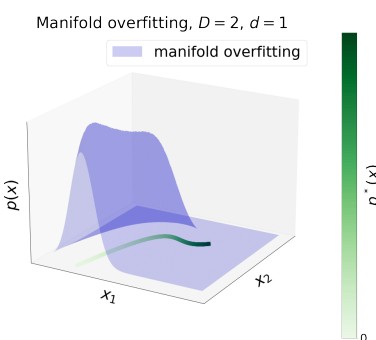

Figure 2: **Left panel**: $\mathbb{P}^*$ (green); $p_t(x) = 0.3 \cdot \mathcal{N}(x; -1, 1/t) + 0.7 \cdot \mathcal{N}(x; 1, 1/t)$ (orange, dashed) for $t = 5$, which converges weakly to $\mathbb{P}^*$ as $t \to \infty$; and $p'_t(x) = 0.8 \cdot \mathcal{N}(x; -1, 1/t) + 0.2 \cdot \mathcal{N}(x; 1, 1/t)$ (purple, dotted) for $t = 5$, which converges weakly to $\mathbb{P}^\dagger = 0.8\delta_{-1} + 0.2\delta_1$ while getting arbitrarily large likelihoods under $\mathbb{P}^*$, i.e. $p'_t(x) \to \infty$ as $t \to \infty$ for $x \in \mathcal{M}$; Gaussian VAE density (blue, solid). **Right panel**: Analogous phenomenon with $D = 2$ and $d = 1$, with the blue density "spiking" around $\mathcal{M}$ in a manner unlike $\mathbb{P}^*$ (green) while achieving large likelihoods.

attempting to address manifold overfitting), and our two-step procedures are applicable to all likelihood-based DGMs. We empirically verify that simply adding Gaussian noise to the data and fitting a maximum-likelihood DGM as usual is not enough to avoid manifold overfitting in practice. Our results highlight that manifold overfitting can manifest itself empirically even when the data is close to a manifold rather than exactly on one, and that naïvely adding noise does not fix it. We hope that our work will encourage further advances aiming to address manifold overfitting, including ones based on adding noise.

## 3 Manifold Overfitting

### 3.1 An Illustrative Example

Consider the simple case where $D = 1$, $d = 0$, $\mathcal{M} = \{-1, 1\}$, and $\mathbb{P}^* = 0.3\delta_{-1} + 0.7\delta_1$, where $\delta_x$ denotes a point mass at $x$. Suppose the data is modelled with a mixture of Gaussians $p(x) = \lambda \cdot \mathcal{N}(x; m_1, \sigma^2) + (1 - \lambda) \cdot \mathcal{N}(x; m_2, \sigma^2)$ parameterized by a mixture weight $\lambda \in [0, 1]$, means $m_1, m_2 \in \mathbb{R}$, and a shared variance $\sigma^2 \in \mathbb{R}_{>0}$, which we will think of as a flexible density. This model can learn the correct distribution in the limit $\sigma^2 \to 0$, as shown on the left panel of Fig. 2 (dashed line in orange). However, arbitrarily large likelihood values can be achieved by other densities – the one shown with a purple dotted line approximates a distribution $\mathbb{P}^\dagger$ on $\mathcal{M}$ which *is not* $\mathbb{P}^*$ but nonetheless has large likelihoods. The implication is simple: maximum-likelihood estimation will not necessarily recover the data-generating distribution $\mathbb{P}^*$. Our choice of $\mathbb{P}^\dagger$ (see figure caption) was completely arbitrary, hence any distribution on $\mathcal{M}$ other than $\delta_{-1}$ or $\delta_1$ could be recovered with likelihoods diverging to infinity. Recovering $\mathbb{P}^*$ is then a coincidence which we should not expect to occur when training via maximum-likelihood. In other words, we should expect maximum-likelihood to recover the manifold (i.e. $m_1 = \pm 1$, $m_2 = \mp 1$ and $\sigma^2 \to 0$), but not the distribution on it (i.e. $\lambda \notin \{0.3, 0.7\}$). We also plot the density learned by a Gaussian VAE (see App. C.2) in blue to show this issue empirically. While this model assigns some probability outside of $\{-1, 1\}$ due to limited capacity, the probabilities assigned around $-1$ and $1$ are far off from $0.3$ and $0.7$, respectively; even after quantizing with the sign function, the VAE only assigns probability $0.53$ to $x = 1$.

The underlying issue here is that $\mathcal{M}$ is "too thin in $\mathbb{R}^D$" (it has Lebesgue measure 0), and thus $p(x)$ can "spike to infinity" at *every* $x \in \mathcal{M}$. If the dimensionalities were correctly matched this could not happen, as the requirement that $p$ integrate to 1 would be violated. We highlight that this issue is not only a problem with data having intrinsic dimension $d = 0$, and can happen whenever $d < D$. The right panel of Fig. 2 shows another example of this phenomenon with $d = 1$ and $D = 2$, where a distribution $\mathbb{P}^*$ (green curve) is poorly approximated with a density $p$ (blue surface) which nonetheless would achieve high likelihoods by "spiking around $\mathcal{M}$". Looking ahead to our experiments, the middle panel of Fig. 4 shows a 2-dimensional

EBM suffering from this issue, spiking around the ground truth manifold on the left panel, but not correctly recovering the distribution on it. The intuition provided by these examples is that if a flexible $D$-dimensional density $p$ is trained with maximum-likelihood when $\mathbb{P}^*$ is supported on a low-dimensional manifold, it is possible to simultaneously achieve large likelihoods while being close to *any* $\mathbb{P}^\dagger$, rather than close to $\mathbb{P}^*$. We refer to this phenomenon as *manifold overfitting*, as the density will concentrate around the manifold, but will do so in an incorrect way, recovering an arbitrary distribution on the manifold rather than the correct one. Note that the problem is not that the likelihood can be arbitrarily large (e.g. intended behaviour in Fig. 2), but that large likelihoods can be achieved *while not recovering* $\mathbb{P}^*$. Manifold overfitting thus calls into question the validity of maximum-likelihood as a training objective in the setting where the data lies on a low-dimensional manifold.

### 3.2 The Manifold Overfitting Theorem

We now formalize the intuition developed so far. We assume some familiarity with measure theory (Billingsley, 2008) and with smooth (Lee, 2013) and Riemannian manifolds (Lee, 2018). Nonetheless, we provide a measure theory primer in App. A, where we informally review relevant concepts such as absolute continuity of measures ($\ll$), densities as Radon-Nikodym derivatives, weak convergence, properties holding almost surely with respect to a probability measure, and pushforward measures. We also use the concept of Riemannian measure (Pennec, 2006), which plays an analogous role on manifolds to that of the Lebesgue measure on Euclidean spaces. We briefly review Riemannian measures in App. B.1, and refer the reader to Dieudonné (1973) for a thorough treatment.[3] We begin by defining a useful condition on probability distributions for the following theorems, which captures the intuition of "continuously spreading mass all around $\mathcal{M}$".

**Definition 1 (Smoothness of Probability Measures):** Let $\mathcal{M}$ be a finite-dimensional $C^1$ manifold, and let $\mathbb{P}$ be a probability measure on $\mathcal{M}$. Let $\mathfrak{g}$ be a Riemannian metric on $\mathcal{M}$ and $\mu_{\mathcal{M}}^{(\mathfrak{g})}$ the corresponding Riemannian measure. We say that $\mathbb{P}$ is *smooth* if $\mathbb{P} \ll \mu_{\mathcal{M}}^{(\mathfrak{g})}$ and it admits a continuous density $p : \mathcal{M} \to \mathbb{R}_{>0}$ with respect to $\mu_{\mathcal{M}}^{(\mathfrak{g})}$.

Note that smoothness of $\mathbb{P}$ is independent of the choice of Riemannian metric $\mathfrak{g}$ (see App. B.1). We emphasize that this is a weak requirement, corresponding in the Euclidean case to $\mathbb{P}$ admitting a continuous and positive density with respect to the Lebesgue measure, and that it is not required of $\mathbb{P}^*$ in our first theorem below. Denoting the Lebesgue measure on $\mathbb{R}^D$ as $\mu_D$, we now state our first result.

**Theorem 1 (Manifold Overfitting):** Let $\mathcal{M} \subset \mathbb{R}^D$ be an analytic $d$-dimensional embedded submanifold of $\mathbb{R}^D$ with $d < D$, and $\mathbb{P}^\dagger$ a smooth probability measure on $\mathcal{M}$. Then there exists a sequence of probability measures $(\mathbb{P}_t)_{t=1}^\infty$ on $\mathbb{R}^D$ such that:

1. $\mathbb{P}_t \to \mathbb{P}^\dagger$ weakly as $t \to \infty$.

2. For every $t \geq 1$, $\mathbb{P}_t \ll \mu_D$ and $\mathbb{P}_t$ admits a density $p_t : \mathbb{R}^D \to \mathbb{R}_{>0}$ with respect to $\mu_D$ such that:

   (a) $\lim_{t \to \infty} p_t(x) = \infty$ for every $x \in \mathcal{M}$.

   (b) $\lim_{t \to \infty} p_t(x) = 0$ for every $x \notin \text{cl}(\mathcal{M})$, where $\text{cl}(\cdot)$ denotes closure in $\mathbb{R}^D$.

**Proof sketch:** We construct $\mathbb{P}_t$ by convolving $\mathbb{P}^\dagger$ with 0 mean, $\sigma_t^2 I_D$ covariance Gaussian noise for a sequence $(\sigma_t^2)_{t=1}^\infty$ satisfying $\sigma_t^2 \to 0$ as $t \to \infty$, and then carefully verify that the stated properties of $\mathbb{P}_t$ indeed hold. See App. B.2 for the full formal proof.

Informally, part 1 says that $\mathbb{P}_t$ can get arbitrarily close to $\mathbb{P}^\dagger$, and part 2 says that this can be achieved with densities diverging to infinity on all $\mathcal{M}$. The relevance of this statement is that large likelihoods of a model do not imply it is adequately learning the target distribution $\mathbb{P}^*$, showing that maximum-likelihood is not a valid objective when data has low-dimensional manifold structure. Maximizing $\frac{1}{N} \sum_{n=1}^N \log p(x_n)$, or $\mathbb{E}_{X \sim \mathbb{P}^*}[\log p(X)]$ in the nonparametric regime, over a $D$-dimensional density $p$ need not recover $\mathbb{P}^*$: since $\mathbb{P}^*$ is supported on $\mathcal{M}$, it follows by Theorem 1 that not only can the objective be made arbitrarily large, but that this can be done while recovering *any* $\mathbb{P}^\dagger$, which need not match $\mathbb{P}^*$. The failure to recover $\mathbb{P}^*$ is caused

---

[3]See especially Sec. 22 of Ch. 16. Note Riemannian measures are called Lebesgue measures in this reference.

by the density being able to take arbitrarily large values on all of $\mathcal{M}$, thus *overfitting to the manifold*. When $p$ is a flexible density, as for many DGMs with universal approximation properties (Hornik, 1991; Koehler et al., 2021), manifold overfitting becomes a key deficiency of maximum-likelihood – which we fix in Sec. 4.

Note also that the proof of Theorem 1 applied to the specific case where $\mathbb{P}^{\dagger} = \mathbb{P}^*$ formalizes the intuition that adding small amounts of Gaussian noise to $\mathbb{P}^*$ results in highly peaked densities, suggesting that the resulting distribution, which denoising methods aim to estimate, might be empirically difficult to learn. More generally, even if there exists a ground truth $D$-dimensional density which allocates most of its mass around $\mathcal{M}$, this density will be highly peaked. In other words, even if Theorem 1 does not technically apply in this setting, it still provides useful intuition as manifold overfitting might still happen in practice. Indeed, we empirically confirm in Sec. 6 that even if $\mathbb{P}^*$ is only "very close" to $\mathcal{M}$, manifold overfitting remains a problem.

**Differences from regular overfitting** Manifold overfitting is fundamentally different from regular overfitting. At its core, regular overfitting involves memorizing observed datapoints as a direct consequence of maximizing the finite-sample objective $\frac{1}{N}\sum_{n=1}^{N} \log p(x_n)$. This memorization can happen in different ways, e.g. the empirical distribution $\hat{\mathbb{P}}_N = \frac{1}{N}\sum_{n=1}^{N} \delta_{x_n}$ could be recovered.[4] Recovering $\hat{\mathbb{P}}_N$ requires increased model capacity as $N$ increases, as new data points have to be memorized. In contrast, manifold overfitting only requires enough capacity to concentrate mass around the manifold. Regular overfitting can happen in other ways too: a classical example (Bishop, 2006) being $p(x) = \frac{1}{2}\mathcal{N}(x; 0, I_D) + \frac{1}{2}\mathcal{N}(x; x_1, \sigma^2 I_D)$, which achieves arbitrarily large likelihoods as $\sigma^2 \to 0$ and only requires memorizing $x_1$. On the other hand, manifold overfitting does not arise from memorizing datapoints, and unlike regular overfitting, can persist even when maximizing the nonparametric objective $\mathbb{E}_{X \sim \mathbb{P}^*}[\log p(X)]$. Manifold overfitting is thus a more severe problem than regular overfitting, as it does not disappear in the infinite data regime. This property of manifold overfitting also makes detecting it more difficult: an unseen test datapoint $x_{N+1} \in \mathcal{M}$ will still be assigned very high likelihood – in line with the training data – under manifold overfitting, yet very low likelihood under regular overfitting. Comparing train and test likelihoods is thus not a valid way of detecting manifold overfitting, once again contrasting with regular overfitting, and highlighting that manifold overfitting is the more acute problem of the two.

**A note on divergences** Maximum-likelihood is often thought of as minimizing the KL divergence $\mathbb{KL}(\mathbb{P}^*||\mathbb{P})$ over the model distribution $\mathbb{P}$. Naïvely one might believe that this contradicts the manifold overfitting theorem, but this is not the case. In order for $\mathbb{KL}(\mathbb{P}^*||\mathbb{P}) < \infty$, it is required that $\mathbb{P}^* \ll \mathbb{P}$, which does not happen when $\mathbb{P}^*$ is a distribution on $\mathcal{M}$ and $\mathbb{P} \ll \mu_D$. For example, $\mathbb{KL}(\mathbb{P}^*||\mathbb{P}_t) = \infty$, for every $t \geq 1$ even if $\mathbb{E}_{X \sim \mathbb{P}^*}[\log p_t(X)]$ varies in $t$. In other words, minimizing the KL divergence is not equivalent to maximizing the likelihood in the setting of dimensionality mismatch, and the manifold overfitting theorem elucidates the effect of maximum-likelihood training in this setting. Similarly, other commonly considered divergences – such as $f$-divergences – cannot be meaningfully minimized. Arjovsky et al. (2017) propose using the Wasserstein distance as it is well-defined even in the presence of support mismatch, although we highlight once again that estimating and/or minimizing this distance is difficult in practice.

**Non-convergence of maximum-likelihood** The manifold overfitting theorem shows that any smooth distribution $\mathbb{P}^{\dagger}$ on $\mathcal{M}$ can be recovered through maximum-likelihood, even if it does not match $\mathbb{P}^*$. It does not, however, guarantee that *some* $\mathbb{P}^{\dagger}$ will even be recovered. It is thus natural to ask whether it is possible to have a sequence of distributions achieving arbitrarily large likelihoods while not converging at all. The result below shows this to be true: in other words, training a $D$-dimensional model could result in maximum-likelihood not even converging.

**Corollary 1:** Let $\mathcal{M} \subset \mathbb{R}^D$ be an analytic $d$-dimensional embedded submanifold of $\mathbb{R}^D$ with more than a single element, and $d < D$. Then, there exists a sequence of probability measures $(\mathbb{P}_t)_{t=1}^{\infty}$ on $\mathbb{R}^D$ such that:

1. $(\mathbb{P}_t)_{t=1}^{\infty}$ does not converge weakly.

2. For every $t \geq 1$, $\mathbb{P}_t \ll \mu_D$ and $\mathbb{P}_t$ admits a density $p_t : \mathbb{R}^D \to \mathbb{R}_{>0}$ with respect to $\mu_D$ such that:
   (a) $\lim_{t \to \infty} p_t(x) = \infty$ for every $x \in \mathcal{M}$.

---

[4]For example, the flexible model $p(x) = \frac{1}{N}\sum_{n=1}^{N} \mathcal{N}(x; x_n, \sigma^2 I_D)$ with $\sigma^2 \to 0$ recovers $\hat{\mathbb{P}}_N$.

(b) $\lim_{t\to\infty} p_t(x) = 0$ for every $x \notin \mathrm{cl}(\mathcal{M})$.

**Proof:** Let $\mathbb{P}^{\dagger 1}$ and $\mathbb{P}^{\dagger 2}$ be two different smooth probability measures on $\mathcal{M}$, which exist since $\mathcal{M}$ has more than a single element. Let $(\mathbb{P}_t^1)_{t=1}^\infty$ and $(\mathbb{P}_t^2)_{t=1}^\infty$ be the corresponding sequences from Theorem 1. The sequence $(\mathbb{P}_t)_{t=1}^\infty$, given by $\mathbb{P}_t = \mathbb{P}_t^1$ if $t$ is even and $\mathbb{P}_t = \mathbb{P}_t^2$ otherwise, satisfies the above requirements. $\qquad\square$

## 4 Fixing Manifold Overfitting

### 4.1 The Two-Step Correctness Theorem

The previous section motivates the development of likelihood-based methods which work correctly even in the presence of dimensionality mismatch. Intuitively, fixing the mismatch should be enough, which suggests ($i$) first reducing the dimension of the data to some $d$-dimensional representation, and then ($ii$) applying maximum-likelihood density estimation on the lower-dimensional dataset. The following theorem, where $\mu_d$ denotes the Lebesgue measure on $\mathbb{R}^d$, confirms that this intuition is correct.

**Theorem 2 (Two-Step Correctness):** Let $\mathcal{M} \subseteq \mathbb{R}^D$ be a $C^1$ $d$-dimensional embedded submanifold of $\mathbb{R}^D$, and let $\mathbb{P}^*$ be a distribution on $\mathcal{M}$. Assume there exist measurable functions $G : \mathbb{R}^d \to \mathbb{R}^D$ and $g : \mathbb{R}^D \to \mathbb{R}^d$ such that $G(g(x)) = x$, $\mathbb{P}^*$-almost surely. Then:

1. $G_\#(g_\#\mathbb{P}^*) = \mathbb{P}^*$, where $h_\#\mathbb{P}$ denotes the pushforward of measure $\mathbb{P}$ through the function $h$.

2. Moreover, if $\mathbb{P}^*$ is smooth, and $G$ and $g$ are $C^1$, then:
   (a) $g_\#\mathbb{P}^* \ll \mu_d$.
   (b) $G(g(x)) = x$ for every $x \in \mathcal{M}$, and the functions $\tilde{g} : \mathcal{M} \to g(\mathcal{M})$ and $\tilde{G} : g(\mathcal{M}) \to \mathcal{M}$ given by $\tilde{g}(x) = g(x)$ and $\tilde{G}(z) = G(z)$ are diffeomorphisms and inverses of each other.

**Proof:** See App. B.3.

We now discuss the implications of Theorem 2.

**Assumptions and correctness** The condition $G(g(x)) = x$, $\mathbb{P}^*$-almost surely, is what one should expect to obtain during the dimensionality reduction step, for example through an autoencoder (AE) (Rumelhart et al., 1985) where $\mathbb{E}_{X\sim\mathbb{P}^*}[\|G(g(X)) - X\|_2^2]$ is minimized over $G$ and $g$, provided these have enough capacity and that population-level expectations can be minimized. We do highlight however that we allow for a much more general class of procedures than just autoencoders, nonetheless we still refer to $g$ and $G$ as the "encoder" and "decoder", respectively. Part 1, $G_\#(g_\#\mathbb{P}^*) = \mathbb{P}^*$, justifies using a first step where $g$ reduces the dimension of the data, and then having a second step attempting to learn the low-dimensional distribution $g_\#\mathbb{P}^*$: if a model $\mathbb{P}_Z$ on $\mathbb{R}^d$ matches the encoded data distribution, i.e. $\mathbb{P}_Z = g_\#\mathbb{P}^*$, it follows that $G_\#\mathbb{P}_Z = \mathbb{P}^*$. In other words, matching the distribution of encoded data and then decoding recovers the target distribution.

Part 2a guarantees that maximum-likelihood can be used to learn $g_\#\mathbb{P}^*$: note that if the model $\mathbb{P}_Z$ is such that $\mathbb{P}_Z \ll \mu_d$ with density (i.e. Radon-Nikodym derivative) $p_Z = \mathrm{d}\mathbb{P}_Z/\mathrm{d}\mu_d$, and $g_\#\mathbb{P}^* \ll \mu_d$, then both distributions are dominated by $\mu_d$. Their KL divergence can then be expressed in terms of their densities:

$$\mathbb{KL}(g_\#\mathbb{P}^*||\mathbb{P}_Z) = \int_{g(\mathcal{M})} p_Z^* \log \frac{p_Z^*}{p_Z} \mathrm{d}\mu_d, \tag{1}$$

where $p_Z^* = \mathrm{d}g_\#\mathbb{P}^*/\mathrm{d}\mu_d$ is the density of the encoded ground truth distribution. Assuming that $|\int_{g(\mathcal{M})} p_Z^* \log p_Z^* \mathrm{d}\mu_d| < \infty$, the usual decomposition of KL divergence into expected log-likelihood and entropy applies, and it thus follows that maximum-likelihood over $p_Z$ is once again equivalent to minimizing $\mathbb{KL}(g_\#\mathbb{P}^*||\mathbb{P}_Z)$ over $\mathbb{P}_Z$. In other words, learning the distribution of encoded data through maximum-likelihood with a flexible density approximator such as a VAE, AVB, NF, EBM, or ARM, and then decoding the result is a valid way of learning $\mathbb{P}^*$ which avoids manifold overfitting.

**Density evaluation** Part 2b of the two-step correctness theorem bears some resemblance to injective NFs. However, note that the theorem does not imply $G$ is injective: it only implies its restriction to $g(\mathcal{M})$, $G|_{g(\mathcal{M})}$, is injective (and similarly for $g$).

Fig. 3 exemplifies how this can happen even if $g$ and $G$ are not injective. As with injective NFs, the density $p_X$ of $G_\#\mathbb{P}_Z$ for a model $\mathbb{P}_Z$ on $g(\mathcal{M})$ is given by the injective change-of-variable formula:[5]

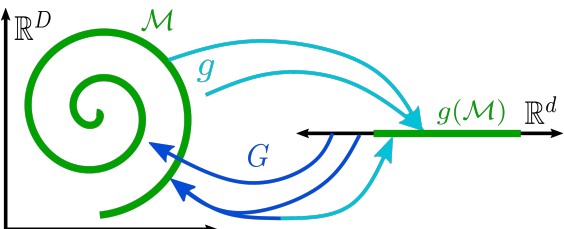

$$p_X(x) = p_Z(g(x)) \left| \det J_G^\top(g(x)) J_G(g(x)) \right|^{-\frac{1}{2}}, \quad (2)$$

for $x \in \mathcal{M}$, where $J_G(g(x)) \in \mathbb{R}^{D \times d}$ is the Jacobian matrix of $G$ evaluated at $g(x)$. Practically, this observation enables density evaluation of a trained two-step model, for example for OOD detection. Implementation-wise, we use the approach proposed

Figure 3: Illustration of how $g$ and $G$ can biject between $\mathcal{M}$ (spiral) and $g(\mathcal{M})$ (line segment) while not being fully bijective between $\mathbb{R}^D$ and $\mathbb{R}^d$.

by Caterini et al. (2021) in the context of injective NFs, which uses forward-mode automatic differentiation (Baydin et al., 2018) to efficiently construct the Jacobian in (2). We highlight that, unlike Caterini et al. (2021), we do not train our models through (2). Furthermore, injectivity is not enforced in $G$, but rather achieved at optimality of the encoder/decoder pair, and only on $g(\mathcal{M})$.

### 4.2 Generalized Autoencoders

We now explain different approaches for obtaining $G$ and $g$. As previously mentioned, a natural choice would be an AE minimizing $\mathbb{E}_{X \sim \mathbb{P}^*}[\|G(g(X)) - X\|_2^2]$ over $G$ and $g$. However, many other choices are also valid. We call a *generalized autoencoder* (GAE) any procedure in which both ($i$) low-dimensional representations $z_n = g(x_n)$ are recovered for $n = 1, \ldots, N$, and ($ii$) a function $G$ is learned with the intention that $G(z_n) = x_n$ for $n = 1, \ldots, N$.

As alternatives to an AE, some DGMs can be used as GAEs, either because they directly provide $G$ and $g$ or can be easily modified to do so. These methods alone might obtain a $G$ which correctly maps to $\mathcal{M}$, but might not be correctly recovering $\mathbb{P}^*$. From the manifold overfitting theorem, this is what we should expect from likelihood-based models, and we argue it is not unreasonable to expect from other models as well. For example, the high quality of samples generated from adversarial methods (Brock et al., 2019) suggests they are indeed learning $\mathcal{M}$, but issues such as mode collapse (Che et al., 2017) suggest they might not be recovering $\mathbb{P}^*$ (Arbel et al., 2021). Among other options (Wang et al., 2020), we can use the following explicit DGMs as GAEs: ($i$) VAEs or ($ii$) AVB, using the mean of the encoder as $g$ and the mean of the decoder as $G$. We can also use the following implicit DGMs as GAEs: ($iii$) Wasserstein autoencoders (WAEs) (Tolstikhin et al., 2018) or any of its follow-ups (Kolouri et al., 2018; Patrini et al., 2020), again using the decoder as $G$ and the encoder as $g$, ($iv$) bidirectional GANs (BiGANs) (Donahue et al., 2017; Dumoulin et al., 2017), taking $G$ as the generator and $g$ as the encoder, or ($v$) any GAN, by fixing $G$ as the generator and then learning $g$ by minimizing reconstruction error $\mathbb{E}_{X \sim \mathbb{P}^*}[\|G(g(X)) - X\|_2^2]$.

Note that explicit construction of $g$ can be avoided as long as the representations $\{z_n\}_{n=1}^N$ are learned, which could be achieved through non-amortized models (Gershman & Goodman, 2014; Kim et al., 2018), or with optimization-based GAN inversion methods (Xia et al., 2021).

We summarize our two-step procedure class once again:

1. Learn $G$ and $\{z_n\}_{n=1}^N$ from $\{x_n\}_{n=1}^N$ with a GAE.

2. Learn $p_Z$ from $\{z_n\}_{n=1}^N$ with a likelihood-based DGM.

The final model is then given by pushing $p_Z$ forward through $G$. Any choice of GAE and likelihood-based DGM gives a valid instance of a two-step procedure. Note that $G$, and $g$ if it is also explicitly constructed, are fixed throughout the second step.

---

[5]The density $p_X$ is with respect to the Riemannian measure on $\mathcal{M}$ corresponding to the Riemannian metric inherited from $\mathbb{R}^D$. This measure can be understood as the volume form on $\mathcal{M}$ in that integrating against them yields the same results.

## 5 Towards Unifying Deep Generative Models

**Making implicit models explicit**   As noted above, some DGMs are themselves GAEs, including some implicit models for which density evaluation is not typically available, such as WAEs, BiGANs, and GANs. Ramesh & LeCun (2018) use (2) to train implicit models, but they do not train a second-step DGM and thus have no mechanism to encourage trained models to satisfy the change-of-variable formula. Dieng et al. (2019) aim to provide GANs with density evaluation, but add $D$-dimensional Gaussian noise in order to achieve this, resulting in an adversarially-trained explicit model, rather than truly making an implicit model explicit. The two-step correctness theorem not only fixes manifold overfitting for explicit likelihood-based DGMs, but also enables density evaluation for these implicit models through (2) once a low-dimensional likelihood model has been trained on $g(\mathcal{M})$. We highlight the relevance of training the second-step model $p_Z$ for (2) to hold: even if $G$ mapped some base distribution on $\mathbb{R}^d$, e.g. a Gaussian, to $\mathbb{P}^*$, it need not be injective to achieve this, and could map distinct inputs to the same point on $\mathcal{M}$ (see Fig. 3). Such a $G$ could be the result of training an implicit model, e.g. a GAN, which correctly learned its target distribution. Training $g$, and $p_Z$ on $g(\mathcal{M}) \subseteq \mathbb{R}^d$, is still required to ensure $G|_{g(\mathcal{M})}$ is injective and (2) can be applied, even if the end result of this additional training is that the target distribution remains properly learned. Endowing implicit models with density evaluation addresses a significant downside of these models, and we show in Sec. 6.3 how this newfound capability can be used for OOD detection.

**Two-step procedures**   Several methods can be seen through the lens of our two-step approach, and can be interpreted as addressing manifold overfitting thanks to Theorem 2. Dai & Wipf (2019) use a two-step VAE, where both the GAE and DGM are taken as VAEs. Xiao et al. (2019) use a standard AE along with an NF. Brehmer & Cranmer (2020), and Kothari et al. (2021) use an AE as the GAE where $G$ is an injective NF and $g$ its left inverse and use an NF as the DGM. Ghosh et al. (2020) use an AE with added regularizers along with a Gaussian mixture model. Rombach et al. (2022) use a VAE along with a diffusion model (Ho et al., 2020) and obtain highly competitive empirical performance, which is justified by our theoretical results.

Other methods, while not exact instances, are philosophically aligned. Razavi et al. (2019) first obtain discrete low-dimensional representations of observed data and then train an ARM on these, which is similar to a discrete version of our own approach. Arbel et al. (2021) propose a model which they show is equivalent to pushing forward a low-dimensional EBM through $G$. The design of this model fits squarely into our framework, although a different training procedure is used.

The methods of Zhang et al. (2020c), Caterini et al. (2021), and Ross & Cresswell (2021) simultaneously optimize $G$, $g$, and $p_Z$ rather than using a two-step approach, combining in their loss a reconstruction term with a likelihood term as in (2). The validity of these methods however is not guaranteed by the two-step correctness theorem, and we believe a theoretical understanding of their objectives to be an interesting direction for future work.

## 6 Experiments

We now experimentally validate the advantages of our proposed two-step procedures across a variety of settings. We use the nomenclature A+B to refer to the two-step model with A as its GAE and B as its DGM. All experimental details are provided in App. C, including a brief summary of the losses of the individual models we consider. For all experiments on images, we set $d = 20$ as a hyperparameter,[6] which we did not tune. We chose this value as it was close to the intrinsic dimension estimates obtained by Pope et al. (2021). Our code[7] provides baseline implementations of all our considered GAEs and DGMs, which we hope will be useful to the community even outside of our proposed two-step methodology.

---

[6]We slightly abuse notation when talking about $d$ for a given model, since $d$ here does not refer to the true intrinsic dimension anymore, but rather the dimension over which $p_Z$ is defined (and which $G$ maps from and $g$ maps to), which need not match the true and unknown intrinsic dimension.

[7]https://github.com/layer6ai-labs/two_step_zoo

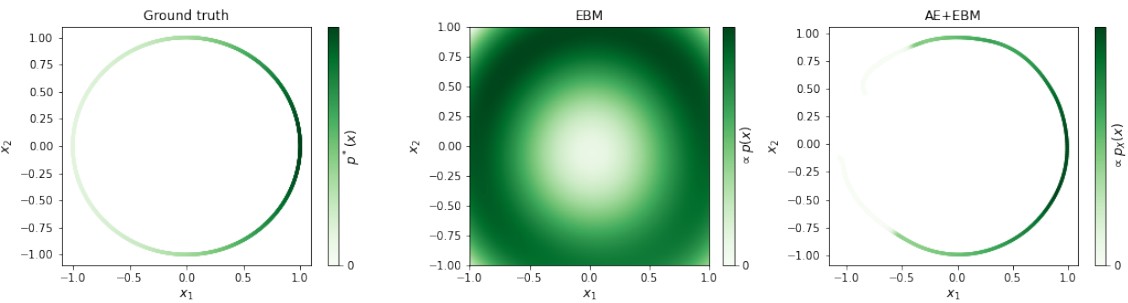

Figure 4: Results on simulated data: von Mises ground truth **(left)**, EBM **(middle)**, and AE+EBM **(right)**.

## 6.1 Simulated Data

We consider a von Mises distribution on the unit circle in Fig. 4. We learn this distribution both with an EBM and a two-step AE+EBM model. While the EBM indeed concentrates mass around the circle, it assigns higher density to an incorrect region of it (the top, rather than the right), corroborating manifold overfitting. The AE+EBM model not only learns the manifold more accurately, it also assigns higher likelihoods to the correct part of it. We show additional results on simulated data in App. D.1, where we visually confirm that the reason two-step models outperform single-step ones trained through maximum-likelihood is the data being supported on a low-dimensional manifold.

## 6.2 Comparisons Against Maximum-Likelihood

We now show that our two-step methods empirically outperform maximum-likelihood training. Conveniently, some likelihood-based DGMs recover low-dimensional representations and hence are GAEs too, providing the opportunity to compare two-step training and maximum-likelihood training directly. In particular, AVB and VAEs both maximize a lower bound of the log-likelihood, so we can train a first model as a GAE, recover low-dimensional representations, and then train a second-step DGM. Any performance difference compared to maximum-likelihood is then due to the second-step DGM rather than the choice of GAE.

We show the results in Table 1 for MNIST, FMNIST (Xiao et al., 2017), SVHN (Netzer et al., 2011), and CIFAR-10 (Krizhevsky, 2009). We use Gaussian decoders with learnable scalar variance for both models, even for MNIST and FMNIST, as opposed to Bernoulli or other common choices (Loaiza-Ganem & Cunningham, 2019) in order to properly model the data as continuous and allow for manifold overfitting to happen. While ideally we would compare models based on log-likelihood, this is only sensible for models sharing the same dominating measure;

Table 1: FID scores (lower is better). Means ± standard errors across 3 runs are shown. The superscript "+" indicates a larger model, and the subscript "$\sigma$" indicates added Gaussian noise. Unreliable FID scores are highlighted in red (see text for description).

| MODEL | MNIST | FMNIST | SVHN | CIFAR-10 |
|---|---|---|---|---|
| AVB | $219.0 \pm 4.2$ | $235.9 \pm 4.5$ | $356.3 \pm 10.2$ | $289.0 \pm 3.0$ |
| AVB$^+$ | $205.0 \pm 3.9$ | $216.2 \pm 3.9$ | $352.6 \pm 7.6$ | $297.1 \pm 1.1$ |
| AVB$_\sigma^+$ | $205.2 \pm 1.0$ | $223.8 \pm 5.4$ | $353.0 \pm 7.2$ | $305.8 \pm 8.7$ |
| AVB+ARM | $86.4 \pm 0.9$ | $78.0 \pm 0.9$ | $56.6 \pm 0.6$ | $182.5 \pm 1.0$ |
| AVB+AVB | $133.3 \pm 0.9$ | $143.9 \pm 2.5$ | $74.5 \pm 2.5$ | $183.9 \pm 1.7$ |
| AVB+EBM | $96.6 \pm 3.0$ | $103.3 \pm 1.4$ | $61.5 \pm 0.8$ | $189.7 \pm 1.8$ |
| AVB+NF | $83.5 \pm 2.0$ | $77.3 \pm 1.1$ | $55.4 \pm 0.8$ | $181.7 \pm 0.8$ |
| AVB+VAE | $106.2 \pm 2.5$ | $105.7 \pm 0.6$ | $59.9 \pm 1.3$ | $186.7 \pm 0.9$ |
| VAE | $197.4 \pm 1.5$ | $188.9 \pm 1.8$ | $311.5 \pm 6.9$ | $270.3 \pm 3.2$ |
| VAE$^+$ | $184.0 \pm 0.7$ | $179.1 \pm 0.2$ | $300.1 \pm 2.1$ | $257.8 \pm 0.6$ |
| VAE$_\sigma^+$ | $185.9 \pm 1.8$ | $183.4 \pm 0.7$ | $302.2 \pm 2.0$ | $257.8 \pm 1.7$ |
| VAE+ARM | $69.7 \pm 0.8$ | $70.9 \pm 1.0$ | $52.9 \pm 0.3$ | $175.2 \pm 1.3$ |
| VAE+AVB | $117.1 \pm 0.8$ | $129.6 \pm 3.1$ | $64.0 \pm 1.3$ | $176.7 \pm 2.0$ |
| VAE+EBM | $74.1 \pm 1.0$ | $78.7 \pm 2.2$ | $63.7 \pm 3.3$ | $181.7 \pm 2.8$ |
| VAE+NF | $70.3 \pm 0.7$ | $73.0 \pm 0.3$ | $52.9 \pm 0.3$ | $175.1 \pm 0.9$ |
| ARM$^+$ | $98.7 \pm 10.6$ | $72.7 \pm 2.1$ | $168.3 \pm 4.1$ | $162.6 \pm 2.2$ |
| ARM$_\sigma^+$ | $34.7 \pm 3.1$ | $23.1 \pm 0.9$ | $149.2 \pm 10.7$ | $136.1 \pm 4.2$ |
| AE+ARM | $72.0 \pm 1.3$ | $76.0 \pm 0.3$ | $60.1 \pm 3.0$ | $186.9 \pm 1.0$ |
| EBM$^+$ | $84.2 \pm 4.3$ | $135.6 \pm 1.6$ | $228.4 \pm 5.0$ | $201.4 \pm 7.9$ |
| EBM$_\sigma^+$ | $101.0 \pm 12.3$ | $135.3 \pm 0.9$ | $235.0 \pm 5.6$ | $200.6 \pm 4.8$ |
| AE+EBM | $75.4 \pm 2.3$ | $83.1 \pm 1.9$ | $75.2 \pm 4.1$ | $187.4 \pm 3.7$ |

here this is not the case as the single-step models are $D$-dimensional, while our two-step models are not. We thus use the FID score (Heusel et al., 2017) as a measure of how well models recover $\mathbb{P}^*$. Table 1 shows that our two-step procedures consistently outperform single-step maximum-likelihood training, even when adding Gaussian noise to the data, thus highlighting that manifold overfitting is still an empirical issue even

when the ground truth distribution is $D$-dimensional but highly peaked around a manifold. We emphasize that we did not tune our two-step models, and thus the takeaway from Table 1 should not be about which combination of models is the best performing one, but rather how consistently two-step models outperform single-step models trained through maximum-likelihood. We also note that some of the baseline models are significantly larger, e.g. the VAE$^+$ on MNIST has approximately 824k parameters, while the VAE model has 412k, and the VAE+EBM only 416k. The parameter efficiency of two-step models highlights that our empirical gains are not due to increasing model capacity but rather from addressing manifold overfitting. We show in App. C.4.3 a comprehensive list of parameter counts, along with an accompanying discussion.

Table 1 also shows comparisons between single and two-step models for ARMs and EBMs, which unlike AVB and VAEs, are not GAEs themselves; we thus use an AE as the GAE for these comparisons. Although FID scores did not consistently improve for these two-step models over their corresponding single-step baselines, we found the visual quality of samples was significantly better for almost all two-step models, as demonstrated in the first two columns of Fig. 5, and by the additional samples shown in App. D.2. We thus highlight with red the corresponding FID scores as unreliable in Table 1. We believe these failures modes of the FID metric itself, wherein the scores do not correlate with visual quality, emphasize the importance of further research on sample-based scalar evaluation metrics for DGMs (Borji, 2022), although developing such metrics falls outside our scope. We also show comparisons using precision and recall (Kynkäänniemi et al., 2019) in App. D.4, and observe that two-step models still outperform single-step ones.

We also point out that one-step EBMs exhibited training difficulties consistent with maximum-likelihood non-convergence (App. D.3). Meanwhile, Langevin dynamics (Welling & Teh, 2011) for AE+EBM exhibit better and faster convergence, yielding good samples even when not initialized from the training buffer (see Fig. 13 in App. D.3), and AE+ARM speeds up sampling over the baseline ARM by a factor of $\mathcal{O}(D/d)$, in both cases because there are fewer coordinates in the sample space. All of the 44 two-step models shown in Table 1 visually outperformed their single-step counterparts (App. D.2), empirically corroborating our theoretical findings.

Finally, we have omitted some comparisons verified in prior work: Dai & Wipf (2019) show VAE+VAE outperforms VAE, and Xiao et al. (2019) that AE+NF outperforms NF. We also include some preliminary experiments where we attempted to improve upon a GAN's generative performance on high resolution images in App. D.5. We used an optimization-based GAN inversion method, but found the reconstruction errors were too large to enable empirical improvements from adding a second-step model.

## 6.3 OOD Detection with Implicit Models

Having verified that, as predicted by Theorem 2, two-step models outperform maximum-likelihood training, we now turn our attention to the other consequence of this theorem, namely endowing implicit models with density evaluation after training a second-step DGM. We demonstrate that our approach advances fully-unsupervised likelihood-based out-of-distribution detection. Nalisnick et al. (2019) discovered the counter-intuitive phenomenon that likelihood-based DGMs sometimes assign higher likelihoods to OOD data than to in-distribution data. In particular, they found models trained on FMNIST and CIFAR-10 assigned higher likelihoods to MNIST and SVHN, respectively. While there has been a significant amount of research trying to remedy and explain this situation (Choi et al., 2018; Ren et al., 2019; Zisselman & Tamar, 2020; Zhang et al., 2020a; Kirichenko et al., 2020; Le Lan & Dinh, 2020; Caterini & Loaiza-Ganem, 2021), there is little work achieving good OOD performance using only likelihoods of models trained in a fully-unsupervised way to recover $\mathbb{P}^*$ rather than explicitly trained for OOD detection. Caterini et al. (2021) achieve improvements in this regard, although their method remains computationally expensive and has issues scaling (e.g. no results are reported on the CIFAR-10 $\rightarrow$ SVHN task).

We train several two-step models where the GAE is either a BiGAN or a WAE, which do not by themselves allow for likelihood evaluation, and then use the resulting log-likelihoods (or lower bounds/negative energy functions) for OOD detection. Two-step models allow us to use either the high-dimensional $\log p_X$ from (2) or low-dimensional $\log p_Z$ as metrics for this task. We conjecture that the latter is more reliable, since ($i$) the base measure is always $\mu_d$, and ($ii$) the encoder-decoder is unlikely to exactly satisfy the conditions of Theorem 2. Hence, we use $\log p_Z$ here, and show results for $\log p_X$ in App. D.6.

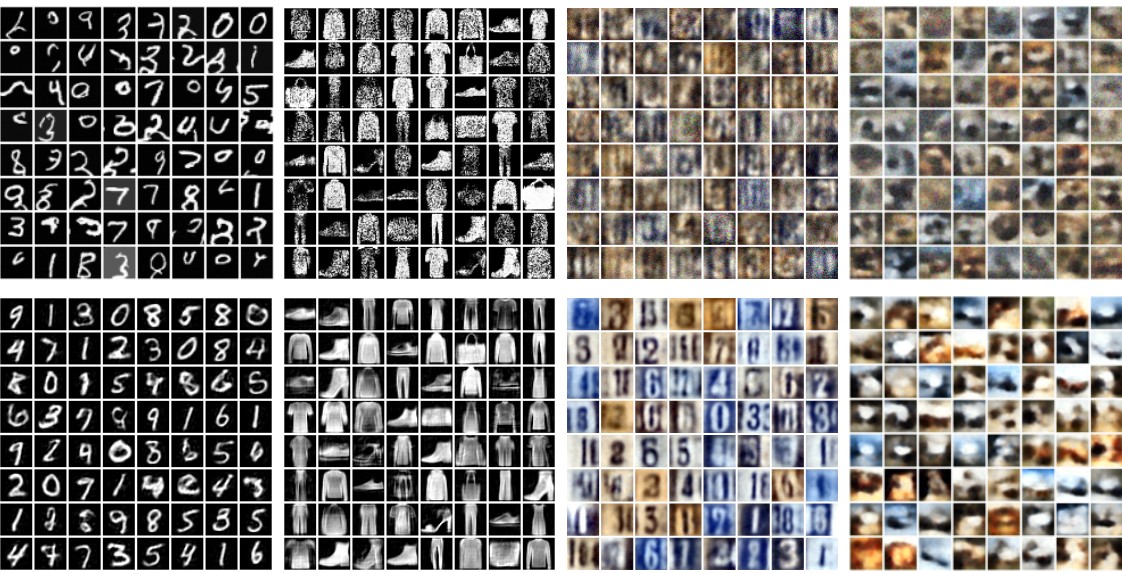

Figure 5: Uncurated samples from single-step models (**first row**, showing $\text{ARM}_\sigma^+$, $\text{EBM}^+$, $\text{AVB}^+$, and VAE) and their respective two-step counterparts (**second row**, showing AE+ARM, AE+EBM, AVB+NF, and VAE+AVB), for MNIST (**first column**), FMNIST (**second column**), SVHN (**third column**), and CIFAR-10 (**fourth column**).

Table 2 shows the (balanced) classification accuracy of a decision stump given only the log-likelihood; we show some corresponding histograms in App. D.6. The stump is forced to assign large likelihoods as in-distribution, so that accuracies below 50% indicate it incorrectly assigned higher likelihoods to OOD data. We correct the classification accuracy to account for datasets of different size (details in App. D.6), resulting in an easily interpretable metric which can be understood as the expected classification accuracy if two same-sized samples of in-distribution and OOD data were compared. Not only did we enable implicit models to perform OOD detection, but we also outperformed likelihood-based single-step models in this setting. To the best of our knowledge, no other model achieves nearly 50% (balanced) accuracy on CIFAR-10→SVHN using *only* likelihoods. Although admittedly the problem is not yet solved, we have certainly made progress on a challenging task for fully-unsupervised methods.

Table 2: OOD (balanced) classification accuracy as a percentage (higher is better). Means $\pm$ standard errors across 3 runs are shown. Arrows point from in-distribution to OOD data.

| MODEL | FMNIST → MNIST | CIFAR-10 → SVHN |
|---|---|---|
| $\text{ARM}^+$ | $9.9 \pm 0.6$ | $15.5 \pm 0.0$ |
| BiGAN+ARM | $81.9 \pm 1.4$ | $38.0 \pm 0.2$ |
| WAE+ARM | $69.8 \pm 13.9$ | $40.1 \pm 0.2$ |
| $\text{AVB}^+$ | $96.0 \pm 0.5$ | $23.4 \pm 0.1$ |
| BiGAN+AVB | $59.5 \pm 3.1$ | $36.4 \pm 2.0$ |
| WAE+AVB | $90.7 \pm 0.7$ | $43.5 \pm 1.9$ |
| $\text{EBM}^+$ | $32.5 \pm 1.1$ | $46.4 \pm 3.1$ |
| BiGAN+EBM | $51.2 \pm 0.2$ | $48.8 \pm 0.1$ |
| WAE+EBM | $57.2 \pm 1.3$ | $49.3 \pm 0.2$ |
| $\text{NF}^+$ | $36.4 \pm 0.2$ | $18.6 \pm 0.3$ |
| BiGAN+NF | $84.2 \pm 1.0$ | $40.1 \pm 0.2$ |
| WAE+NF | $95.4 \pm 1.6$ | $46.1 \pm 1.0$ |
| $\text{VAE}^+$ | $96.1 \pm 0.1$ | $23.8 \pm 0.2$ |
| BiGAN+VAE | $59.7 \pm 0.2$ | $38.1 \pm 0.1$ |
| WAE+VAE | $92.5 \pm 2.7$ | $41.4 \pm 0.2$ |

For completeness, we show samples from these models in App. D.2 and FID scores in App. D.4. Implicit models see less improvement in FID from adding a second-step DGM than explicit models, suggesting that manifold overfitting is a less dire problem for implicit models. Nonetheless, we do observe some improvements, particularly for BiGANs, hinting that our two-step methodology not only endows these models with density evaluation, but that it can also improve their generative performance. We further show in App. D.6 that OOD improvements obtained by two-step models apply to explicit models as well.

Interestingly, whereas the VAEs used in Nalisnick et al. (2019) have Bernoulli likelihoods, we find that our single-step likelihood-based Gaussian-decoder VAE and AVB models perform quite well on distinguishing

FMNIST from MNIST, yet still fail on the CIFAR-10 task. Studying this is of future interest but is outside the scope of this work.

## 7   Conclusions, Scope, and Limitations

In this paper we diagnosed manifold overfitting, a fundamental problem of maximum-likelihood training with flexible densities when the data lives in a low-dimensional manifold. We proposed to fix manifold overfitting with a class of two-step procedures which remedy the issue, theoretically justify a large group of existing methods, and endow implicit models with density evaluation after training a low-dimensional likelihood-based DGM on encoded data.

Our two-step correctness theorem remains nonetheless a nonparametric result. In practice, the reconstruction error will be positive, i.e. $\mathbb{E}_{X\sim\mathbb{P}^*}[\|G(g(X)) - X\|_2^2] > 0$. Note that this can happen even when assuming infinite capacity, as $\mathcal{M}$ needs to be diffeomorphic to $g(\mathcal{M})$ for some $C^1$ function $g : \mathbb{R}^D \to \mathbb{R}^d$ for the reconstruction error to be 0. We leave a study of learnable topologies of $\mathcal{M}$ for future work. The density in (2) might then not be valid, either if the reconstruction error is positive, or if $p_Z$ assigns positive probability outside of $g(\mathcal{M})$. However, we note that our approach at least provides a mechanism to encourage our trained encoder-decoder pair to invert each other, suggesting that (2) might not be too far off. We also believe that a finite-sample extension of our result, while challenging, would be a relevant direction for future work. We hope our work will encourage follow-up research exploring different ways of addressing manifold overfitting, or its interaction with the score-matching objective.

Finally, we treated $d$ as a hyperparameter, but in practice $d$ is unknown and improvements can likely be had by estimating it (Levina & Bickel, 2004), as overspecifying it should not fully remove manifold overfitting, and underspecifying it would make learning $\mathcal{M}$ mathematically impossible. Still, we observed significant empirical improvements across a variety of tasks and datasets, demonstrating that manifold overfitting is not just a theoretical issue in DGMs, and that two-step methods are an important class of procedures to deal with it.

### Broader Impact Statement

Generative modelling has numerous applications besides image generation, including but not limited to: audio generation (van den Oord et al., 2016a; Engel et al., 2017), biology (Lopez et al., 2020), chemistry (Gómez-Bombarelli et al., 2018), compression (Townsend et al., 2019; Ho et al., 2019; Golinski & Caterini, 2021; Yang et al., 2022), genetics (Riesselman et al., 2018), neuroscience (Sussillo et al., 2016; Gao et al., 2016; Loaiza-Ganem et al., 2019), physics (Otten et al., 2021; Padmanabha & Zabaras, 2021), text generation (Bowman et al., 2016; Devlin et al., 2019; Brown et al., 2020), text-to-image generation (Zhang et al., 2017; Ramesh et al., 2022; Saharia et al., 2022), video generation (Vondrick et al., 2016; Weissenborn et al., 2020), and weather forecasting (Ravuri et al., 2021). While each of these applications can have positive impacts on society, it is also possible to apply deep generative models inappropriately, or create negative societal impacts through their use (Brundage et al., 2018; Urbina et al., 2022). When datasets are biased, accurate generative models will inherit those biases (Steed & Caliskan, 2021; Humayun et al., 2022). Inaccurate generative models may introduce new biases not reflected in the data. Our paper addresses a ubiquitous problem in generative modelling with maximum likelihood estimation – manifold overfitting – that causes models to fail to learn the distribution of data correctly. In this sense, correcting manifold overfitting should lead to more accurate generative models, and representations that more closely reflect the data.

### Acknowledgments

We thank the anonymous reviewers whose suggestions helped improved our work. In particular, we thank anonymous reviewer Cev4, as well as Taiga Abe, both of whom pointed out the mixture of two Gaussians regular overfitting example from Bishop (2006), which was lacking from a previous version of our manuscript. We wrote our code in Python (Van Rossum & Drake, 2009), and specifically relied on the following packages: Matplotlib (Hunter, 2007), TensorFlow (Abadi et al., 2015) (particularly for TensorBoard), Jupyter Notebook (Kluyver et al., 2016), PyTorch (Paszke et al., 2019), nflows (Durkan et al., 2020), NumPy (Harris et al., 2020), prdc (Naeem et al., 2020), pytorch-fid (Seitzer, 2020), and functorch (He & Zou, 2021).

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

# A    Informal Measure Theory Primer

Before stating Theorems 1 and 2, and studying their implications, we provide a brief tutorial on some aspects of measure theory that are relevant to follow our discussion. This review is not meant to be comprehensive, and we prioritize intuition over formalism. Readers interested in the topic may consult textbooks such as Billingsley (2008).

## A.1    Probability Measures

Let us first motivate the need for measure theory in the first place and consider the question: *what is a density*? Intuitively, the density $p_X$ of a random variable $X$ is a function having the property that integrating $p_X$ over any set $A$ gives back the probability that $X \in A$. This density characterizes the distribution of $X$, in that it can be used to answer any probabilistic question about $X$. It is common knowledge that discrete random variables are not specified through a density, but rather a probability mass function. Similarly, in our setting, where $X$ might always take values in $\mathcal{M}$, such a density will not exist. To see this, consider the case where $A = \mathcal{M}$, so that the integral of $p_X$ over $\mathcal{M}$ would have to be 1, which cannot happen since $\mathcal{M}$ has volume 0 in $\mathbb{R}^D$ (or more formally, Lebesgue measure 0). Measure theory provides the tools necessary to properly specify *any distribution*, subsuming as special cases probability mass functions, densities of continuous random variables, and distributions on manifolds.

A measure $\mu$ on $\mathbb{R}^D$ is a function mapping subsets $A \subseteq \mathbb{R}^D$ to $\mathbb{R}_{\geq 0}$, obeying the following properties: $(i)$ $\mu(A) \geq 0$ for every $A$, $(ii)$ $\mu(\emptyset) = 0$, where $\emptyset$ denotes the empty set, and $(iii)$ $\mu(\cup_{k=1}^\infty A_k) = \sum_{k=1}^\infty \mu(A_k)$ for any sequence of pairwise disjoint sets $A_1, A_2, \ldots$ (i.e. $A_i \cap A_j = \emptyset$ whenever $i \neq j$). Note that most measures of interest are only defined over a large class of subsets of $\mathbb{R}^D$ (called $\sigma$-algebras, the most notable one being the Borel $\sigma$-algebra) rather than for every possible subset due to technical reasons, but we omit details in the interest of better conveying intuition. A measure is called a *probability* measure if it also satisfies $\mu(\mathbb{R}^D) = 1$. To any random variable $X$ corresponds a probability measure $\mu_X$, having the property that $\mu_X(A)$ is the probability that $X \in A$ for any $A$. Analogously to probability mass functions or densities of continuous random variables, $\mu_X$ allows us to answer any probabilistic question about $X$. The probability measure $\mu_X$ is often called the *distribution* or *law* of $X$. Throughout our paper, $\mathbb{P}^*$ is the distribution from which we observe data.

Let us consider two examples to show how probability mass functions and densities of continuous random variables are really just specifying distributions. Given $a_1, \ldots, a_K \in \mathbb{R}^D$, consider the probability mass function of a random variable $X$ given by $p_X(x) = 1/K$ for $x = a_1, a_2, \ldots, a_K$ and 0 otherwise. This probability mass function is simply specifying the distribution $\mu_X(A) = 1/K \cdot \sum_{k=1}^K \mathbb{1}(a_k \in A)$, where $\mathbb{1}(\cdot \in A)$ denotes the indicator function for $A$, i.e. $\mathbb{1}(a \in A)$ is 1 if $a \in A$, and 0 otherwise. Now consider a standard Gaussian random variable $X$ in $\mathbb{R}^D$ with density $p_X(x) = \mathcal{N}(x; 0, I_D)$. Similarly to how the probability mass function from the previous example characterized a distribution, this density does so as well through $\mu_X(A) = \int_A \mathcal{N}(x; 0, I_D)\mathrm{d}x$. We will see in the next section how these ideas can be extended to distributions on manifolds.

The concept of integrating a function $h : \mathbb{R}^D \to \mathbb{R}$ with respect to a measure $\mu$ on $\mathbb{R}^D$ is fundamental in measure theory, and can be thought of as "weighting the inputs of $h$ according to $\mu$". In the case of the Lebesgue measure $\mu_D$ (which assigns to subsets $A$ of $\mathbb{R}^D$ their "volume" $\mu_D(A)$), integration extends the concept of Riemann integrals commonly taught in calculus courses, and in the case of random variables integration defines expectations, i.e. $\mathbb{E}_{X \sim \mu_X}[h(X)] = \int h \mathrm{d}\mu_X$. In the next section we will talk about the interplay between integration and densities.

We finish this section by explaining the relevant concept of a property holding almost surely with respect to a measure $\mu$. A property is said to hold $\mu$-*almost surely* if the set $A$ over which it does not hold is such that $\mu(A) = 0$. For example, if $\mu_X$ is the distribution of a standard Gaussian random variable $X$ in $\mathbb{R}^D$, then we can say that $X \neq 0$ holds $\mu_X$-almost surely, since $\mu_X(\{0\}) = 0$. The assumption that $G(g(x)) = x$, $\mathbb{P}^*$-almost surely in Theorem 2 thus means that $\mathbb{P}^*(\{x \in \mathbb{R}^D : G(g(x)) \neq x\}) = 0$.

## A.2 Absolute Continuity

So far we have seen that probability measures allow us to talk about distributions in full generality, and that probability mass functions and densities of continuous random variables can be used to specify probability measures. A distribution on a manifold $\mathcal{M}$ embedded in $\mathbb{R}^D$ can simply be thought of as a probability measure $\mu$ such that $\mu(\mathcal{M}) = 1$. We would like to define densities on manifolds in an analogous way to probability mass functions and densities of continuous random variables, in such a way that they allow us to characterize distributions on the manifold. Absolute continuity of measures is a concept that allows us to formalize the concept of *density with respect to a dominating measure*, and encompasses probability mass functions, densities of continuous random variables, and also allows us to define densities on manifolds. We will see that our intuitive definition of a density as a function which, when integrated over a set gives back its probability, is in fact correct, just as long as we specify the measure we integrate with respect to.

Given two measures $\mu$ and $\nu$, we say that $\mu$ is *absolutely continuous* with respect to $\nu$ if for every $A$ such that $\nu(A) = 0$, it also holds that $\mu(A) = 0$. If $\mu$ is absolutely continuous with respect to $\nu$, we also say that $\nu$ *dominates* $\mu$, and denote this property as $\mu \ll \nu$. The Radon-Nikodym theorem states that, under some mild assumptions on $\mu$ and $\nu$ which hold for all the measures considered in this paper, $\mu \ll \nu$ implies the existence of a function $h$ such that $\mu(A) = \int_A h \mathrm{d}\nu$ for every $A$. This result provides the means to formally define densities: $h$ is called the *density* or *Radon-Nikodym derivative* of $\mu$ with respect to $\nu$, and is often written as $\mathrm{d}\mu/\mathrm{d}\nu$.

Before explaining how this machinery allows us to talk about densities on manifolds, we first continue our examples to show that probability mass functions and densities of continuous random variables are Radon-Nikodym derivatives with respect to appropriate measures. Let us reconsider the example where $p_X(x) = 1/K$ for $x = a_1, a_2, \ldots, a_K$ and 0 otherwise, and $\mu_X(A) = 1/K \cdot \sum_{k=1}^{K} \mathbb{1}(a_k \in A)$. Consider the measure $\nu(A) = \sum_{k=1}^{K} \mathbb{1}(a_k \in A)$, which essentially just counts the number of $a_k$s in $A$. Clearly $\mu_X \ll \nu$, and so it follows that $\mu_X$ admits a density with respect to $\nu$. This density turns out to be $p_X$, since $\mu_X(A) = \int_A p_X \mathrm{d}\nu$. In other words, the probability mass function $p_X$ can be thought of as a Radon-Nikodym derivative, i.e. $p_X = \mathrm{d}\mu_X/\mathrm{d}\nu$. Let us now go back to the continuous density example where $p_X(x) = \mathcal{N}(x; 0, I_D)$ and $\mu_X$ is given by the Riemann integral $\mu_X(A) = \int_A \mathcal{N}(x; 0, I_D) \mathrm{d}x$. In this case, $\nu = \mu_D$, and since the Lebesgue integral extends the Riemann integral, it follows that $\mu_X(A) = \int_A p_X \mathrm{d}\mu_D$, so that the density $p_X$ is actually also a density in the formal sense of being a Radon-Nikodym derivative, so that $p_X = \mathrm{d}\mu_X/\mathrm{d}\mu_D$. We can thus see that the formal concept of density or Radon-Nikodym derivative generalizes both probability mass functions and densities of continuous random variables as we usually think of them, allowing to specify distributions in a general way.

The concept of Radon-Nikodym derivative also allows us to obtain densities on manifolds, the only missing ingredient being a dominating measure on the manifold. Riemannian measures (App. B.1) play this role on manifolds, in the same way that the Lebesgue measure plays the usual role of dominating measure to define densities of continuous random variables on $\mathbb{R}^D$.

## A.3 Weak Convergence

A key point in Theorem 1 is weak convergence of the sequence of probability measures $(\mathbb{P}_t)_{t=1}^{\infty}$ to $\mathbb{P}^\dagger$. The intuitive interpretation that this statement simply means that "$\mathbb{P}_t$ converges to $\mathbb{P}^\dagger$" is correct, although formally defining convergence of a sequence of measures is still required. Weak convergence provides such a definition, and $\mathbb{P}_t$ is said to *converge weakly* to $\mathbb{P}^\dagger$ if the sequence of scalars $\mathbb{P}_t(A)$ converges to $\mathbb{P}^\dagger(A)$ for every $A$ satisfying a technical condition (for intuitive purposes, one can think of this property as holding for every $A$). In this sense weak convergence is a very natural way of defining convergence of measures: in the limit, $\mathbb{P}_t$ will assign the same probability to every set as $\mathbb{P}^\dagger$.

## A.4 Pushforward Measures

We have seen that to a random variable $X$ in $\mathbb{R}^D$ corresponds a distribution $\mu_X$. Applying a function $h : \mathbb{R}^D \to \mathbb{R}^d$ to $X$ will result in a new random variable, $h(X)$ in $\mathbb{R}^d$, and it is natural to ask what its distribution is. This distribution is called the *pushforward measure* of $\mu_X$ through $h$, which is denoted as

$h_\#\mu_X$, and is defined as $h_\#\mu_X(B) = \mu_X(h^{-1}(B))$ for every subset $B$ of $\mathbb{R}^d$. A way to intuitively understand this concept is that if one could sample $X$ from $\mu_X$, then sampling from $h_\#\mu_X$ can be done by simply applying $h$ to $X$. Note that here $h_\#\mu_X$ is a measure on $\mathbb{R}^d$.

The concept of pushforward measure is relevant in Theorem 2 as it allows us to formally reason about e.g. the distribution of encoded data, $g_\#\mathbb{P}^*$. Similarly, for a distribution $\mathbb{P}_Z$ corresponding to our second-step model, we can reason about the distribution obtained after decoding, i.e. $G_\#\mathbb{P}_Z$.

## B Proofs

### B.1 Riemannian Measures

We begin with a quick review of a Riemannian measures. Let $\mathcal{M}$ be a $d$-dimensional Riemannian manifold with Riemannian metric $\mathfrak{g}$, and let $(U, \phi)$ be a chart. The local Riemannian measure $\mu_{\mathcal{M},\phi}^{(\mathfrak{g})}$ on $\mathcal{M}$ (with its Borel $\sigma$-algebra) is given by:

$$\mu_{\mathcal{M},\phi}^{(\mathfrak{g})}(A) = \int_{\phi(A \cap U)} \sqrt{\det\left(\mathfrak{g}\left(\frac{\partial}{\partial\phi^i}, \frac{\partial}{\partial\phi^j}\right)\right)} \mathrm{d}\mu_d \tag{3}$$

for any measurable $A \subseteq \mathcal{M}$. The Riemannian measure $\mu_{\mathcal{M}}^{(\mathfrak{g})}$ on $\mathcal{M}$ is such that:

$$\mu_{\mathcal{M}}^{(\mathfrak{g})}(A \cap U) = \mu_{\mathcal{M},\phi}^{(\mathfrak{g})}(A) \tag{4}$$

for every measurable $A \subseteq \mathcal{M}$ and every chart $(U, \phi)$.

If $\mathfrak{g}_1$ and $\mathfrak{g}_2$ are two Riemannian metrics on $\mathcal{M}$, then $\mu_{\mathcal{M}}^{(\mathfrak{g}_1)} \ll \mu_{\mathcal{M}}^{(\mathfrak{g}_2)}$ and $\mu_{\mathcal{M}}^{(\mathfrak{g}_1)}$ admits a continuous and positive density with respect to $\mu_{\mathcal{M}}^{(\mathfrak{g}_2)}$. Thus, as mentioned in the main manuscript, smoothness of probability measures is indeed independent of the choice of Riemannian metric.

Below we prove a lemma which we will later use, showing that much like the Lebesgue measure, Riemannian measures assign positive measure to nonempty open sets. While we are sure this is a known property, we could not find a proof and thus provide one.

**Lemma 1:** Let $\mathcal{M}$ be a $d$-dimensional Riemannian manifold, and $\mu_{\mathcal{M}}^{(\mathfrak{g})}$ a Riemannian measure on it. Let $A \subseteq \mathcal{M}$ be a nonempty open set in $\mathcal{M}$. Then $\mu_{\mathcal{M}}^{(\mathfrak{g})}(A) > 0$.

**Proof:** Let $(U, \phi)$ be a chart such that $U \cap A \neq \emptyset$, which exists because $A \neq \emptyset$. Clearly $U \cap A$ is open, and since $\phi$ is a diffeomorphism onto its image, it follows that $\phi(U \cap A) \subseteq \mathbb{R}^d$ is also open and nonempty, and thus $\mu_d(\phi(U \cap A)) > 0$. As a result,

$$\mu_{\mathcal{M}}^{(\mathfrak{g})}(A) \geq \mu_{\mathcal{M}}^{(\mathfrak{g})}(U \cap A) = \int_{\phi(U \cap A)} \sqrt{\det\left(\mathfrak{g}\left(\frac{\partial}{\partial\phi^i}, \frac{\partial}{\partial\phi^j}\right)\right)} \mathrm{d}\mu_d > 0, \tag{5}$$

where the last inequality follows since the integrand is positive and the integration set has positive measure.

$\square$

### B.2 Manifold Overfitting Theorem

We restate the manifold overfitting theorem below for convenience:

**Theorem 1 (Manifold Overfitting):** Let $\mathcal{M} \subset \mathbb{R}^D$ be an analytic $d$-dimensional embedded submanifold of $\mathbb{R}^D$ with $d < D$, and $\mathbb{P}^\dagger$ a smooth probability measure on $\mathcal{M}$. Then there exists a sequence of probability measures $(\mathbb{P}_t)_{t=1}^\infty$ on $\mathbb{R}^D$ such that:

1. $\mathbb{P}_t \to \mathbb{P}^\dagger$ weakly as $t \to \infty$.

2. For every $t \geq 1$, $\mathbb{P}_t \ll \mu_D$ and $\mathbb{P}_t$ admits a density $p_t : \mathbb{R}^D \to \mathbb{R}_{>0}$ with respect to $\mu_D$ such that:

   (a) $\lim_{t \to \infty} p_t(x) = \infty$ for every $x \in \mathcal{M}$.

   (b) $\lim_{t \to \infty} p_t(x) = 0$ for every $x \notin \mathrm{cl}(\mathcal{M})$, where $\mathrm{cl}(\cdot)$ denotes closure in $\mathbb{R}^D$.

Before proving the theorem, note that $\mathbb{P}^\dagger$ is a distribution on $\mathcal{M}$ and $\mathbb{P}_t$ is a distribution on $\mathbb{R}^D$, with their respective Borel $\sigma$-algebras. Weak convergence is defined for measures on the same probability space, and so we slightly abuse notation and think of $\mathbb{P}^\dagger$ as a measure on $\mathbb{R}^D$ assigning to any measurable set $A \subseteq \mathbb{R}^D$ the probability $\mathbb{P}^\dagger(A \cap \mathcal{M})$, which is well-defined as $\mathcal{M}$ is an embedded submanifold of $\mathbb{R}^D$. We do not differentiate between $\mathbb{P}^\dagger$ on $\mathcal{M}$ and $\mathbb{P}^\dagger$ on $\mathbb{R}^D$ to avoid cumbersome notation.

**Proof:** Let $Y$ be a random variable whose law is $\mathbb{P}^\dagger$, and let $(Z_t)_{t=1}^\infty$ be a sequence of i.i.d. standard Gaussians in $\mathbb{R}^D$, independent of $Y$. We assume all the variables are defined on the same probability space $(\Omega, \mathcal{F}, \mathbb{P})$. Let $X_t = Y + \sigma_t Z_t$ where $(\sigma_t)_{t=1}^\infty$ is a positive sequence converging to 0. Let $\mathbb{P}_t$ be the law of $X_t$.

First we prove 1. Clearly $\sigma_t Z_t \to 0$ in probability and $Y \to Y$ in distribution as $t \to \infty$. Since $\sigma_t Z_t$ converges in probability to a constant, it follows that $X_t \to Y$ in distribution, and thus $\mathbb{P}_t \to \mathbb{P}^\dagger$ weakly.

Now we prove that $\mathbb{P}_t \ll \mu_D$. Let $A \subseteq \mathbb{R}^D$ be a measurable set such that $\mu_D(A) = 0$. We denote the law of $\sigma_t Z_t$ as $\mathbb{G}_t$ and the Gaussian density in $\mathbb{R}^D$ with mean $m$ and covariance matrix $\Sigma$ evaluated at $w$ as $\mathcal{N}(w; m, \Sigma)$. Let $B = \{(w, y) \in \mathbb{R}^D \times \mathcal{M} : w + y \in A\}$. By Fubini's theorem:

$$\mathbb{P}_t(A) = \mathbb{P}(Y + \sigma_t Z_t \in A) = \int_B \mathrm{d}\mathbb{G}_t \times \mathbb{P}^\dagger(w, y) = \int_B \mathcal{N}(w; 0, \sigma_t^2 \mathrm{I}_D) \, \mathrm{d}\mu_D \times \mathbb{P}^\dagger(w, y) \tag{6}$$

$$= \int_{A \times \mathcal{M}} \mathcal{N}(x - y; 0, \sigma_t^2 \mathrm{I}_D) \, \mathrm{d}\mu_D \times \mathbb{P}^\dagger(x, y) = \int_\mathcal{M} \int_A \mathcal{N}(x - y; 0, \sigma_t^2 \mathrm{I}_D) \, \mathrm{d}\mu_D(x) \, \mathrm{d}\mathbb{P}^\dagger(y) \tag{7}$$

$$= \int_\mathcal{M} 0 \, \mathrm{d}\mathbb{P}^\dagger(y) = 0. \tag{8}$$

Then, $\mathbb{P}_t \ll \mu_D$, proving the first part of 2. Note also that:

$$p_t(x) = \int_\mathcal{M} \mathcal{N}(x - y; 0, \sigma_t^2 \mathrm{I}_D) \, \mathrm{d}\mathbb{P}^\dagger(y) \tag{9}$$

is a valid density for $\mathbb{P}_t$ with respect to $\mu_D$, once again by Fubini's theorem since, for any measurable set $A \subseteq \mathbb{R}^D$:

$$\int_A p_t(x) \, \mathrm{d}\mu_D(x) = \int_A \int_\mathcal{M} \mathcal{N}(x - y; 0, \sigma_t^2 \mathrm{I}_D) \, \mathrm{d}\mathbb{P}^\dagger(y) \, \mathrm{d}\mu_D(x) \tag{10}$$

$$= \int_{A \times \mathcal{M}} \mathcal{N}(x - y; 0, \sigma_t^2 \mathrm{I}_D) \mathrm{d}\mu_D \times \mathbb{P}^\dagger(x, y) = \mathbb{P}_t(A). \tag{11}$$

We now prove 2a. Since $\mathbb{P}^\dagger$ being smooth is independent of the choice of Riemannian measure, we can assume without loss of generality that the Riemannian metric $\mathfrak{g}$ on $\mathcal{M}$ is the metric inherited from thinking of $\mathcal{M}$ as a submanifold of $\mathbb{R}^D$, and we can then take a continuous and positive density $p^\dagger$ with respect to the Riemannian measure $\mu_\mathcal{M}^{(\mathfrak{g})}$ associated with this metric.

Take $x \in \mathcal{M}$ and let $B_r^{\mathcal{M}}(x) = \{y \in \mathcal{M} : d_{\mathcal{M}}^{(\mathfrak{g})}(x, y) \leq r\}$ denote the geodesic ball on $\mathcal{M}$ of radius $r$ centered at $x$, where $d_{\mathcal{M}}^{(\mathfrak{g})}$ is the geodesic distance. We then have:

$$p_t(x) = \int_{\mathcal{M}} \mathcal{N}(x - y; 0, \sigma_t^2 \mathrm{I}_D) \, \mathrm{d}\mathbb{P}^\dagger(y) \geq \int_{B_{\sigma_t}^{\mathcal{M}}(x)} \mathcal{N}(x - y; 0, \sigma_t^2 \mathrm{I}_D) \, \mathrm{d}\mathbb{P}^\dagger(y) \tag{12}$$

$$= \int_{B_{\sigma_t}^{\mathcal{M}}(x)} p^\dagger(y) \cdot \mathcal{N}(x - y; 0, \sigma_t^2 \mathrm{I}_D) \, \mathrm{d}\mu_{\mathcal{M}}^{(\mathfrak{g})}(y) \geq \int_{B_{\sigma_t}^{\mathcal{M}}(x)} \inf_{y' \in B_{\sigma_t}^{\mathcal{M}}(x)} p^\dagger(y') \mathcal{N}(x - y'; 0, \sigma_t^2 \mathrm{I}_D) \, \mathrm{d}\mu_{\mathcal{M}}^{(\mathfrak{g})}(y) \tag{13}$$

$$= \mu_{\mathcal{M}}^{(\mathfrak{g})}(B_{\sigma_t}^{\mathcal{M}}(x)) \cdot \inf_{y' \in B_{\sigma_t}^{\mathcal{M}}(x)} p^\dagger(y') \mathcal{N}(x - y'; 0, \sigma_t^2 \mathrm{I}_D) \tag{14}$$

$$\geq \mu_{\mathcal{M}}^{(\mathfrak{g})}(B_{\sigma_t}^{\mathcal{M}}(x)) \cdot \inf_{y' \in B_{\sigma_t}^{\mathcal{M}}(x)} \mathcal{N}(x - y'; 0, \sigma_t^2 \mathrm{I}_D) \cdot \inf_{y' \in B_{\sigma_t}^{\mathcal{M}}(x)} p^\dagger(y'). \tag{15}$$

Since $B_{\sigma_t}^{\mathcal{M}}(x)$ is compact in $\mathcal{M}$ for small enough $\sigma_t$ and $p^\dagger$ is continuous in $\mathcal{M}$ and positive, it follows that $\inf_{y' \in B_{\sigma_t}^{\mathcal{M}}(x)} p^\dagger(y')$ is bounded away from 0 as $t \to \infty$. It is then enough to show that as $t \to \infty$,

$$\mu_{\mathcal{M}}^{(\mathfrak{g})}(B_{\sigma_t}^{\mathcal{M}}(x)) \cdot \inf_{y' \in B_{\sigma_t}^{\mathcal{M}}(x)} \mathcal{N}(x - y'; 0, \sigma_t^2 \mathrm{I}_D) \to \infty \tag{16}$$

in order to prove that 2a holds. Let $B_r^d(0)$ denote an $L_2$ ball of radius $r$ in $\mathbb{R}^d$ centered at $0 \in \mathbb{R}^d$, and let $\mu_d$ denote the Lebesgue measure on $\mathbb{R}^d$, so that $\mu_d(B_r^d(0)) = C_d r^d$, where $C_d > 0$ is a constant depending only on $d$. It is known that $\mu_{\mathcal{M}}^{(\mathfrak{g})}(B_r^{\mathcal{M}}(x)) = \mu_d(B_r^d(0)) \cdot (1 + \mathcal{O}(r^2))$ for analytic $d$-dimensional Riemannian manifolds (Gray, 1974), and thus:

$$\mu_{\mathcal{M}}^{(\mathfrak{g})}(B_{\sigma_t}^{\mathcal{M}}(x)) \cdot \inf_{y' \in B_{\sigma_t}^{\mathcal{M}}(x)} \mathcal{N}(x - y'; 0, \sigma_t^2 \mathrm{I}_D)$$

$$= C_d \sigma_t^d \left(1 + \mathcal{O}(\sigma_t^2)\right) \cdot \inf_{y' \in B_{\sigma_t}^{\mathcal{M}}(x)} \frac{1}{\sigma_t^D (2\pi)^{D/2}} \exp\left\{-\frac{\|x - y'\|_2^2}{2\sigma_t^2}\right\} \tag{17}$$

$$= \frac{C_d}{(2\pi)^{D/2}} \cdot \left(1 + \mathcal{O}(\sigma_t^2)\right) \cdot \sigma_t^{d-D} \cdot \exp\left\{-\frac{\sup_{y' \in B_{\sigma_t}^{\mathcal{M}}(x)} \|x - y'\|_2^2}{2\sigma_t^2}\right\}. \tag{18}$$

The first term is a positive constant, and the second term converges to 1. The third term goes to infinity since $d < D$, which leaves only the last term. Thus, as long as the last term is bounded away from 0 as $t \to \infty$, we can be certain that the product of all four term goes to infinity. In particular, verifying the following equation would be enough:

$$\sup_{y' \in B_{\sigma_t}^{\mathcal{M}}(x)} \|x - y'\|_2^2 \leq \sigma_t^2. \tag{19}$$

This equation holds, since for any $x, y' \in \mathcal{M}$, it is the case that $\|x - y'\|_2 \leq d_{\mathcal{M}}^{(\mathfrak{g})}(x, y')$ as $\mathfrak{g}$ is inherited from $\mathcal{M}$ being a submanifold of $\mathbb{R}^D$.

Now we prove 2b for $p_t$. Let $x \in \mathbb{R}^D \setminus \mathrm{cl}(\mathcal{M})$. We have:

$$p_t(x) = \int_{\mathcal{M}} \mathcal{N}(x - y; 0, \sigma_t^2 \mathrm{I}_D) \, \mathrm{d}\mathbb{P}^\dagger(y) \leq \int_{\mathcal{M}} \sup_{y' \in \mathcal{M}} \mathcal{N}(x - y'; 0, \sigma_t^2 \mathrm{I}_D) \, \mathrm{d}\mathbb{P}^\dagger(y) = \sup_{y' \in \mathcal{M}} \mathcal{N}(x - y'; 0, \sigma_t^2 \mathrm{I}_D) \tag{20}$$

$$= \sup_{y' \in \mathcal{M}} \frac{1}{\sigma_t^D (2\pi)^{D/2}} \exp\left\{-\frac{\|x - y'\|_2^2}{2\sigma_t^2}\right\} = \frac{1}{\sigma_t^D (2\pi)^{D/2}} \cdot \exp\left\{-\frac{\inf_{y' \in \mathcal{M}} \|x - y'\|_2^2}{2\sigma_t^2}\right\} \xrightarrow{t \to \infty} 0, \tag{21}$$

where convergence to 0 follows from $x \notin \mathrm{cl}(\mathcal{M})$ implying that $\inf_{y' \in \mathcal{M}} \|x - y'\|_2^2 > 0$.

$\square$

### B.3 Two-Step Correctness Theorem

We restate the two-step correctness theorem below for convenience:

**Theorem 2 (Two-Step Correctness):** Let $\mathcal{M} \subseteq \mathbb{R}^D$ be a $C^1$ $d$-dimensional embedded submanifold of $\mathbb{R}^D$, and let $\mathbb{P}^*$ be a distribution on $\mathcal{M}$. Assume there exist measurable functions $G : \mathbb{R}^d \to \mathbb{R}^D$ and $g : \mathbb{R}^D \to \mathbb{R}^d$ such that $G(g(x)) = x$, $\mathbb{P}^*$-almost surely. Then:

1. $G_\#(g_\#\mathbb{P}^*) = \mathbb{P}^*$, where $h_\#\mathbb{P}$ denotes the pushforward of measure $\mathbb{P}$ through the function $h$.

2. Moreover, if $\mathbb{P}^*$ is smooth, and $G$ and $g$ are $C^1$, then:

   (a) $g_\#\mathbb{P}^* \ll \mu_d$.

   (b) $G(g(x)) = x$ for every $x \in \mathcal{M}$, and the functions $\tilde{g} : \mathcal{M} \to g(\mathcal{M})$ and $\tilde{G} : g(\mathcal{M}) \to \mathcal{M}$ given by $\tilde{g}(x) = g(x)$ and $\tilde{G}(z) = G(z)$ are diffeomorphisms and inverses of each other.

Similarly to the manifold overfitting theorem, we think of $\mathbb{P}^*$ as a distribution on $\mathbb{R}^D$, assigning to any Borel set $A \subseteq \mathbb{R}^D$ the probability $\mathbb{P}^*(A \cap \mathcal{M})$, which once again is well-defined since $\mathcal{M}$ is an embedded submanifold of $\mathbb{R}^D$.

**Proof:** We start with part 1. Let $A = \{x \in \mathbb{R}^D : G(g(x)) \neq x\}$, which is a null set under $\mathbb{P}^*$ by assumption. By applying the definition of pushforward measure twice, for any measurable set $B \subseteq \mathcal{M}$:

$$G_\#(g_\#\mathbb{P}^*)(B) = g_\#\mathbb{P}^*(G^{-1}(B)) = \mathbb{P}^*(g^{-1}(G^{-1}(B))) = \mathbb{P}^*\left(g^{-1}\left(G^{-1}\left((B \setminus A) \cup (A \cap B)\right)\right)\right) \tag{22}$$

$$= \mathbb{P}^*\left(g^{-1}\left(G^{-1}\left(B \setminus A\right)\right) \cup g^{-1}\left(G^{-1}\left(A \cap B\right)\right)\right) = \mathbb{P}^*(g^{-1}\left(G^{-1}\left(B \setminus A\right)\right)) \tag{23}$$

$$= \mathbb{P}^*(B \setminus A) = \mathbb{P}^*(B), \tag{24}$$

where we used that $g^{-1}(G^{-1}(A \cap B)) \subseteq A$, and thus $G_\#(g_\#\mathbb{P}^*) = \mathbb{P}^*$. Note that this derivation requires thinking of $\mathbb{P}^*$ as a measure on $\mathbb{R}^D$ to ensure that $A$ and $g^{-1}(G^{-1}(A \cap B))$ can be assigned 0 probability.

We now prove 2b. We begin by showing that $G(g(x)) = x$ for all $x \in \mathcal{M}$. Consider $\mathbb{R}^D \times \mathcal{M}$ endowed with the product topology. Clearly $\mathbb{R}^D \times \mathcal{M}$ is Hausdorff since both $\mathbb{R}^D$ and $\mathcal{M}$ are Hausdorff ($\mathcal{M}$ is Hausdorff by the definition of a manifold). Let $E = \{(x, x) \in \mathbb{R}^D \times \mathcal{M} : x \in \mathcal{M}\}$, which is then closed in $\mathbb{R}^D \times \mathcal{M}$ (since diagonals of Hausdorff spaces are closed). Consider the function $H : \mathcal{M} \to \mathbb{R}^D \times \mathcal{M}$ given by $H(x) = (G(g(x)), x)$, which is clearly continuous. It follows that $H^{-1}(E) = \{x \in \mathcal{M} : G(g(x)) = x\}$ is closed in $\mathcal{M}$, and thus $\mathcal{M} \setminus H^{-1}(E) = \{x \in \mathcal{M} : G(g(x)) \neq x\}$ is open in $\mathcal{M}$, and by assumption $\mathbb{P}^*(\mathcal{M} \setminus H^{-1}(E)) = 0$. It follows by Lemma 1 in App. B.1 that $\mathcal{M} \setminus H^{-1}(E) = \emptyset$, and thus $G(g(x)) = x$ for all $x \in \mathcal{M}$.

We now prove that $\tilde{g}$ is a diffeomorphism. Clearly $\tilde{g}$ is surjective, and since it admits a left inverse (namely $G$), it is also injective. Then $\tilde{g}$ is bijective, and since it is clearly $C^1$ due to $g$ being $C^1$ and $\mathcal{M}$ being an embedded submanifold of $\mathbb{R}^D$, it only remains to show that its inverse is also $C^1$. Since $G(g(x)) = x$ for every $x \in \mathcal{M}$, it follows that $G(g(\mathcal{M})) = \mathcal{M}$, and thus $\tilde{G}$ is well-defined (i.e. the image of its domain is indeed contained in its codomain). Clearly $\tilde{G}$ is a left inverse to $\tilde{g}$, and by bijectivity of $\tilde{g}$, it follows $\tilde{G}$ is its inverse. Finally, $\tilde{G}$ is also $C^1$ since $G$ is $C^1$, so that $\tilde{g}$ is indeed a diffeomorphism.

Now, we prove 2a. Let $K \subset \mathbb{R}^d$ be such that $\mu_d(K) = 0$. We need to show that $g_\#\mathbb{P}^*(K) = 0$ in order to complete the proof. We have that:

$$g_\#\mathbb{P}^*(K) = \mathbb{P}^*\left(g^{-1}(K)\right) = \mathbb{P}^*\left(g^{-1}(K) \cap \mathcal{M}\right). \tag{25}$$

Let $\mathfrak{g}$ be a Riemannian metric on $\mathcal{M}$. Since $\mathbb{P}^* \ll \mu_{\mathcal{M}}^{(\mathfrak{g})}$ by assumption, it is enough to show that $\mu_{\mathcal{M}}^{(\mathfrak{g})}(g^{-1}(K) \cap \mathcal{M}) = 0$. Let $\{U_\alpha\}_\alpha$ be an open (in $\mathcal{M}$) cover of $g^{-1}(K) \cap \mathcal{M}$. Since $\mathcal{M}$ is second countable by definition, by Lindelöf's lemma there exists a countable subcover $\{V_\beta\}_{\beta \in \mathbb{N}}$. Since $g|_{\mathcal{M}}$ is a diffeomorphism onto its image,

$(V_\beta, g|_{V_\beta})$ is a chart for every $\beta \in \mathbb{N}$. We have:

$$\mu_{\mathcal{M}}^{(\mathfrak{g})}\left(g^{-1}(K) \cap \mathcal{M}\right) = \mu_{\mathcal{M}}^{(\mathfrak{g})}\left(g^{-1}(K) \cap \mathcal{M} \cap \bigcup_{\beta \in \mathbb{N}} V_\beta\right) = \mu_{\mathcal{M}}^{(\mathfrak{g})}\left(\bigcup_{\beta \in \mathbb{N}} g^{-1}(K) \cap \mathcal{M} \cap V_\beta\right) \tag{26}$$

$$\leq \sum_{\beta \in \mathbb{N}} \mu_{\mathcal{M}}^{(\mathfrak{g})}(g^{-1}(K) \cap \mathcal{M} \cap V_\beta) \tag{27}$$

$$= \sum_{\beta \in \mathbb{N}} \int_{g|_{V_\beta}(g^{-1}(K) \cap \mathcal{M} \cap V_\beta)} \sqrt{\det\left(\mathfrak{g}\left(\frac{\partial}{\partial g|_{V_\beta}^i}, \frac{\partial}{\partial g|_{V_\beta}^j}\right)\right)} \, d\mu_d = 0, \tag{28}$$

where the final equality follows from $g|_{V_\beta}\left(g^{-1}(K) \cap \mathcal{M} \cap V_\beta\right) \subseteq K$ for every $\beta \in \mathbb{N}$ and $\mu_d(K) = 0$.

$\square$

## C   Experimental Details

### C.1   Model Losses

Throughout this section we use $\mathcal{L}$ to denote the loss of different models. We use notation that assumes all of these are first-step models, i.e. datapoints are denoted as $x_n$, but we highlight that when trained as second-step models, the datapoints actually correspond to $z_n$. Similarly, whenever a loss includes $D$, this should be understood as $d$ for second-step models. The description of these losses here is meant only for reference, and we recommend that any reader unfamiliar with these see the relevant citations in the main manuscript. Unlike our main manuscript, measure-theoretic notation is not needed to describe these models, and we thus drop it.

**AEs**   As mentioned in the main manuscript, we train autoencoders with a squared reconstruction error:

$$\mathcal{L}(g, G) = \frac{1}{N} \sum_{n=1}^{N} \|G(g(x_n)) - x_n\|_2^2. \tag{29}$$

**ARMs**   The loss of autoregressive models is given by the negative log-likelihood:

$$\mathcal{L}(p) = -\frac{1}{N} \sum_{n=1}^{N} \left(\log p(x_{n,1}) + \sum_{m=2}^{D} \log p(x_{n,m}|x_{n,1}, \ldots, x_{n,m-1})\right), \tag{30}$$

where $x_{n,m}$ denotes the $m^{th}$ coordinate of $x_n$.

**AVB**   Adversarial variational Bayes is highly related to VAEs (see description below), except the approximate posterior is defined implicitly, so that a sample $U$ from $q(\cdot|x)$ can be obtained as $U = \tilde{g}(x, \epsilon)$, where $\epsilon \sim p_\epsilon(\cdot)$, which is often taken as a standard Gaussian of dimension $\tilde{d}$, and $\tilde{g} : \mathbb{R}^D \times \mathbb{R}^{\tilde{d}} \to \mathbb{R}^d$. Since $q(\cdot|x)$ cannot be evaluated, the ELBO used to train VAEs becomes intractable, and thus a discriminator $T : \mathbb{R}^D \times \mathbb{R}^d \to \mathbb{R}$ is introduced, and for fixed $q(\cdot|x)$, trained to minimize:

$$\mathcal{L}(T) = -\sum_{n=1}^{N} \left(\mathbb{E}_{U \sim q(\cdot|x_n)}[\log s(T(x_n, U))] + \mathbb{E}_{U \sim p_U(\cdot)}[\log(1 - s(T(x_n, U)))]\right), \tag{31}$$

where $s(\cdot)$ denotes the sigmoid function. Denoting the optimal $T$ as $T^*$, the rest of the model components are trained through:

$$\mathcal{L}(G, \sigma_X) = \frac{1}{N} \sum_{n=1}^{N} \mathbb{E}_{U \sim q(\cdot|x_n)}[T^*(x_n, U) - \log p(x_n|U)], \tag{32}$$

where $p(x_n|U)$ depends on $G$ and $\sigma_X$ in an identical way as in VAEs (see below). Analogously to VAEs, this training procedure maximizes a lower bound on the log-likelihood, which is tight when the approximate posterior matches the true one. Finally, $z_n$ can either be taken as:

$$z_n = \mathbb{E}_{U \sim q(\cdot|x_n)}[U] = \mathbb{E}_{\epsilon \sim p_\epsilon(\cdot)}[\tilde{g}(x_n, \epsilon)], \qquad \text{or} \qquad z_n = \tilde{g}(x_n, 0). \tag{33}$$

We use the former, and approximate the expectation through a Monte Carlo average. Note that both options define $g$ through $\tilde{g}$ in such a way that $z_n = g(x_n)$. Finally, in line with Goodfellow et al. (2014), we found that using the "log trick" to avoid saturation in the adversarial loss further improved performance.

**BiGAN** Bidirectional GANs model the data as $X = G(Z)$, where $Z \sim \tilde{p}_Z$, and $\tilde{p}_Z$ is taken as a $d$-dimensional standard Gaussian. Note that this $\tilde{p}_Z$ is different from $p_Z$ in the main manuscript (which corresponds to the density of the second-step model), hence why we use different notation. BiGANs also aim to recover the $z_n$ corresponding to each $x_n$, and so also use an encoder $g$, in addition to a discriminator $T : \mathbb{R}^D \times \mathbb{R}^d \to \mathbb{R}$. All the components are trained through the following objective:

$$\mathcal{L}(g, G; T) = \mathbb{E}_{Z \sim \tilde{p}_Z(\cdot)}[T(G(Z), Z)] - \frac{1}{N} \sum_{n=1}^{N} T(x_n, g(x_n)), \tag{34}$$

which is minimized with respect to $g$ and $G$, but maximized with respect to $T$. We highlight that this objective is slightly different than the originally proposed BiGAN objective, as we use the Wasserstein loss (Arjovsky et al., 2017) instead of the original Jensen-Shannon. In practice we penalize the gradient of $T$ as is often done for the Wasserstein objective (Gulrajani et al., 2017). We also found that adding a square reconstruction error as an additional regularization term helped improve performance.

**EBM** Energy-based models use an energy function $E : \mathbb{R}^D \to \mathbb{R}$, which implicitly defines a density on $\mathbb{R}^D$ as:

$$p(x) = \frac{e^{-E(x)}}{\int_{\mathbb{R}^D} e^{-E(x')} \mathrm{d}\mu_D(x')}. \tag{35}$$

These models attempt to minimize the negative log-likelihood:

$$\mathcal{L}(E) = -\frac{1}{N} \sum_{n=1}^{N} \log p(x_n), \tag{36}$$

which is seemingly intractable due to the integral in (35). However, when parameterizing $E$ with $\theta$ as a neural network $E_\theta$, gradients of this loss can be obtained thanks to the following identity:

$$-\nabla_\theta \log p_\theta(x_n) = \nabla_\theta E_\theta(x_n) - \mathbb{E}_{X \sim p_\theta}[\nabla_\theta E_\theta(X)], \tag{37}$$

where we have also made the dependence of $p$ on $\theta$ explicit. While it might seem that the expectation in (37) is just as intractable as the integral in (35), in practice approximate samples from $p_\theta$ are obtained through Langevin dynamics and are used to approximate this expectation.

**NFs** Normalizing flows use a bijective neural network $h : \mathbb{R}^D \to \mathbb{R}^D$, along with a base density $p_U$ on $\mathbb{R}^D$, often taken as a standard Gaussian, and model the data as $X = h(U)$, where $U \sim p_U$. Thanks to the change-of-variable formula, the density of the model can be evaluated:

$$p(x) = p_U(h^{-1}(x))|\det J_{h^{-1}}(x)|, \tag{38}$$

and flows can thus be trained via maximum-likelihood:

$$\mathcal{L}(h) = -\frac{1}{N} \sum_{n=1}^{N} \left( \log p_U(h^{-1}(x_n)) + \log |\det J_{h^{-1}}(x_n)| \right). \tag{39}$$

In practice $h$ is constructed in such a way that not only ensures it is bijective, but also ensures that $\log |\det J_{h^{-1}}(x_n)|$ can be efficiently computed.

**VAEs** Variational autoencoders define the generative process for the data as $U \sim p_U$, $X|U \sim p(\cdot|U)$. Typically, $p_U$ is a standard $d$-dimensional Gaussian (although a learnable prior can also be used), and in our case, $p(\cdot|u)$ is given by a Gaussian:

$$p(x|u) = \mathcal{N}(x; G(u), \sigma_X^2(u)I_D), \tag{40}$$

where $\sigma_X : \mathbb{R}^d \to \mathbb{R}$ is a neural network. Maximum-likelihood is intractable since the latent variables $u_n$ corresponding to $x_n$ are unobserved, so instead an approximate posterior $q(u|x)$ is introduced. We take $q$ to be Gaussian:

$$q(u|x) = \mathcal{N}(u; g(x), \mathrm{diag}(\sigma_U^2(x))), \tag{41}$$

where $\sigma_U^2 : \mathbb{R}^D \to \mathbb{R}_{>0}^d$ is a neural network, and $\mathrm{diag}(\sigma_U^2(x))$ denotes a diagonal matrix whose nonzero elements are given by $\sigma_U^2(x)$. An objective called the negative ELBO is then minimized:

$$\mathcal{L}(g, G, \sigma_U, \sigma_X) = \frac{1}{N} \sum_{n=1}^{N} \left( \mathbb{KL}(q(\cdot|x_n)|p_U(\cdot)) - \mathbb{E}_{U \sim q(\cdot|x_n)} [\log p(x_n|U)] \right). \tag{42}$$

The ELBO can be shown to be a lower bound to the log-likelihood, which becomes tight as the approximate posterior matches the true posterior. Note that $z_n$ corresponds to the mean of the unobserved latent $u_n$:

$$z_n = \mathbb{E}_{U \sim q(\cdot|x_n)}[U] = g(x_n). \tag{43}$$

We highlight once again that the notation we use here corresponds to VAEs when used as first-step models. When used as second-step models, as previously mentioned, the observed datapoint $x_n$ becomes $z_n$, but in this case the encoder and decoder functions do not correspond to $g$ and $G$ anymore. Similarly, for second-step models, the unobserved variables $u_n$ become "irrelevant" in terms of the main contents of our paper, and are not related to $z_n$ in the same way as in first-step models. For second-step models, we keep the latent dimension as $d$ still.

**WAEs** Wasserstein autoencoders, similarly to BiGANs, model the data as $X = G(Z)$, where $Z \sim \tilde{p}_Z$, which is taken as a $d$-dimensional standard Gaussian, and use a discriminator $T : \mathbb{R}^d \to \mathbb{R}$. The WAE objective is given by:

$$\mathcal{L}(g, G; T) = \frac{1}{N} \sum_{n=1}^{N} \left( \|G(g(x_n)) - x_n\|_2^2 + \lambda \log(1 - s(T(g(x_n)))) \right) + \lambda \mathbb{E}_{Z \sim \tilde{p}_Z(\cdot)}[\log s(T(Z))], \tag{44}$$

where $s(\cdot)$ denotes the sigmoid function, $\lambda > 0$ is a hyperparameter, and the objective is minimized with respect to $g$ and $G$, and maximized with respect to $T$. Just as in AVB, we found that using the "log trick" of Goodfellow et al. (2014) in the adversarial loss further improved performance.

## C.2 VAE from Fig. 2

We generated $N = 1000$ samples from $\mathbb{P}^* = 0.3\delta_{-1} + 0.7\delta_1$, resulting in a dataset containing 1 a total of 693 times. The Gaussian VAE had $d = 1$, $D = 1$, and both the encoder and decoder have a single hidden layer with 25 units and ReLU activations. We use the Adam optimizer (Kingma & Ba, 2015) with learning rate 0.001 and train for 200 epochs. We use gradient norm clipping with a value of 10.

## C.3 Simulated Data

For the ground truth, we use a von Mises distribution with parameter $\kappa = 1$, and transform to Cartesian coordinates to obtain a distribution on the unit circle in $\mathbb{R}^D = \mathbb{R}^2$. We generate $N = 1000$ samples from this distribution. For the EBM model, we use an energy function with two hidden layers of 25 units each and Swish activations (Ramachandran et al., 2017). We use the Adam optimizer with learning rate 0.01, and gradient norm clipping with value of 1. We train for 100 epochs. We follow Du & Mordatch (2019) for the training of the EBM, and use 0.1 for the objective regularization value, iterate Langevin dynamics for 60

iterations at every training step, use a step size of 10 within Langevin dynamics, sample new images with probability 0.05 in the buffer, use Gaussian noise with standard deviation 0.005 in Langevin dynamics, and truncate gradients to $(-0.03, 0.03)$ in Langevin dynamics. For the AE+EBM model, we use an AE with $d = 1$ and two hidden layers of 20 units each with ELU activations (Clevert et al., 2016). We use the Adam optimizer with learning rate 0.001 and train for 200 epochs. We use gradient norm clipping with a value of 10. For the EBM of this model, we use an energy function with two hidden layers of 15 units each, and all the other parameters are identical to the single step EBM. We observed some variability with respect to the seed for both the EBM and the AE+EBM models; the manuscript shows the best performing versions.

For the additional results of Sec. D.1, the ground truth is given by a Gaussian whose first coordinate has mean 0 and variance 2, while the second coordinate has mean 1 and variance 1, and they have a covariance of 0.5. The VAEs are identical to those from Fig. 2, except their input and output dimensions change accordingly.

### C.4 Comparisons Against Maximum-Likelihood and OOD Detection with Implicit Models

For all experiments, we use the Adam optimizer, typically with learning rate 0.001. For all experiments we also clip gradient entries larger than 10 during optimization. We also set $d = 20$ in all experiments.

#### C.4.1 Single and First-Step Models

For all single and first-step models, unless specified otherwise, we pre-process the data by scaling it, i.e. dividing by the maximum absolute value entry. All convolutions have a kernel size of 3 and stride 1. For all versions with added Gaussian noise, we tried standard deviation values $\sigma \in \{1, 0.1, 0.01, 0.001, 0.0001\}$ and kept the best performing one ($\sigma = 0.1$, as measured by FID) unless otherwise specified.

**AEs** For MNIST and FMNIST, we use MLPs for the encoder and decoder, with ReLU activations. The encoder and decoder have each a single hidden layer with 256 units. For SVHN and CIFAR-10, we use convolutional networks. The encoder and decoder have 4 convolutional layers with $(32, 32, 16, 16)$ and $(16, 16, 32, 32)$ channels, respectively, followed by a flattening operation and a fully-connected layer. The convolutional networks also use ReLU activations, and have kernel size 3 and stride 1. We perform early stopping on reconstruction error with a patience of 10 epochs, for a maximum of 100 epochs.

**ARMs** We use an updated version of RNADE (Uria et al., 2013), where we use an LSTM (Hochreiter & Schmidhuber, 1997) to improve performance. More specifically, every pixel is processed sequentially through the LSTM, and a given pixel is modelled with a mixture of Gaussians whose parameters are given by transforming the hidden state obtained from all the previous pixels through a linear layer. The dimension of a pixel is given by the number of channels, so that MNIST and FMNIST use mixtures of 1-dimensional Gaussians, whereas SVHN and CIFAR-10 use mixtures of 3-dimensional Gaussians. We also tried a continuous version of the PixelCNN model (van den Oord et al., 2016b), where we replaced the discrete distribution over pixels with a mixture of Gaussians, but found this model highly unstable – which is once again consistent with manifold overfitting – and thus opted for the LSTM-based model. We used 10 components for the Gaussian mixtures, and used an LSTM with 2 layers and hidden states of size 256. We train for a maximum of 100 epochs, and use early stopping on log-likelihood with a patience of 10. We also use cosine annealing on the learning rate. For the version with added Gaussian noise, we used $\sigma = 1.0$. We observed some instabilities in training these single step models, particularly when not adding noise, where the final model was much worse than average (over 100 difference in FID score). We treated these runs as failed runs and excluded them from the averages and standard errors reported in our paper.

**AVB** We use the exact same configuration for the encoder and decoder as in AEs, and use an MLP with 2 hidden layers of size 256 each for the discriminator, which also uses ReLU activations. We train the MLPs for a maximum of 50 epochs, and CNNs for 100 epochs, using cosine annealing on the learning rates. For the large version, AVB$^+$, we use two hidden layers of 256 units for the encoder and decoder MLPs, and increase the encoder and decoder number of hidden channels to $(64, 64, 32, 32)$ and $(32, 32, 64, 64)$, respectively, for convolutional networks. In all cases, the encoder takes in 256-dimensional Gaussian noise with covariance

$9 \cdot I_D$. We also tried having the decoder output per-pixel variances, but found this parameterization to be numerically unstable, which is again consistent with manifold overfitting.

**BiGAN**  As mentioned in Sec. C.1, we used a Wasserstein-GAN (W-GAN) objective (Arjovsky et al., 2017) with gradient penalties (Gulrajani et al., 2017) where both the data and latents are interpolated between the real and generated samples. The gradient penalty weight was 10. The generator-encoder loss includes the W-GAN loss, and the reconstruction loss (joint latent regressor from Donahue et al. (2017)), equally weighted. For both small and large versions, we use the exact same configuration for the encoder, decoder, and discriminator as for AVB. We used learning rates of 0.0001 with cosine annealing over 200 epochs. The discriminator was trained for two steps for every step taken with the encoder/decoder.

**EBMs**  For MNIST and FMNIST, our energy functions use MLPs with two hidden layers with 256 and 128 units, respectively. For SVHN and CIFAR-10, the energy functions have 4 convolutional layers with hidden channels $(64, 64, 32, 32)$. We use the Swish activation function and spectral normalization in all cases. We set the energy function's output regularization coefficient to 1 and the learning rate to 0.0003. Otherwise, we use the same hyperparameters as on the simulated data. At the beginning of training, we scale all the data to between 0 and 1. We train for 100 epochs without early stopping, which tended to halt training too early.

**NFs**  We use a rational quadratic spline flow (Durkan et al., 2019) with 128 hidden units, 4 layers, and 3 blocks per layer. We train using early stopping on validation loss with a patience of 30 epochs, up to a maximum of 100 epochs. We use a learning rate of 0.0005, and use a whitening transform at the start of training to make the data zero-mean and marginally unit-variance, whenever possible (some pixels, particularly in MNIST, were only one value throughout the entire training set); note that this affine transformation does not affect the manifold structure of the data.

**VAEs**  The settings for VAEs were largely identical to those of AVB, except we did not do early stopping and always trained for 100 epochs, in addition to not needing a discriminator. For large models a single hidden layer of 512 units was used for each of the encoder and decoder MLPs. We also tried the same decoder per-pixel variance parameterization that we attempted with AVB and obtained similar numerical instabilities, once again in line with manifold overfitting.

**WAEs**  We use the adversarial variant rather than the maximum mean discrepancy (Gretton et al., 2012) one. We weight the adversarial loss with a coefficient of 10. The settings for WAEs were identical to those of AVB, except (*i*) we used a patience of 30 epochs, trained for a maximum of 300 epochs, (*ii*) we used no learning rate scheduling, with a discriminator learning rate of $2.5 \times 10^{-4}$ and an encoder-decoder learning rate of $5 \times 10^{-4}$, and (*iii*) we used only convolutional encoders and decoders, with $(64, 64, 32, 32)$ and $(32, 32, 64, 64)$ hidden channels, respectively. For large models the number of hidden channels was increased to $(96, 96, 48, 48)$ and $(48, 48, 96, 96)$ for the encoder and decoder, respectively.

### C.4.2  Second-Step Models

All second-step models, unless otherwise specified, pre-process the encoded data by standardizing it (i.e. subtracting the mean and dividing by the standard deviation).

**ARMs**  We used the same configuration for second-step ARMs as for the first-step version, except the LSTM has a single hidden layer with hidden states of size 128.

**AVB**  We used the same configuration for second-step AVB as we did for the first-step MLP version of AVB, except that we do not do early stopping and train for 100 epochs. The latent dimension is set to $d$ (i.e. 20).

**EBMs**  We used the same configuration that we used for single-step EBMs, except we use a learning rate of 0.001, we regularize the energy function's output by 0.1, do not use spectral normalization, take the energy function to have two hidden layers with $(64, 32)$ units, and scale the data between $-1$ and 1.

**NFs** We used the same settings for second-step NFs as we did for first-step NFs, except ($i$) we use 64 hidden units, ($ii$) we do not do early stopping, training for a maximum of 100 epochs, and ($iii$) we use a learning rate of 0.001.

**VAEs** We used the same settings for second-step VAEs as we did for first-step VAEs. The latent dimension is also set to $d$ (i.e. 20).

### C.4.3 Parameter Counts

Table 3 includes parameter counts for all the models we consider in Table 1. Two-step models have either fewer parameters than the large one-step model versions, or a roughly comparable amount, except for some exceptions which we now discuss. First, when using normalizing flows as second-step models, we used significantly more complex models than with other two-step models. We did this for added variability in the number of parameters, not because using fewer parameters makes two-step models not outperform their single-step counterparts. Two-step models with an NF as the second-step model outperform other two-step models (see Table 1), but there is a much more drastic improvement from single to two-step models. This difference in improvements further highlights that the main cause for empirical gains is the two-step nature of our models, rather than increased number of parameters. Second, the AE+EBM models use more parameters than their single-step baselines. This was by design, as the architecture of the energy functions mimics that of the encoders of other larger models, except it outputs scalars and thus has fewer parameters, and hence we believe this remains a fair comparison. We also note that AE+EBM models have most of their parameters assigned to the AE, and the second-step EBM contributes only 4k additional parameters. AE+EBM models also train and sample much faster then their single-step EBM$^+$ counterparts. Finally, we finish with the observation that measuring capacity is difficult, and parameter counts simply provide a proxy.

Table 3: Approximate parameter counts in thousands.

| MODEL | MNIST/FMNIST | SVHN/CIFAR-10 |
|---|---|---|
| AVB | 750 | 980 |
| AVB$^+$ / AVB$^+_\sigma$ | 882 | 1725 |
| AVB+ARM | 1021 | 1251 |
| AVB+AVB | 913 | 1143 |
| AVB+EBM | 754 | 984 |
| AVB+NF | 5756 | 5986 |
| AVB+VAE | 771 | 1001 |
| VAE | 412 | 703 |
| VAE$^+$ / VAE$^+_\sigma$ | 824 | 1448 |
| VAE+ARM | 683 | 974 |
| VAE+AVB | 575 | 866 |
| VAE+EBM | 416 | 707 |
| VAE+NF | 5418 | 5709 |
| ARM$^+$ / ARM$^+_\sigma$ | 797 | 799 |
| AE+ARM | 683 | 974 |
| EBM$^+$ / EBM$^+_\sigma$ | 236 | 99 |
| AE+EBM | 416 | 707 |

## D   Additional Experimental Results

### D.1   Simulated Data

As mentioned in the main manuscript, we carry out additional experiments where we have access to the ground truth $\mathbb{P}^*$ in order to further verify that our improvements from two-step models indeed come from mismatched dimensions. Fig. 6 shows the results of running VAE and VAE+VAE models when trying to approximate a nonstandard 2-dimensional Gaussian distribution. First, we can see that when setting the

intrinsic dimension of the models to $d = 2$, the VAE and VAE+VAE models have very similar performance, with the VAE being slightly better. Indeed, there is no reason to suppose the second-step VAE will have an easier time learning encoded data than the first-step VAE learning the actual data. This result visually confirms that two-step models do not outperform single-step models trained with maximum likelihood when the dimension of maximum-likelihood is correctly specified. Second, we can see that both the VAE and the VAE+VAE models with intrinsic dimension $d = 1$ underperform their counterparts with $d = 2$. However, while the VAE model still manages to approximate its target distribution, the VAE+VAE completely fails. This result visually confirms that two-step models significantly underperform single-step models trained with maximum-likelihood if the data has no low-dimensional structure and the two-step model tries to enforce such structure anyway. Together, these results highlight that the reason two-step models outperform maximum-likelihood so strongly in the main manuscript is indeed the dimensionality mismatch caused by not heeding to the manifold hypothesis.

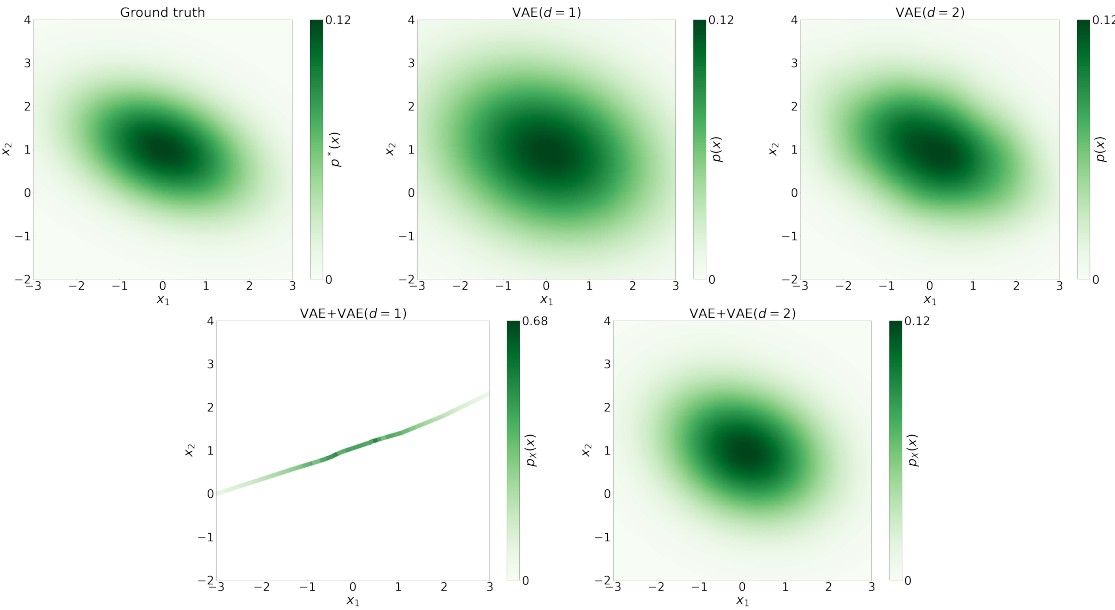

Figure 6: Results on simulated data: Gaussian ground truth **(top left)**, VAE with $d = 1$ **(top middle)**, VAE with $d = 2$ **(top right)**, VAE+VAE with $d = 1$ **(bottom left)**, and VAE+VAE with $d = 2$ **(bottom right)**.

## D.2 Samples

We show samples obtained by the VAE, VAE$^+$, VAE$_\sigma^+$, and VAE+ARM models in Fig. 7. In addition to the FID improvements shown in the main manuscript, we can see a very noticeable qualitative improvement obtained by the two-step models. Note that the VAE in the VAE+ARM model is the same as the single-step VAE model. Similarly, we show samples from AVB$_\sigma^+$, AVB+NF, AVB+EBM, and AVB+VAE in Fig. 8 where two-step models greatly improve visual quality. We also show samples from the ARM$^+$, ARM$_\sigma^+$, and AE+ARM from the main manuscript in Fig. 9; and for the EBM$^+$, EBM$_\sigma^+$, and AE+EBM models in Fig. 10. We can see that FID score is indeed not always indicative of image quality, and that our AE+ARM and AE+EBM models significantly outperform their single-step counterparts (except AE+EBM on MNIST). Finally, the BiGAN and WAE samples shown in Fig. 11 and Fig. 12 respectively are not consistently better for two-step models, but neither BiGANs nor WAEs are trained via maximum likelihood so manifold overfitting is not necessarily implied by Theorem 1. Other two-step combinations not shown gave similar results.

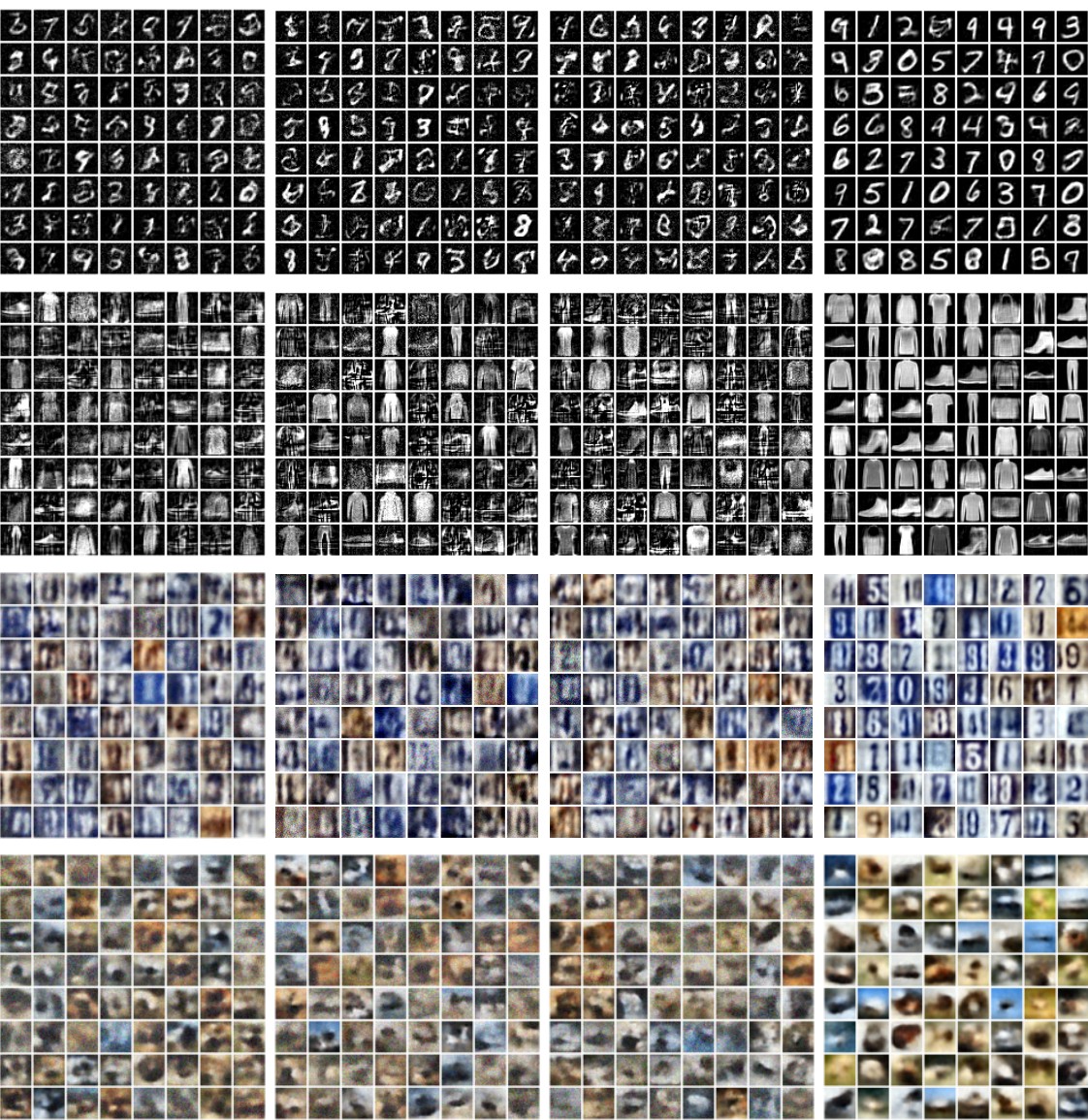

Figure 7: Uncurated samples from models trained on MNIST **(first row)**, FMNIST **(second row)**, SVHN **(third row)**, and CIFAR-10 **(fourth row)**. Models are VAE **(first column)**, VAE$^+$ **(second column)**, VAE$_\sigma^+$ **(third column)**, and VAE+ARM **(fourth column)**.

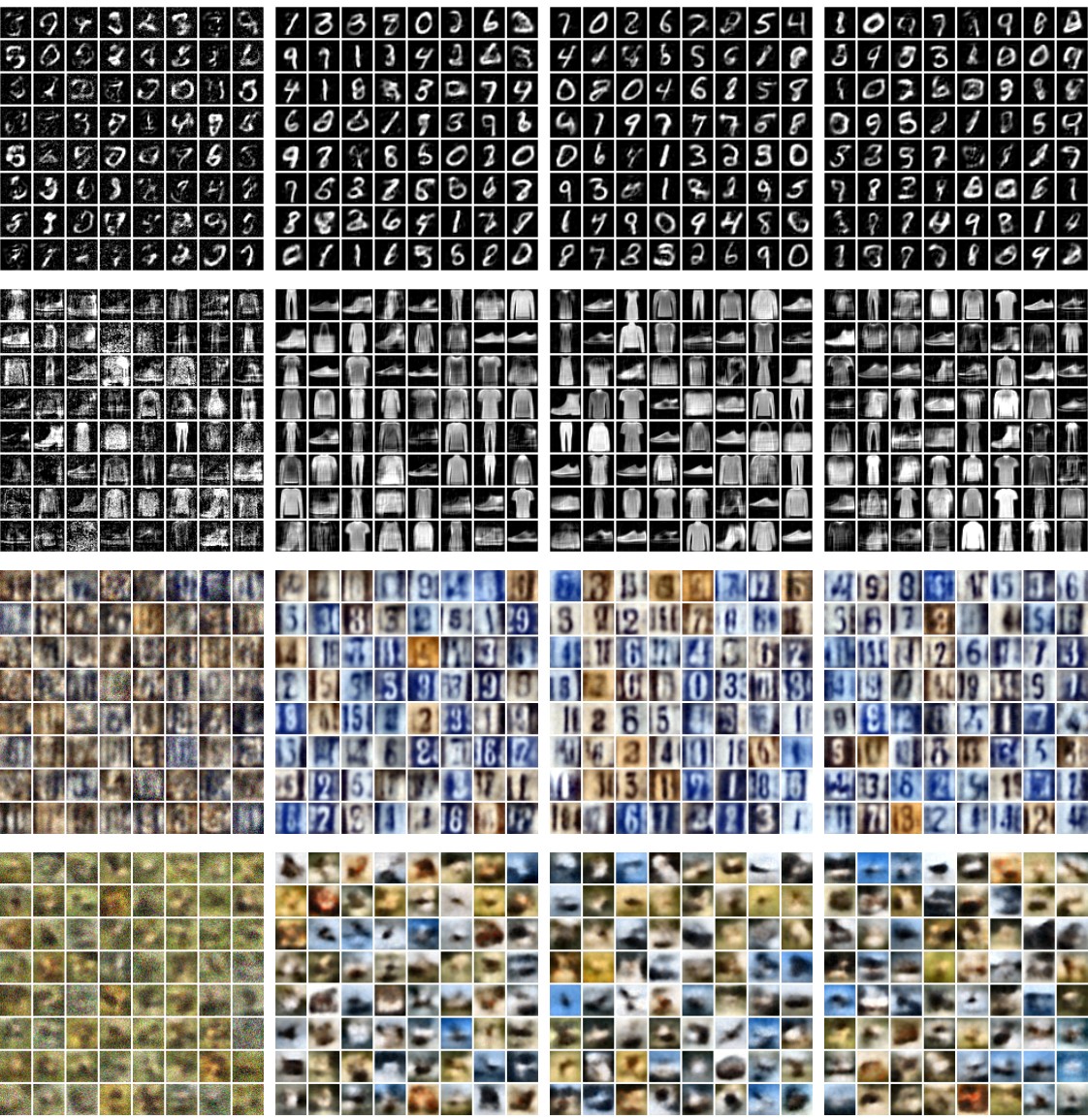

Figure 8: Uncurated samples from models trained on MNIST **(first row)**, FMNIST **(second row)**, SVHN **(third row)**, and CIFAR-10 **(fourth row)**. Models are $\text{AVB}_\sigma^+$ **(first column)**, AVB+EBM **(second column)**, AVB+NF **(third column)**, and AVB+VAE **(fourth column)**.

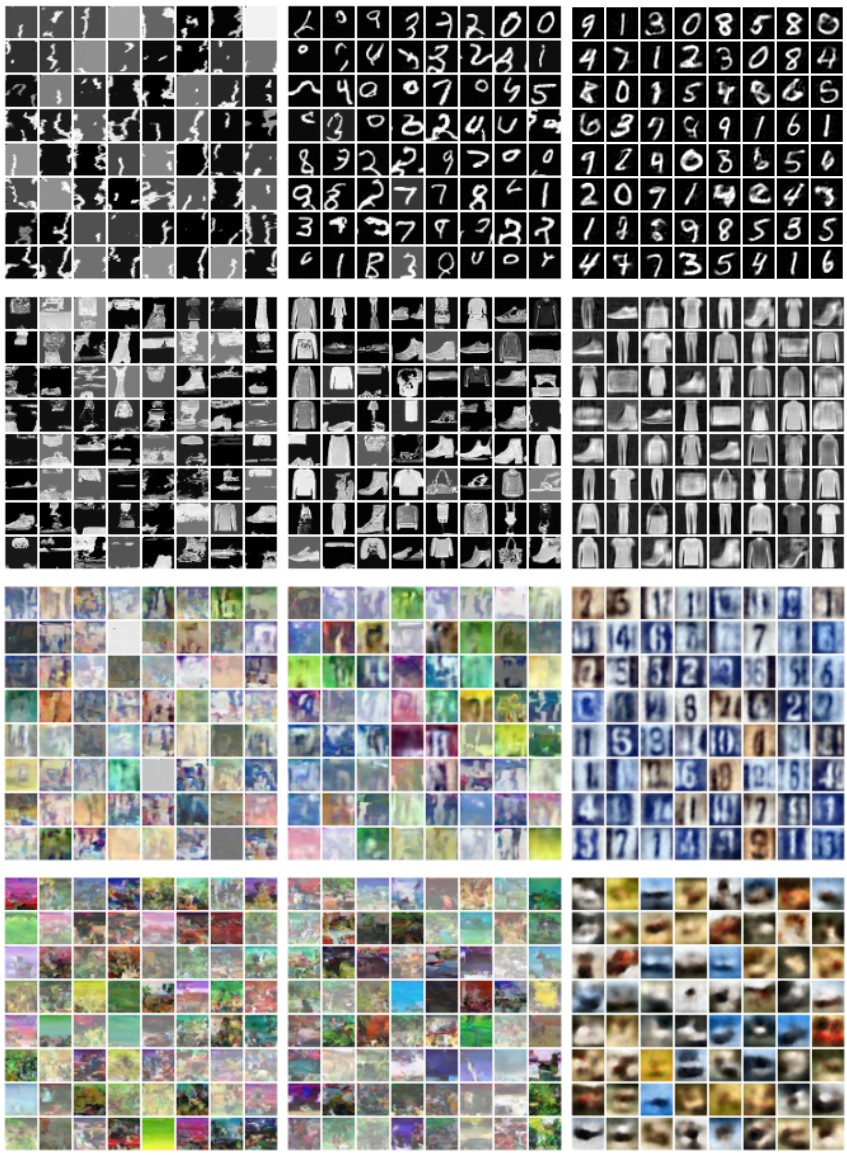

Figure 9: Uncurated samples from models trained on MNIST **(first row)**, FMNIST **(second row)**, SVHN **(third row)**, and CIFAR-10 **(fourth row)**. Models are ARM$^+$ **(first column)**, ARM$_\sigma^+$ **(second column)**, and AE+ARM **(third column)**.

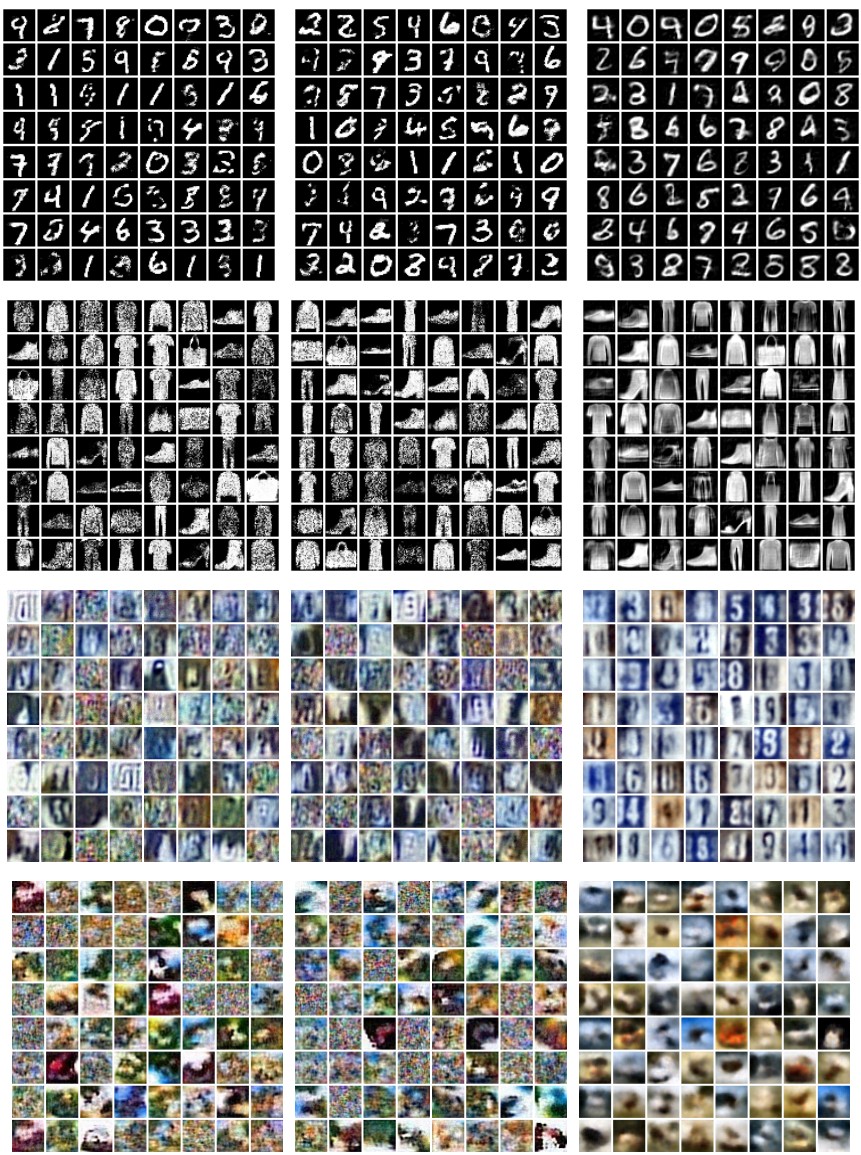

Figure 10: Uncurated samples with Langevin dynamics run for 60 steps initialized from training buffer on MNIST **(first row)**, FMNIST **(second row)**, SVHN **(third row)**, and CIFAR-10 **(fourth row)**. Models are EBM$^+$ **(first column)**, EBM$_\sigma^+$ **(second column)**, and AE + EBM **(third column)**.

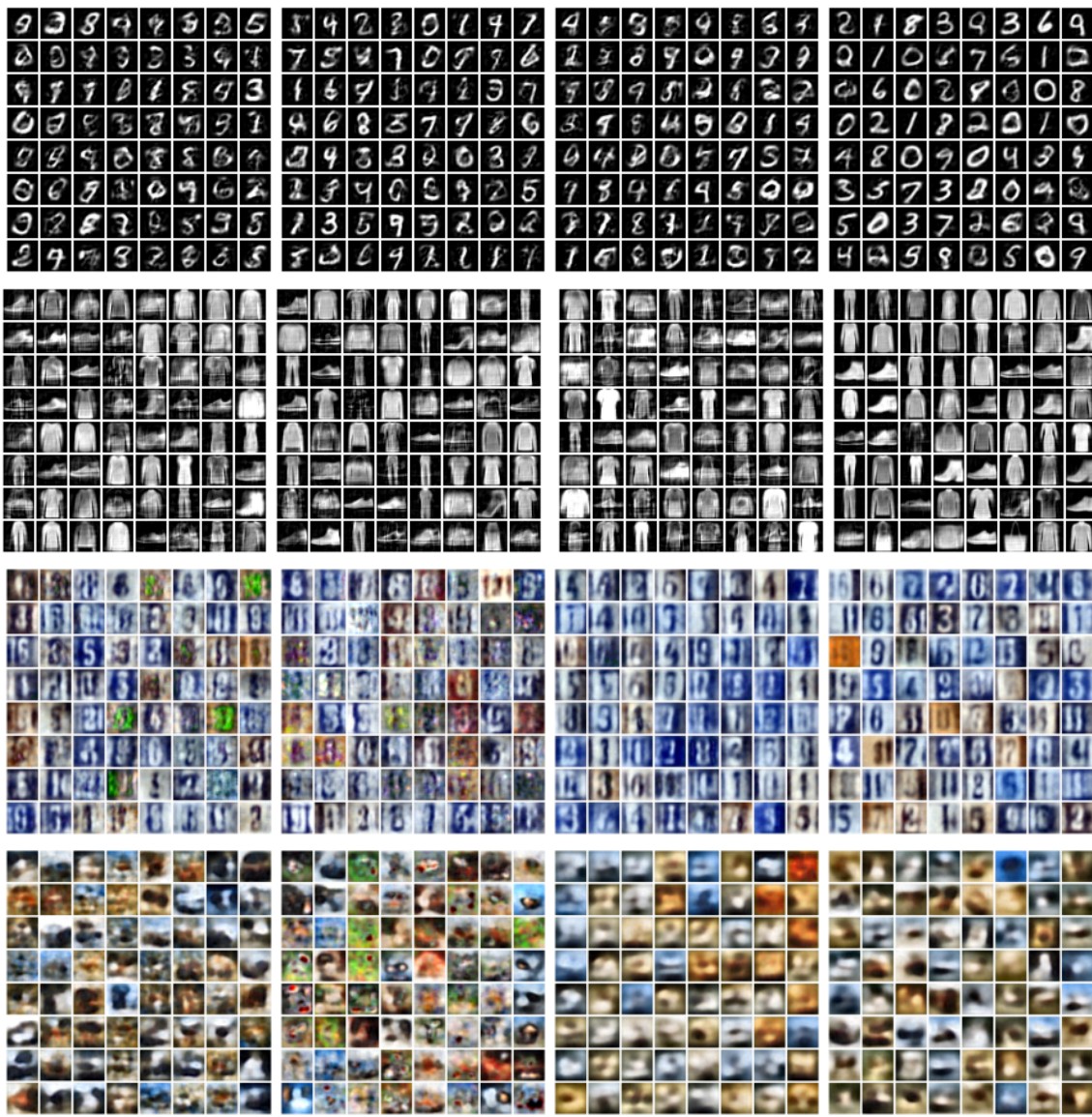

Figure 11: Uncurated samples from models trained on MNIST **(first row)**, FMNIST **(second row)**, SVHN **(third row)**, and CIFAR-10 **(fourth row)**. Models are BiGAN **(first column)**, BiGAN$^+$ **(second column)**, BiGAN+AVB **(third column)**, and BiGAN+NF **(fourth column)**. BiGANs are not trained via maximum-likelihood, so Theorem 1 does not imply that manifold overfitting should occur.

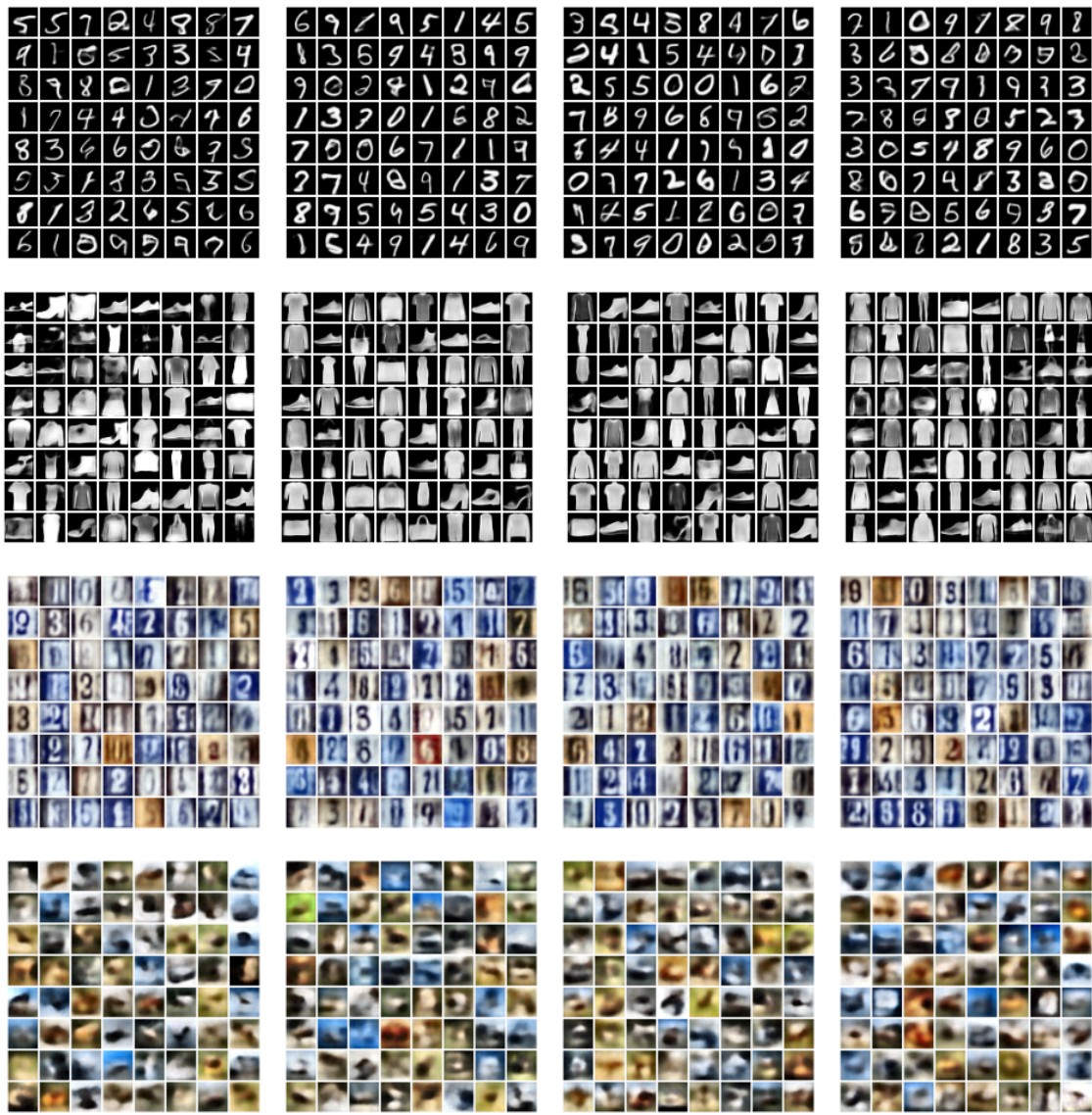

Figure 12: Uncurated samples from models trained on MNIST **(first row)**, FMNIST **(second row)**, SVHN **(third row)**, and CIFAR-10 **(fourth row)**. Models are WAE$^+$ **(first column)**, WAE+ARM **(second column)**, WAE+NF **(third column)**, and WAE+VAE **(fourth column)**. WAEs are not trained via maximum-likelihood, so Theorem 1 does not imply that manifold overfitting should occur.

### D.3 EBM Improvements

Following Du & Mordatch (2019), we evaluated the single-step EBM's sample quality on the basis of samples initialized from the training buffer. However, when MCMC samples were initialized from uniform noise, we observed that all samples would converge to a small collection of low-quality modes (see Fig. 13). Moreover, at each training epoch, these modes would change, even as the loss value decreased.

The described non-convergence in the EBM's model distribution is consistent with Corollary 1. On the other hand, when used as a low-dimensional density estimator in the two-step procedure, this problem vanished: MCMC samples initialized from random noise yielded diverse images. See Fig. 13 for a comparison.

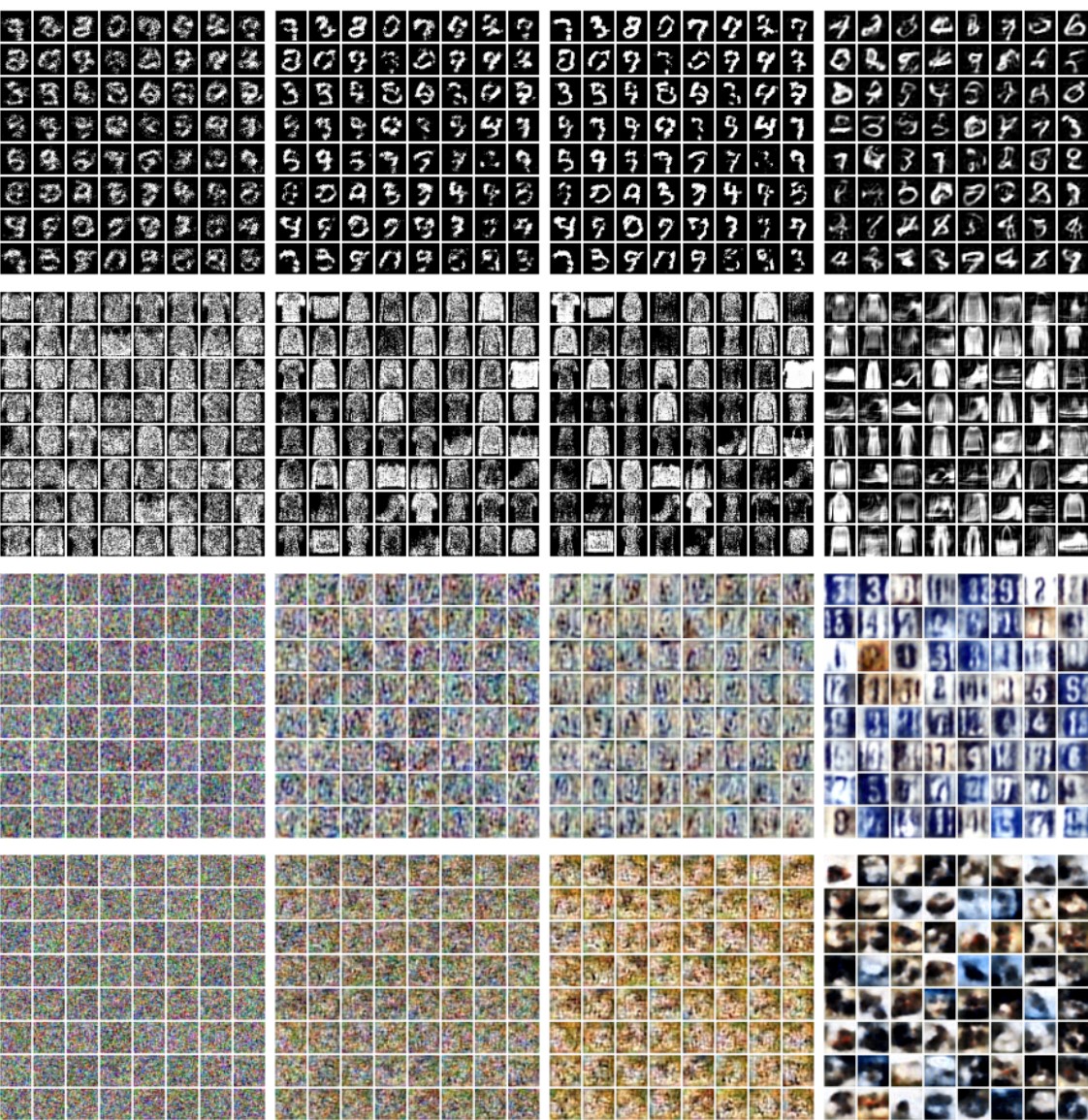

Figure 13: Uncurated samples with Langevin dynamics initialized from random noise (with no buffer) trained on MNIST (**first row**), FMNIST (**second row**), SVHN (**third row**), and CIFAR-10 (**fourth row**). Models are EBM$^+$ with 60 steps (**first column**), EBM$^+$ with 200 steps (**second column**), EBM$^+$ with 500 steps (**third column**), and AE + EBM with 60 steps, (**fourth column**).

### D.4 FID, Precision, and Recall Scores

We show in Tables 4 and 5 precision and recall (along with FID) of all the models used in Sec. 6.2. We opt for the precision and recall scores of Kynkäänniemi et al. (2019) rather than those of Sajjadi et al. (2018) as the former aim to improve on the latter. We also tried the density and coverage metrics proposed by Naeem et al. (2020), but found these metrics to correlate with visual quality less than FID. Similarly, we also considered using the inception score (Salimans et al., 2016), but this metric is known to have issues (Barratt & Sharma, 2018), and the FID is widely preferred over it. We can see in Tables 4 and 5 that two-step models consistently outperform single-step models in recall, while either also outperforming or not underperforming in precision. Much like with FID score, some instances of AE+ARM have worse scores on both precision and recall than their corresponding single-step model. Given the superior visual quality of those two-step models, we also consider these as failure cases of the evaluation metrics themselves, which we highlight in red in Tables 4 and 5. We believe that some non-highlighted results do not properly reflect the magnitude by which the two-step models outperformed single-step models, and encourage the reader to see the corresponding samples.

We show in Table 6 the FID scores of models involving BiGANs and WAEs. These methods are not trained via maximum likelihood, so Theorem 1 does not apply. In contrast to the likelihood-based models from Table 1, there is no significant improvement in FID for BiGANs and WAEs from using a two-step approach, and sometimes two-step models perform worse. However, for BiGANs we observe similar visual quality in samples (see Fig. 11), once again highlighting a failure of the FID score as a metric. We show these failures with red in Table 6.

Table 4: FID (lower is better) and Precision, and Recall scores (higher is better). Means $\pm$ standard errors across 3 runs are shown. Unreliable scores are highlighted in red.

| MODEL | MNIST | | | FMNIST | | |
|---|---|---|---|---|---|---|
| | FID | Precision | Recall | FID | Precision | Recall |
| AVB | $219.0 \pm 4.2$ | $0.0000 \pm 0.0000$ | $0.0008 \pm 0.0007$ | $235.9 \pm 4.5$ | $0.0006 \pm 0.0000$ | $0.0086 \pm 0.0037$ |
| AVB$^+$ | $205.0 \pm 3.9$ | $0.0000 \pm 0.0000$ | $0.0106 \pm 0.0089$ | $216.2 \pm 3.9$ | $0.0008 \pm 0.0002$ | $0.0075 \pm 0.0052$ |
| AVB$^+_\sigma$ | $205.2 \pm 1.0$ | $0.0000 \pm 0.0000$ | $0.0065 \pm 0.0032$ | $223.8 \pm 5.4$ | $0.0007 \pm 0.0002$ | $0.0034 \pm 0.0009$ |
| AVB+ARM | $86.4 \pm 0.9$ | $0.0012 \pm 0.0003$ | $0.0051 \pm 0.0011$ | $78.0 \pm 0.9$ | $0.1069 \pm 0.0055$ | $0.0106 \pm 0.0011$ |
| AVB+AVB | $133.3 \pm 0.9$ | $0.0001 \pm 0.0000$ | $0.0093 \pm 0.0027$ | $143.9 \pm 2.5$ | $0.0151 \pm 0.0015$ | $0.0093 \pm 0.0019$ |
| AVB+EBM | $96.6 \pm 3.0$ | $0.0006 \pm 0.0000$ | $0.0021 \pm 0.0007$ | $103.3 \pm 1.4$ | $0.0386 \pm 0.0016$ | $0.0110 \pm 0.0013$ |
| AVB+NF | $83.5 \pm 2.0$ | $0.0009 \pm 0.0001$ | $0.0059 \pm 0.0015$ | $77.3 \pm 1.1$ | $0.1153 \pm 0.0031$ | $0.0092 \pm 0.0004$ |
| AVB+VAE | $106.2 \pm 2.5$ | $0.0005 \pm 0.0000$ | $0.0088 \pm 0.0005$ | $105.7 \pm 0.6$ | $0.0521 \pm 0.0035$ | $0.0166 \pm 0.0007$ |
| VAE | $197.4 \pm 1.5$ | $0.0000 \pm 0.0000$ | $0.0035 \pm 0.0004$ | $188.9 \pm 1.8$ | $0.0030 \pm 0.0006$ | $0.0270 \pm 0.0048$ |
| VAE$^+$ | $184.0 \pm 0.7$ | $0.0000 \pm 0.0000$ | $0.0036 \pm 0.0006$ | $179.1 \pm 0.2$ | $0.0025 \pm 0.0003$ | $0.0069 \pm 0.0012$ |
| VAE$^+_\sigma$ | $185.9 \pm 1.8$ | $0.0000 \pm 0.0000$ | $0.0070 \pm 0.0012$ | $183.4 \pm 0.7$ | $0.0027 \pm 0.0002$ | $0.0095 \pm 0.0036$ |
| VAE+ARM | $69.7 \pm 0.8$ | $0.0008 \pm 0.0000$ | $0.0041 \pm 0.0001$ | $70.9 \pm 1.0$ | $0.1485 \pm 0.0037$ | $0.0129 \pm 0.0011$ |
| VAE+AVB | $117.1 \pm 0.8$ | $0.0002 \pm 0.0000$ | $0.0123 \pm 0.0002$ | $129.6 \pm 3.1$ | $0.0291 \pm 0.0040$ | $0.0454 \pm 0.0046$ |
| VAE+EBM | $74.1 \pm 1.0$ | $0.0007 \pm 0.0001$ | $0.0015 \pm 0.0006$ | $78.7 \pm 2.2$ | $0.1275 \pm 0.0052$ | $0.0030 \pm 0.0002$ |
| VAE+NF | $70.3 \pm 0.7$ | $0.0009 \pm 0.0000$ | $0.0067 \pm 0.0011$ | $73.0 \pm 0.3$ | $0.1403 \pm 0.0022$ | $0.0116 \pm 0.0016$ |
| ARM$^+$ | $98.7 \pm 10.6$ | $0.0471 \pm 0.0098$ | $0.3795 \pm 0.0710$ | $72.7 \pm 2.1$ | $0.2005 \pm 0.0059$ | $0.4349 \pm 0.0143$ |
| ARM$^+_\sigma$ | $34.7 \pm 3.1$ | $0.0849 \pm 0.0112$ | $0.3349 \pm 0.0063$ | $23.1 \pm 0.9$ | $0.3508 \pm 0.0099$ | $0.5653 \pm 0.0092$ |
| AE+ARM | $72.0 \pm 1.3$ | $0.0006 \pm 0.0001$ | $0.0038 \pm 0.0003$ | $76.0 \pm 0.3$ | $0.0986 \pm 0.0038$ | $0.0069 \pm 0.0005$ |
| EBM$^+$ | $84.2 \pm 4.3$ | $0.4056 \pm 0.0145$ | $0.0008 \pm 0.0006$ | $135.6 \pm 1.6$ | $0.6550 \pm 0.0054$ | $0.0000 \pm 0.0000$ |
| EBM$^+_\sigma$ | $101.0 \pm 12.3$ | $0.3748 \pm 0.0496$ | $0.0013 \pm 0.0008$ | $135.3 \pm 0.9$ | $0.6384 \pm 0.0027$ | $0.0000 \pm 0.0000$ |
| AE+EBM | $75.4 \pm 2.3$ | $0.0007 \pm 0.0001$ | $0.0008 \pm 0.0002$ | $83.1 \pm 1.9$ | $0.0891 \pm 0.0046$ | $0.0037 \pm 0.0009$ |

Table 5: FID (lower is better) and Precision, and Recall scores (higher is better). Means $\pm$ standard errors across 3 runs are shown. Unreliable scores are highlighted in red.

| MODEL | SVHN | | | CIFAR-10 | | |
|---|---|---|---|---|---|---|
| | FID | Precision | Recall | FID | Precision | Recall |
| AVB | $356.3 \pm 10.2$ | $0.0148 \pm 0.0035$ | $0.0000 \pm 0.0000$ | $289.0 \pm 3.0$ | $0.0602 \pm 0.0111$ | $0.0000 \pm 0.0000$ |
| $\text{AVB}^+$ | $352.6 \pm 7.6$ | $0.0088 \pm 0.0018$ | $0.0000 \pm 0.0000$ | $297.1 \pm 1.1$ | $0.0902 \pm 0.0192$ | $0.0000 \pm 0.0000$ |
| $\text{AVB}^+_\sigma$ | $353.0 \pm 7.2$ | $0.0425 \pm 0.0293$ | $0.0000 \pm 0.0000$ | $305.8 \pm 8.7$ | $0.1304 \pm 0.0460$ | $0.0000 \pm 0.0000$ |
| AVB+ARM | $56.6 \pm 0.6$ | $0.6741 \pm 0.0090$ | $0.0206 \pm 0.0011$ | $182.5 \pm 1.0$ | $0.4670 \pm 0.0037$ | $0.0003 \pm 0.0001$ |
| AVB+AVB | $74.5 \pm 2.5$ | $0.5765 \pm 0.0157$ | $0.0224 \pm 0.0008$ | $183.9 \pm 1.7$ | $0.4617 \pm 0.0078$ | $0.0006 \pm 0.0003$ |
| AVB+EBM | $61.5 \pm 0.8$ | $0.6809 \pm 0.0092$ | $0.0162 \pm 0.0020$ | $189.7 \pm 1.8$ | $0.4543 \pm 0.0094$ | $0.0006 \pm 0.0002$ |
| AVB+NF | $55.4 \pm 0.8$ | $0.6724 \pm 0.0078$ | $0.0217 \pm 0.0007$ | $181.7 \pm 0.8$ | $0.4632 \pm 0.0024$ | $0.0009 \pm 0.0001$ |
| AVB+VAE | $59.9 \pm 1.3$ | $0.6698 \pm 0.0105$ | $0.0214 \pm 0.0010$ | $186.7 \pm 0.9$ | $0.4517 \pm 0.0046$ | $0.0006 \pm 0.0001$ |
| VAE | $311.5 \pm 6.9$ | $0.0098 \pm 0.0030$ | $0.0018 \pm 0.0012$ | $270.3 \pm 3.2$ | $0.0805 \pm 0.0016$ | $0.0000 \pm 0.0000$ |
| $\text{VAE}^+$ | $300.1 \pm 2.1$ | $0.0133 \pm 0.0014$ | $0.0000 \pm 0.0000$ | $257.8 \pm 0.6$ | $0.1287 \pm 0.0183$ | $0.0001 \pm 0.0000$ |
| $\text{VAE}^+_\sigma$ | $302.2 \pm 2.0$ | $0.0086 \pm 0.0018$ | $0.0004 \pm 0.0003$ | $257.8 \pm 1.7$ | $0.1328 \pm 0.0152$ | $0.0000 \pm 0.0000$ |
| VAE+ARM | $52.9 \pm 0.3$ | $0.7004 \pm 0.0016$ | $0.0234 \pm 0.0005$ | $175.2 \pm 1.3$ | $0.4865 \pm 0.0055$ | $0.0004 \pm 0.0001$ |
| VAE+AVB | $64.0 \pm 1.3$ | $0.6234 \pm 0.0110$ | $0.0273 \pm 0.0006$ | $176.7 \pm 2.0$ | $0.5140 \pm 0.0123$ | $0.0007 \pm 0.0002$ |
| VAE+EBM | $63.7 \pm 3.3$ | $0.6983 \pm 0.0071$ | $0.0163 \pm 0.0008$ | $181.7 \pm 2.8$ | $0.4849 \pm 0.0098$ | $0.0002 \pm 0.0001$ |
| VAE+NF | $52.9 \pm 0.3$ | $0.6902 \pm 0.0059$ | $0.0243 \pm 0.0011$ | $175.1 \pm 0.9$ | $0.4755 \pm 0.0095$ | $0.0007 \pm 0.0002$ |
| $\text{ARM}^+$ | $168.3 \pm 4.1$ | $0.1425 \pm 0.0086$ | $0.0759 \pm 0.0031$ | $162.6 \pm 2.2$ | $0.6093 \pm 0.0066$ | $0.0313 \pm 0.0061$ |
| $\text{ARM}^+_\sigma$ | $149.2 \pm 10.7$ | $0.1622 \pm 0.0210$ | $0.0961 \pm 0.0069$ | $136.1 \pm 4.2$ | $0.6585 \pm 0.0116$ | $0.0993 \pm 0.0106$ |
| AE+ARM | $60.1 \pm 3.0$ | $0.5790 \pm 0.0275$ | $0.0192 \pm 0.0014$ | $186.9 \pm 1.0$ | $0.4544 \pm 0.0073$ | $0.0008 \pm 0.0002$ |
| $\text{EBM}^+$ | $228.4 \pm 5.0$ | $0.0955 \pm 0.0367$ | $0.0000 \pm 0.0000$ | $201.4 \pm 7.9$ | $0.6345 \pm 0.0310$ | $0.0000 \pm 0.0000$ |
| $\text{EBM}^+_\sigma$ | $235.0 \pm 5.6$ | $0.0983 \pm 0.0183$ | $0.0000 \pm 0.0000$ | $200.6 \pm 4.8$ | $0.6380 \pm 0.0156$ | $0.0000 \pm 0.0000$ |
| AE+EBM | $75.2 \pm 4.1$ | $0.5739 \pm 0.0299$ | $0.0196 \pm 0.0035$ | $187.4 \pm 3.7$ | $0.4586 \pm 0.0117$ | $0.0006 \pm 0.0001$ |

Table 6: FID scores (lower is better) for non-likelihood based GAEs and two-step models. These GAEs are not trained to maximize likelihood, so Theorem 1 does not apply. Means $\pm$ standard errors across 3 runs are shown. Unreliable scores are shown in red. Samples for unreliable scores are provided in Fig. 11.

| MODEL | MNIST | FMNIST | SVHN | CIFAR-10 |
|---|---|---|---|---|
| BiGAN | $150.0 \pm 1.5$ | $139.0 \pm 1.0$ | $105.5 \pm 5.2$ | $170.9 \pm 4.3$ |
| $\text{BiGAN}^+$ | $135.2 \pm 0.2$ | $113.0 \pm 0.6$ | $114.4 \pm 4.9$ | $152.9 \pm 0.6$ |
| BiGAN+ARM | $112.6 \pm 1.6$ | $94.9 \pm 0.7$ | $60.8 \pm 1.6$ | $210.7 \pm 1.6$ |
| BiGAN+AVB | $149.9 \pm 3.3$ | $141.5 \pm 1.7$ | $67.2 \pm 2.6$ | $215.7 \pm 1.0$ |
| BiGAN+EBM | $120.7 \pm 4.7$ | $108.1 \pm 2.4$ | $66.5 \pm 1.3$ | $217.5 \pm 1.8$ |
| BiGAN+NF | $112.4 \pm 1.4$ | $95.0 \pm 0.8$ | $60.2 \pm 1.5$ | $211.6 \pm 1.7$ |
| BiGAN+VAE | $127.9 \pm 1.6$ | $115.5 \pm 1.4$ | $63.6 \pm 1.4$ | $216.3 \pm 1.2$ |
| WAE | $19.8 \pm 1.6$ | $45.1 \pm 0.8$ | $52.7 \pm 0.6$ | $187.4 \pm 0.4$ |
| $\text{WAE}^+$ | $16.7 \pm 0.4$ | $45.2 \pm 0.2$ | $53.2 \pm 0.4$ | $179.7 \pm 1.3$ |
| WAE+ARM | $15.2 \pm 0.5$ | $46.1 \pm 0.3$ | $73.1 \pm 1.8$ | $182.3 \pm 1.7$ |
| WAE+AVB | $17.6 \pm 0.3$ | $47.7 \pm 0.9$ | $60.2 \pm 3.8$ | $157.6 \pm 0.8$ |
| WAE+EBM | $23.7 \pm 1.0$ | $60.2 \pm 1.4$ | $70.6 \pm 1.5$ | $161.0 \pm 4.7$ |
| WAE+NF | $20.7 \pm 2.2$ | $52.1 \pm 2.9$ | $57.6 \pm 3.8$ | $178.2 \pm 2.8$ |
| WAE+VAE | $16.4 \pm 0.6$ | $50.9 \pm 0.5$ | $72.2 \pm 1.9$ | $178.3 \pm 2.6$ |

## D.5 High Resolution Image Generation

As mentioned in the main manuscript, we attempted to use our two-step methodology to improve upon a high-performing GAN model: a StyleGAN2 (Karras et al., 2020b). We used the PyTorch (Paszke et al., 2019) code of Karras et al. (2020a), which implements the optimization-based projection method of Karras et al. (2020b). That is, we did not explicitly construct $g$, and used this optimization-based GAN inversion method to recover $\{z_n\}_{n=1}^N$ on the FFHQ dataset (Karras et al., 2019), with the intention of training low-dimensional DGMs to produce high resolution images. This method projects into the intermediate 512-dimensional space referred to as $\mathcal{W}$ by default (Karras et al., 2020b). We also adapted this method to the GAN's true latent space, referred to as $\mathcal{Z}$, during which we decreased the initial learning rate to 0.01 from the default

of 0.1. In experiments with optimization-based inversion into the latent spaces $g(\mathcal{M}) = \mathcal{W}$ and $g(\mathcal{M}) = \mathcal{Z}$, reconstructions $\{G(z_n)\}_{n=1}^N$ yielded FIDs of 13.00 and 25.87, respectively. In contrast, the StyleGAN2 achieves an FID score of 5.4 by itself, which is much better than the scores achieved by the reconstructions (perfect reconstructions would achieve scores of 0).

The FID between the reconstructions and the ground truth images represents an approximate lower-bound on the FID score attainable by the two-step method, since the second step estimates the distribution of the projected latents $\{z_n\}_{n=1}^N$. Since reconstructing the entire FFHQ dataset of 70000 images would be expensive (for instance, $\mathcal{W}$-space reconstructions take about 90 seconds per image), we computed the FID (again using the code of Karras et al. (2020a)) between the first 10000 images of FFHQ and their reconstructions.

We also experimented with the approach of Huh et al. (2020), which inverts into $\mathcal{Z}$-space, but it takes about 10 minutes per image and was thus prohibitively expensive. Most other GAN inversion work (Xia et al., 2021) has projected images into the extended $512 \times 18$-dimensional $\mathcal{W}+$ space, which describes a different intermediate latent input $w$ for each layer of the generator. Since this latent space is higher in dimension than the true model manifold, we did not pursue these approaches. The main obstacle to improving StyleGAN2's FID using the two-step procedure appears to be reconstruction quality. Since the goal of our experiments is to highlight the benefits of two-step procedures rather than proposing new GAN inversion methods, we did not further pursue this direction, although we hope our results will encourage research improving GAN inversion methods and exploring their benefits within two-step models.

### D.6    OOD Detection

**OOD Metric**    We now precisely describe our classification metric, which properly accounts for datasets of imbalanced size and ensures correct directionality, in that higher likelihoods are considered to be in-distribution. First, using the in- and out-of-sample training likelihoods, we train a decision stump – i.e. a single-threshold-based classifier. Then, calling that threshold $T$, we count the number of in-sample test likelihoods which are greater than $T$, $n_{I>T}$, and the number of out-of-sample test likelihoods which are greater than $T$, $n_{O>T}$. Then, calling the number of in-sample test points $n_I$, and the number of OOD test points $n_O$, our final classification rate `acc` is given as:

$$\texttt{acc} = \frac{n_{I>T} + \frac{n_I}{n_O} \cdot (n_O - n_{O>T})}{2n_I}. \tag{45}$$

Intuitively, we can think of this metric as simply the fraction of correctly-classified points (i.e. $\texttt{acc}' = \frac{n_{O>T} + (n_O - n_{I>T})}{n_I + n_O}$), but with the contributions from the OOD data re-weighted by a factor of $\frac{n_I}{n_O}$ to ensure both datasets are equally weighted in the metric. Note that this metric is sometimes referred to as balanced accuracy, and can also be understood as the average between the true positive and true negative rates.

We show further OOD detection results using $\log p_Z$ in Table 7, and using $\log p_X$ in Table 8. Note that, for one-step models, we record results for $\log p_X$, the log-density of the model, in place of $\log p_Z$ (which is not defined).

Table 7: OOD classification accuracy as a percentage (higher is better), using $\log p_Z$. Means $\pm$ standard errors across 3 runs are shown. Arrows point from in-distribution to OOD data.

| MODEL | FMNIST $\rightarrow$ MNIST | CIFAR-10 $\rightarrow$ SVHN |
|---|---|---|
| AVB$^+$ | $96.0 \pm 0.5$ | $23.4 \pm 0.1$ |
| AVB+ARM | $89.9 \pm 2.4$ | $40.6 \pm 0.2$ |
| AVB+AVB | $74.4 \pm 2.2$ | $45.2 \pm 0.2$ |
| AVB+EBM | $49.5 \pm 0.1$ | $49.0 \pm 0.0$ |
| AVB+NF | $89.2 \pm 0.9$ | $46.3 \pm 0.9$ |
| AVB+VAE | $78.4 \pm 1.5$ | $40.2 \pm 0.1$ |
| VAE$^+$ | $96.1 \pm 0.1$ | $23.8 \pm 0.2$ |
| VAE+ARM | $92.6 \pm 1.0$ | $39.7 \pm 0.4$ |
| VAE+AVB | $80.6 \pm 2.0$ | $45.4 \pm 1.1$ |
| VAE+EBM | $54.1 \pm 0.7$ | $49.2 \pm 0.0$ |
| VAE+NF | $91.7 \pm 0.3$ | $47.1 \pm 0.1$ |
| ARM$^+$ | $9.9 \pm 0.6$ | $15.5 \pm 0.0$ |
| AE+ARM | $86.5 \pm 0.9$ | $37.4 \pm 0.2$ |
| EBM$^+$ | $32.5 \pm 1.1$ | $46.4 \pm 3.1$ |
| AE+EBM | $50.9 \pm 0.2$ | $49.4 \pm 0.6$ |

Table 8: OOD classification accuracy as a percentage (higher is better), using $\log p_X$. Means $\pm$ standard errors across 3 runs are shown. Arrows point from in-distribution to OOD data.

| MODEL | FMNIST $\rightarrow$ MNIST | CIFAR-10 $\rightarrow$ SVHN |
|---|---|---|
| AVB$^+$ | $96.0 \pm 0.5$ | $23.4 \pm 0.1$ |
| AVB+ARM | $90.8 \pm 1.8$ | $37.7 \pm 0.5$ |
| AVB+AVB | $75.0 \pm 2.2$ | $43.7 \pm 2.0$ |
| AVB+EBM | $53.3 \pm 7.1$ | $39.1 \pm 0.9$ |
| AVB+NF | $89.2 \pm 0.8$ | $43.9 \pm 1.3$ |
| AVB+VAE | $78.7 \pm 1.6$ | $40.2 \pm 0.2$ |
| VAE$^+$ | $96.1 \pm 0.1$ | $23.8 \pm 0.2$ |
| VAE+ARM | $93.7 \pm 0.7$ | $37.6 \pm 0.4$ |
| VAE+AVB | $82.4 \pm 2.4$ | $42.2 \pm 1.0$ |
| VAE+EBM | $63.7 \pm 1.7$ | $42.4 \pm 0.9$ |
| VAE+NF | $91.7 \pm 0.3$ | $42.4 \pm 0.3$ |
| ARM$^+$ | $9.9 \pm 0.6$ | $15.5 \pm 0.0$ |
| AE+ARM | $89.5 \pm 0.2$ | $33.8 \pm 0.3$ |
| EBM$^+$ | $32.5 \pm 1.1$ | $46.4 \pm 3.1$ |
| AE+EBM | $56.9 \pm 14.4$ | $34.5 \pm 0.1$ |
| BiGAN+ARM | $81.5 \pm 1.4$ | $35.7 \pm 0.4$ |
| BiGAN+AVB | $59.6 \pm 3.2$ | $34.3 \pm 2.3$ |
| BiGAN+EBM | $57.4 \pm 1.7$ | $47.7 \pm 0.7$ |
| BiGAN+NF | $83.7 \pm 1.2$ | $39.2 \pm 0.3$ |
| BiGAN+VAE | $59.3 \pm 2.1$ | $35.6 \pm 0.4$ |
| WAE+ARM | $89.0 \pm 0.5$ | $38.1 \pm 0.6$ |
| WAE+AVB | $74.5 \pm 1.3$ | $43.1 \pm 0.7$ |
| WAE+EBM | $36.5 \pm 1.6$ | $36.8 \pm 0.4$ |
| WAE+NF | $85.7 \pm 2.8$ | $40.2 \pm 1.8$ |
| WAE+VAE | $87.7 \pm 0.7$ | $38.3 \pm 0.4$ |

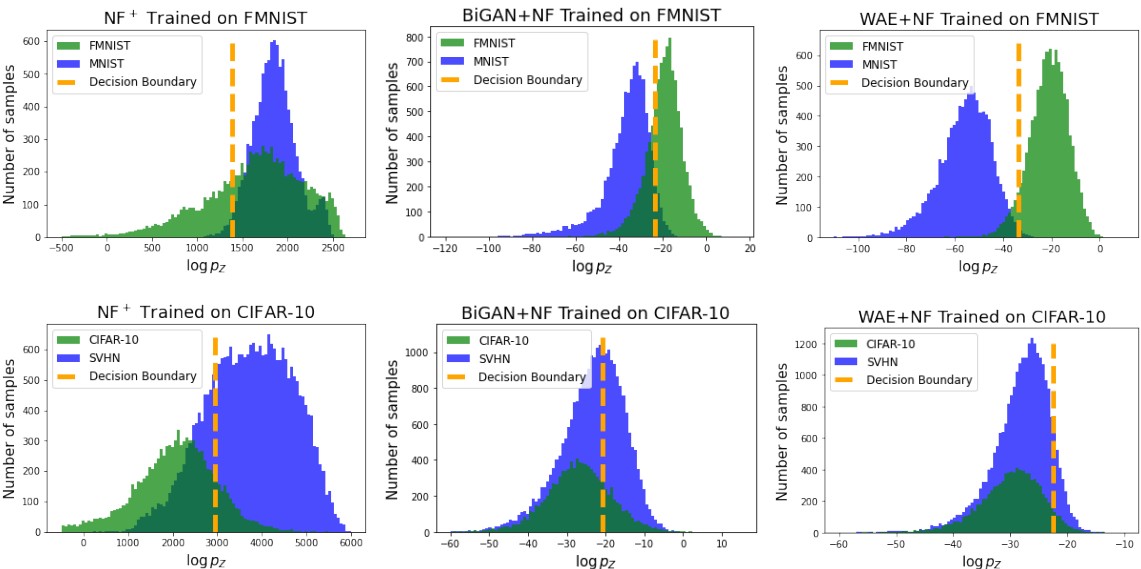

Figure 14: Comparison of the distribution of log-likelihood values between in-distribution (green) and out-of-distribution (blue) data. In both cases, the two-step models push the in-distribution likelihoods further to the right than the NF$^+$ model alone. *N.B.*: The absolute value of the likelihoods in the NF$^+$ model on its own are off by a constant factor because of the aforementioned whitening transform used to scale the data before training. However, the *relative* value within a single plot remains correct.

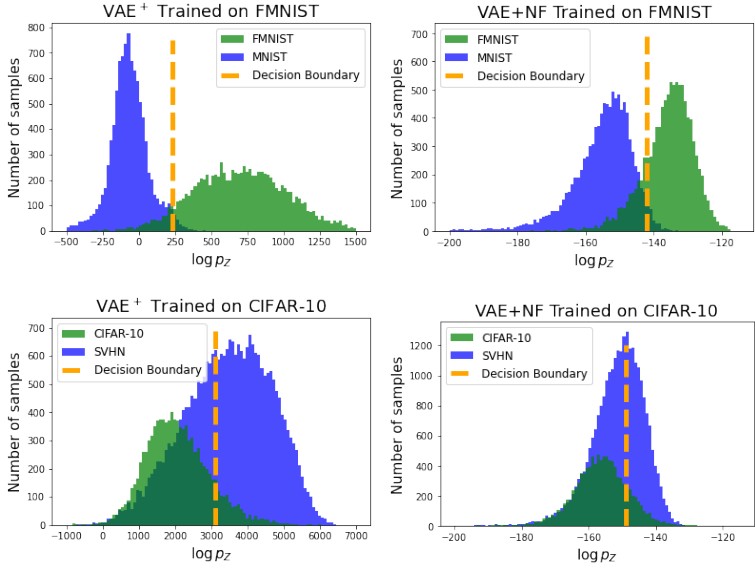

Figure 15: Comparison of the distribution of log-likelihood values between in-distribution (green) and out-of-distribution (blue) data for VAE-based models. While the VAE$^+$ model does well on FMNIST→MNIST, its performance is poor for CIFAR-10→SVHN. The two-step model VAE+NF improves on the CIFAR-10→SVHN task.

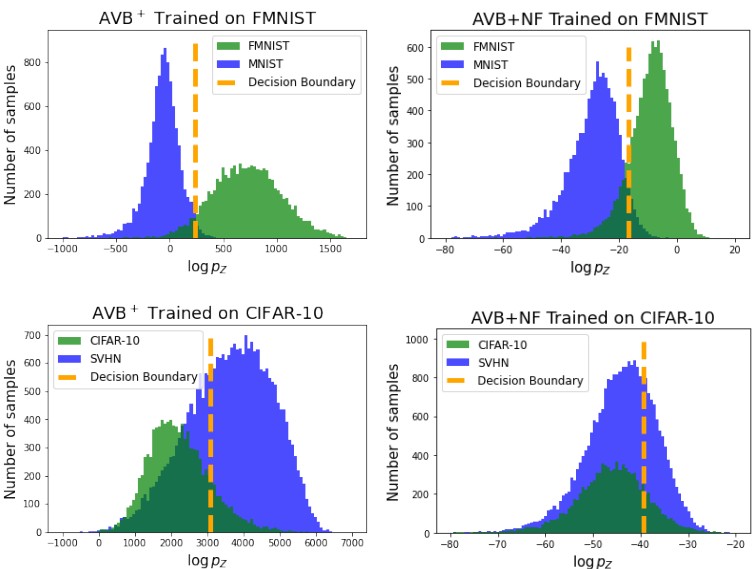

Figure 16: Comparison of the distribution of log-likelihood values between in-distribution (green) and out-of-distribution (blue) data for AVB-based models. While the AVB$^+$ model does well on FMNIST→MNIST, its performance is poor for CIFAR-10→SVHN. The two-step model AVB+NF improves on the CIFAR-10→SVHN task.

