# OpenReview forum: "Diagnosing and Fixing Manifold Overfitting in Deep Generative Models"
_TMLR — Accepted by TMLR_

### Review · Reviewer_SQHa · 2022-06-06

**Summary Of Contributions:**

The paper is concerned with learning a data distribution P* in R^D where the data exists on a manifold in R^D. The authors show that max-likelihood approaches lead to generative models assigning density to the correct values in R^D but do not assign the correct amount of density. The authors refer to this as manifold overfitting.

The authors provide theoretical explanation for this and propose a two-step solution, where by they encode (to a latent) and decode an image, but match the distributions of the latent distributions (as well as ensuring decoder(encoder(x)) = x).

**Broader Impact Concerns:**

Generative models have been used for data augmentation and for learning representations in low data regimes. If existing models *can* place more density on lower probability (i.e. under-represented) samples this is may help a model learn representations for those samples(?). In your setup you encourage the model not to assign as much density to less likely samples, could this perpetuate bias and reduce usefulness of learning in low data regimes? Your OOD result were encouraging, but how well does your model perform compared to single step models when there are some under-represented classes? Some simple experiment on MNIST would be interesting.

**Requested Changes:**

The following changes are critical:

Page 1: M \in R^D should this be d rather than D? This sentence was confusing “unknown d-dimensional manifold M ⊂ R^D, where d < D.”

The authors say that “This modelling choice implies the model has D-dimensional support” however, many models produce samples from a latent distribution which has a dimension << D. Could the authors comment on the connection here, please.

Please provide more information about how d is selected?

Please provide examples of losses used, this will add significantly to the understanding of the paper and make it more practical for others to benefit from this work.

Great to have:
How well does your model perform compared to single step models when there are some under-represented classes?
How does the values of "d" in your model affect performance?


**Strengths And Weaknesses:**

Clarifying question:
Page 1: M \in R^D should this be d rather than D? This sentence was confusing “unknown d-dimensional manifold M ⊂ R^D, where d < D.”

The authors say that “This modelling choice implies the model has D-dimensional support” however, many models produce samples from a latent distribution which has a dimension << D. Could the authors comment on the connection here, please.

Can the authors please check for consistency of D and d notation.

Figure 2: What peak values are expected for the manifold in Figure 2 Left panel? Is it that the area under the graph surrounding each data point integrates to 0.3 and 0.7 and hence he peeks are c.0.6 and c.1.4 respectively? This would be good to clarify please?

Following the example from 3.1 it would be very helpful to be explicit that the “manifold overfitting” problem is not that the model can assign density to the correct values, but that it applies the incorrect density to the point values.

Q:  “The underlying issue here is that M is “too thin in R^D” , and thus p(x) can “spike to infinity” at every x ∈ M.” Is this not true in all dimensions if density is at a point?

Practically speaking, why does it matter if the model overfits to the manifold, provided that density is assigned to the correct parts of the distribution?

Pg6 “P* -almost surely”, not sure of the role of P* in this sentence?

Figure 4: No upper limit value on the colour bar.

Q: How do you pick the dimension of Pz? If this is too large the same problems will exist and if it’s too small you will not be able to capture the manifold?

Q: Optimising (1) does not guarantee a good decoder? And it is possible to learn a g() that maps all samples to the same value (for example). You need a good decoder to avoid this.

“The condition G(g(x)) = x, P ∗ -almost surely, is what one should expect to obtain during the dimensionality reduction step” is also contradicted by Figure 3?


Comments:
“a DGM is performed” <— this sentence does not make sense.

It may make more sense to use more standard generative model notation (either from GAN or VAE literature).

The example in 3.1 is nice to have and it’s great to have the accompanying figure.

State explicitly that m and sigma are mean and standard deviation.

This sentence was a great summary: “The underlying issue here is that M is “too thin in R^D” , and thus p(x) can “spike to infinity” at every x ∈ M.”

This is a very general framework and many existing generative models could easily benefit from this approach. Perhaps it is worth emphasising this more?

Results in Figure 4 appear convincing, but please add a scale to the colour bar.

Very extensive experimentation in Table 1 on a number of datasets and with many model variations. How does the dimension of Pz affect these results?

Table 1 and 2 highlight in bold the best results in each sub-part of the table.

It would help to have one concrete example of the losses used to train a model.

---

> ### Author Response · Authors · 2022-06-11
> **Response to reviewer SQHa (part 1)**
>
> We thank the reviewer for their review, and are pleased that they found our framework highly general and potentially benefiting many existing generative models (we will further highlight this upon revising our manuscript as suggested). We are also glad they considered our experiments "extensive", and appreciated the example in section 3.1. Please see our points below, which we hope address their concerns:
>
> 1. `Page 1: M \in R^D should this be d rather than D? This sentence was confusing “unknown d-dimensional manifold M ⊂ R^D, where d < D.”` `Can the authors please check for consistency of D and d notation.`
>
> Note that $\mathcal{M}$ is not in (i.e. $\in$) $\mathbb{R}^D$, rather $\mathcal{M}$ is a subset (i.e. $\subset$) of $\mathbb{R}^D$. We have reviewed the notation throughout the manuscript and are confident that the notation is consistent. For clarity, $d$ refers to the intrinsic dimensionality of the data, while $D$ refers to the dimension of the ambient space used to represent it. As an example, the left panel of Figure 4 shows a 1-dimensional manifold (the circle) as a subset of $\mathbb{R}^2$. This is indeed at the very core of the manifold hypothesis, which essentially posits the existence of low-dimensional structure in high-dimensional data of interest. If the reviewer still believes the current manuscript is unclear, we will change $\mathcal{M} \subset \mathbb{R}^D$ on page 1 and say that $\mathcal{M}$ is a $d$-dimensional embedded submanifold of $\mathbb{R}^D$, where $d<D$.
>
>
> 2. `The authors say that “This modelling choice implies the model has D-dimensional support” however, many models produce samples from a latent distribution which has a dimension << D. Could the authors comment on the connection here, please.`
>
> Note that the dimension of the latent variable need not be equal to the support of the distribution. For example, a VAE with a Gaussian decoder $p(x\mid z)$ will by construction assign positive density to any $x \in \mathbb{R}^D$, thus resulting in $D$-dimensional support, regardless of the dimension of $z$. The same is true for AVB, and these two models are the only two likelihood-based models that we consider which use low-dimensional latent variables (NFs have latent variables of dimension $D$, EBMs and ARMs have no latent variables, and GANs and WAEs are not likelihood-based). Another way to understand this is that all likelihood-based models have positive densities on the data space $\mathbb{R}^D$, and thus have $D$-dimensional supports.
>
> 3. `Figure 2: What peak values are expected for the manifold in Figure 2 Left panel? Is it that the area under the graph surrounding each data point integrates to 0.3 and 0.7 and hence he peeks are c.0.6 and c.1.4 respectively? This would be good to clarify please?`
>
> Indeed, the behaviour we labelled as "intended" on the left panel of Figure 2 would be such that the density roughly assigns probability $0.3$ to a small neighbourhood of $-1$, and $0.7$ to a small neighbourhood of $1$. We refer to this behaviour as "intended" as it corresponds to recovering the ground truth distribution. The actual peaks achieved by this intended density are however of no particular relevance: we expect the values at the peaks to go to infinity as $\sigma \rightarrow 0$.
>
> 4. `Following the example from 3.1 it would be very helpful to be explicit that the “manifold overfitting” problem is not that the model can assign density to the correct values, but that it applies the incorrect density to the point values.`
>
> We thank the reviewer for the suggestion, and will modify the manuscript accordingly.
>
> 5. `Q: “The underlying issue here is that M is “too thin in R^D” , and thus p(x) can “spike to infinity” at every x ∈ M.” Is this not true in all dimensions if density is at a point?`
>
> We are not sure what is meant by "if density is at a point", but suspect the reviewer might be thinking of regular overfitting where likelihood is concentrated only at the observed points, preventing successful generalization. Our explanation that density will ``spike to infinity'' around the manifold really does require $d < D$, and we note that the statement here refers to every point on the manifold $\mathcal{M}$, not just the observed data (which lies on the manifold of course). If the reviewer would kindly rephrase the question we are happy to discuss more.

---

> > ### Author Response · Authors · 2022-06-11
> > **Response to reviewer SQHa (part 2)**
> >
> > 6. `Practically speaking, why does it matter if the model overfits to the manifold, provided that density is assigned to the correct parts of the distribution?`
> >
> > When manifold overfitting occurs, high likelihoods (tending to infinity) are assigned to the manifold but not in a manner that recovers the target distribution, as shown in section 3, figure 2, and figure 4. Practically this means sampling from a model that has overfit to the manifold will produce realistic samples, but not in the proportions one would expect according to the target distribution.
> > For example, a model that has overfit to the MNIST manifold may produce a significantly unequal distribution of digits, which contrasts with the equiprobability of each digit in the groundtruth MNIST dataset (although this is not the only was that manifold overfitting may occur!). Additionally, when inference is performed on new samples the model will effectively assign likelihoods arbitrarily, again not respecting the target distribution.
> >
> > 7. `Pg6 “P* -almost surely”, not sure of the role of P* in this sentence?`
> > `“The condition G(g(x)) = x, P ∗ -almost surely, is what one should expect to obtain during the dimensionality reduction step” is also contradicted by Figure 3?`
> >
> > This is a technical, measure-theoretic requirement. It can be understood as a slightly weaker requirement than $G(g(x))=x$ for every $x \in \mathcal{M}$. The modifier ``$\mathbb{P}^\*$-almost surely" means we require the set of $x$s for which $G(g(x))=x$ holds to be assigned probability $1$ under $\mathbb{P}^\*$. As mentioned in the manuscript, this assumption is exactly what one should expect when perfectly minimizing reconstruction error. Subsets of off-manifold points are not assigned positive probability by $\mathbb{P}^*$, so $G(g(x))=x$ is not required to hold off-manifold, which means Figure 3 does $\textbf{not}$ contradict the condition. In the figure, $G$ and $g$ properly biject between $\mathcal{M}$ and $g(\mathcal{M})$, but not between $\mathbb{R}^D$ and $\mathbb{R}^d$. We will include an explanation of a property holding almost surely in the measure theory primer in the appendix.
> >
> > 8. `Figure 4: No upper limit value on the colour bar.`
> > `Results in Figure 4 appear convincing, but please add a scale to the colour bar.`
> >
> > We are pleased that the reviewer found these results convincing. In this case we intentionally left off a scale, as any upper value would be uninformative: the middle and right plots show $p$ up to its normalizing constant, as the models shown are EBMs, and thus any re-scaling of the functions would be equally valid. The left panel indeed corresponds to a density, but we did not include a value on the y-axis for consistency with the other panels, and since we do not think showing this value to be relevant to the point being made in the figure, even if properly normalized.
> >
> > 9. `Q: How do you pick the dimension of Pz? If this is too large the same problems will exist and if it’s too small you will not be able to capture the manifold?`
> > `Very extensive experimentation in Table 1 on a number of datasets and with many model variations. How does the dimension of Pz affect these results?`
> >
> > Please see point 1 of our general rebuttal.
> >
> > 10. `Q: Optimising (1) does not guarantee a good decoder? And it is possible to learn a g() that maps all samples to the same value (for example). You need a good decoder to avoid this.`
> >
> > Note that a fundamental aspect of two-step models is that $g$ and $G$ are trained $\textbf{first}$, and then $p\_Z$ is trained later. In other words, equation (1) is not optimized with respect to $g$, only with respect to $p_Z$: $g$ is fixed at this point.
> >
> > 11. `Comments: “a DGM is performed” <— this sentence does not make sense.`
> >
> > Thank you for pointing out an area where we can further improve the clarity of the writing. We will rephrase this sentence as "In the second step, maximum-likelihood estimation is performed on the low-dimensional representations $(g(x_n))_{n=1}^N$ using a DGM."
> >
> > 12. `It may make more sense to use more standard generative model notation (either from GAN or VAE literature).`
> >
> > We use standard notation as much as possible, for example, using $G$ as the "generator", and $z_n$ as the "latent variable" for the $n^{th}$ datapoint, $x_n$. However, since we present a unifying framework, we are not able to always exactly use the most prevalent notation from each particular DGM without sacrificing consistency of our own notation. We also note that, due to the nature of our results, some measure-theoretic notation cannot be avoided. We need to differentiate between probability distributions (measures) and densities (which we also denote in the common notation as $p_X$ and $p_Z$).
> >
> > 13. `State explicitly that m and sigma are mean and standard deviation.`
> >
> > Thank you for your suggestion; we will include this when we revise the manuscript.

---

> > > ### Author Response · Authors · 2022-06-11
> > > **Response to reviewer SQHa (part 3)**
> > >
> > > 14. `Table 1 and 2 highlight in bold the best results in each sub-part of the table.`
> > >
> > > We purposefully opted against bolding; the main point of Table 1 is to highlight the difference in performance between two-step and single-step models. We did not extensively tune our two-step models, and the best performing model on the table should not be taken as the best GAE+DGM combination, as tuning could likely change the ordering of the two-step model's results. We believe bolding the best performing models would shift attention to the best performing model rather than to the fact that two-step models consistently outperform single-step ones trained through maximum-likelihood, which is our conclusion from the experiments. Similarly, Table 2 aims to show that our result on density estimation for implicit models can be used for OOD detection, but we did not extensively tune the two-step models. We thus believe that the best performing model is irrelevant here, since the main point of focus should be that two-step models outperform single-step models.
> > >
> > > 15. `It would help to have one concrete example of the losses used to train a model.`
> > >
> > > The losses we used for all our models are the standard ones from the relevant papers that developed them (which we cite), including:
> > >
> > > - AEs: reconstruction error.
> > > - ARMs: log-likelihood.
> > > - Gaussian VAEs (and AVB): negative ELBO.
> > > - NFs: log-likelihood.
> > > - EBMs: log-likelihood.
> > > - BiGANs: we use a Wasserstein loss with gradient penalty (we specify this in the appendix).
> > > - WAEs: we use the adversarial loss variant, rather than the MMD one.
> > >
> > > The only difference with how we train our models compared to "standard training" is the two-step nature of our models, as also mentioned in answer 10.
> > >
> > > 16. **Please see our general rebuttal for our answer to comments about broader impact and potentially imbalanced datasets.**
> > >
> > > Finally, we believe the clarifications above address all the raised concerns, please let us know if this is not the case.

---

> > > > ### Comment · Reviewer_SQHa · 2022-06-21
> > > > **Thanks for the response.**
> > > >
> > > > I would like to thank the authors for their clear response. I'm happy with the rebuttal, provided that the suggested changes are included in the final paper.

---

### Review · Reviewer_UUHX · 2022-06-09

**Summary Of Contributions:**

Leveraging measure theory, this paper analyzes the difficulty of maximum likelihood training when the true data density lies on a low-dimensional manifold. An alternative method is thus proposed to address this difficulty, with a two-step idea where a (generalized) autoencoder is first trained to capture the low dimensional manifold before a second likelihood-based model is trained with samples in the latent space. Experiments have demonstrated that this two-step training procedure leads to higher sample quality as well as improved outlier detection results.

**Broader Impact Concerns:**

I disagree with the authors that their approach does not have any negative societal consequences. Two-step training leads to better generative models, which themselves have a lot of issues when applied inappropriately in the real world.

**Requested Changes:**

1. Include model parameter counts as an additional column in Table 1.
2. Perform experiments on synthetic datasets that do not lie on low-dimensional manifolds, and check if the results are consistent with authors' theory.

**Strengths And Weaknesses:**

# Strengths

* Theoretical analysis is rigorous and has minimal assumptions. The theoretical investigation and the proposed two step training can explain practical tricks widely used in many previous works on generative modeling.

* Comprehensive discussion of related work.

# Weaknesses

* Although having a rigorous theoretical analysis is desirable, the proposed two-step training has already been widely used in the community. It is also widely accepted that combining manifold learning and likelihood-based generative modeling can lead to better sample quality (see, e.g., [1]) and better results on outlier detection (see, e.g., [2]). The experimental results are therefore not surprising and do not provide much information to the research community.

* Experiments can be more rigorous:
  1. When comparing with baselines, need to make sure the improved performance is not due to having more model parameters. I know authors have briefly described the difference in model parameters in the main text, but I think it can be more clear to include such information directly in Table 1.
  2. Need to verify whether the performance improvement of two-step training only happens for data on low dimensional manifolds. Authors should report results on synthetic datasets in which datapoints do not have a low-dimensional submanifold structure.

* Paper can be more comprehensive by including additional related work in the discussion. For example, ref. [3] discusses the difficulty of learning autoregressive generative models on low-dimensional data manifolds, and proposes to perturb data with Gaussian noise before training autoregressive models for significantly improved sample quality.

References:

[1] Razavi, A., Van den Oord, A., & Vinyals, O. (2019). Generating diverse high-fidelity images with vq-vae-2. Advances in neural information processing systems, 32.

[2] Caterini, A. L., Loaiza-Ganem, G., Pleiss, G., & Cunningham, J. P. (2021). Rectangular flows for manifold learning. Advances in Neural Information Processing Systems, 34.

[3] Meng, C., Song, J., Song, Y., Zhao, S., & Ermon, S. (2021). Improved autoregressive modeling with distribution smoothing. arXiv preprint arXiv:2103.15089.

---

> ### Author Response · Authors · 2022-06-11
> **Response to reviewer UUHX**
>
> We thank the reviewer for their review, and for finding that our work is theoretically rigorous, uses minimal assumptions, and explains widely observed phenomena in deep generative models. Please see our points below, which we hope address their concerns:
>
> 1. `Although having a rigorous theoretical analysis is desirable, the proposed two-step training has already been widely used in the community. It is also widely accepted that combining manifold learning and likelihood-based generative modeling can lead to better sample quality (see, e.g., [1]) and better results on outlier detection (see, e.g., [2]). The experimental results are therefore not surprising and do not provide much information to the research community.`
>
> We actually think of this as a major strength: our work theoretically justifies these procedures. We see some of these methods already being in use as making our work immediately relevant.  We also point out that previous work has always proposed specific instances of two-step models, and our work greatly generalizes this type of model, which again we see as a strength. We also highlight that Caterini et al. [2] do not report OOD results on CIFAR-10->SVHN due to scaling issues. While indeed good performance for OOD might not be surprising, once again, we see this good performance as a strength. Additionally, we also point out that the main goal of section 6.3 and table 2 is to show that implicit models such as WAEs and BiGANs can be made explicit. We believe this is actually quite unexpected, and yet again a strength of our work. We also point out that we cite and discuss Razavi et al. [1]. We will happily keep discussing if the reviewer further explains why they consider these points as weaknesses.
>
> 2. `When comparing with baselines, need to make sure the improved performance is not due to having more model parameters. I know authors have briefly described the difference in model parameters in the main text, but I think it can be more clear to include such information directly in Table 1.`
>
> While including this information in Table 1 causes spacing issues and adds many numbers to an already packed table, we agree this information is valuable and should be added somewhere, and thank the reviewer for the suggestion. We will include a table detailing the number of parameters of all models from Table 1 in the appendix, along with a discussion.
>
> 3. `Need to verify whether the performance improvement of two-step training only happens for data on low dimensional manifolds. Authors should report results on synthetic datasets in which datapoints do not have a low-dimensional submanifold structure.`
>
> We thank the reviewer for this suggestion, as we agree that showing that a synthetic experiment where $(i)$ the data has no manifold structure and is known to be $D$-dimensional, and $(ii)$ we show similar performance between a single-step likelihood-based model and a two-step model where the dimension of $\mathbb{P}_Z$ is set to $D$, would indeed further confirm our theoretical results (although we also point out that two-step models outperforming single-step ones in this setting would not necessarily be contradictory to our results either, as two-step models may have further benefits that we have not explicitly studied). We ask the reviewer to please let us know if we misunderstood their suggestion, and will otherwise include such an experiment in the appendix upon revising the paper.
>
> 4. `Paper can be more comprehensive by including additional related work in the discussion. For example, ref. [3] discusses the difficulty of learning autoregressive generative models on low-dimensional data manifolds, and proposes to perturb data with Gaussian noise before training autoregressive models for significantly improved sample quality.`
>
> We thank the reviewer for pointing out the work of Meng et al. [3], which we will cite in the "adding noise" paragraph of section 2. We note that our first theorem can easily be understood as justifying the method of Meng et al. [3] in a rigorous manner, just as other methods adding noise can be understood as attempting to avoid manifold overfitting. However, we are somewhat confused if there are more references that we should consider adding, as the reviewer has also said the `"comprehensive discussion of related work"` was a strength of the paper. Can this please be clarified?
>
> 5. **Please see our general rebuttal for our answer to comments about broader impact.**
>
> Finally, we believe the clarifications above address all the raised concerns, please let us know if this is not the case.

---

> > ### Comment · Reviewer_UUHX · 2022-06-11
> > **Thanks for the response**
> >
> > I would like to thank the authors for providing a detailed response very quickly.
> >
> > Regarding the extra experiment, I have one additional suggestion: the dimensionality of $P_z$ needs to be smaller than $D$ and should be set to the same as in other (existing) experiments. The goal is to verify whether the performance gain of two-step models can be fully explained by authors' analysis on low-dimensional data distributions. It would be a bad sign if two-step models still significantly outperform baselines in this synthetic experiment, as this will indicate the existence of confounders and challenge the relevance of authors' theoretical analysis.
> >
> > Regarding related work, I do think this paper is fairly comprehensive in discussing previous papers and is already above the bar for publication. That said, no paper is perfect and I believe this paper can benefit from relevant observations given by Meng et al. Still, this point is relatively minor so I didn't include this one in the required changes.

---

> > > ### Author Response · Authors · 2022-06-13
> > > **Reply to reviewer UUHX**
> > >
> > > We thank the reviewer for their quick reply and added clarification on the requested experiment. We point out that it is mathematically impossible to achieve perfect reconstructions if the intrinsic dimension is set too small and the encoder/decoder functions are continuous. For example, if $D=2$, $\mathbb{P}^\*$ has full support (e.g. $2$-dimensional Gaussian, which implies the true intrinsic dimension in this example is also $2$), and $g:\mathbb{R}^2 \rightarrow \mathbb{R}$ and $G:\mathbb{R} \rightarrow \mathbb{R}^2$ are continuous, then it is impossible to achieve $\mathbb{E}_{X \sim \mathbb{P}^\*}[||G(g(X)) - X||_2^2]=0$, since the image of $G$, $G(\mathbb{R})$, will by construction be $1$-dimensional. In other words, it would only be possible to learn a "$1$-dimensional slice" of the manifold. Please let us know if such a visualization would not be found satisfactory, and we will include it in the appendix otherwise.

---

### Review · Reviewer_Cev4 · 2022-06-10

**Summary Of Contributions:**

- The paper identifies a failure mode of maximum likelihood training, where the data distribution is concentrated on a low-dimensional manifold, but a density model is defined over the full input space. In this case, the authors argue that the density model can learn an arbitrary distribution over the data manifold.
- The authors prove the possibility of manifold overfitting theoretically.
- The authors propose a two-step procedure, where first the data is projected in a low-dimensional embedding space, and then a density model is learned in the embedding space. The embedding is done with a generalized auto-encoder model.
- The authors show that the two step procedure allows for improvements in FID scores and OOD detection.

**Broader Impact Concerns:**

No ethical concerns.

**Requested Changes:**

I think this is a strong paper and I recommend acceptance. So, I don't request any changes that are _required_ for acceptance, just some suggestions for what can strengthen the paper in my opinion.

- I would be excited to see an experiment showing that manifold overfitting is an issue in practice. [**not required**]
- It would be interesting to see more ablations in the experiments. In particular, what is the effect of the parameter $d$, dimension of the learned manifold? What if you under or over-estimate it. [**not required**]
- More controlled experiments would be interesting. In particular, what if the learned manifold in Fig 4 was different from the ground truth (shifted circle or a square)? [**not required**]
- I would love to see experiments showing that learning a distribution in the low-dimensionality space in the two-step procedure helps with FID scores for state-of-the-art models. As far as I understand, learning a density model in the embedding space should be cheap, so you may even be able to do it on a dataset like ImageNet, with a state-of-the-art GAN model. There, an improvement in FID scores would be quite exciting. [**not required**]

**Strengths And Weaknesses:**

**Strengths.**

- The paper is very well written. It includes fairly technical mathematical statements, but they are presented in a clear way, with supporting intuition. I enjoyed reading the paper.
- The presented observations are interesting and insightful. While in hindsight the main statement (manifold overfitting) may be fairly clear, I don't think I thought about this particular phenomenon deeply before.
- The proposed method is very simple and practical.
- The empirical results are promising.

**Weaknesses and other thoughts.**
- The theory in the paper focuses on the case when the data actually lies on a low-dimensional manifold. However, it is not obvious how much of a concern this scenario is in practice. In particular, in practice people often use dequantization which leads to a slightly softer distribution, which doesn't strictly lie on a manifold. The authors comment on this point and show that adding noise to the data is not sufficient to achieve the gains in FID scores from the proposed two-stage procedure. However, I don't think this point can be fully dismissed based on the presented results.
-  Generally, it is not obvious from the results in the paper if manifold overfitting happens in generative models in practice. The main evidence is the synthetic experiment in Figure 4, and the experiments showing improved FID scores and OOD detection with the two-step procedure. It would be great to see experiments directly showing that a version of manifold overfitting happens in realistic generative models.
- Methods similar to the proposed two-stage procedure have been considered before, at least in the context of OOD detection (but likely also in generative models more generally). For example [1-3] all consider density modeling in the embedding space, showing strong results.
-

**Classical mixture of Gaussians example**

The authors mention that the manifold overfitting is quite different from regular density model overfitting on page 5, and I generally agree. However, I wanted to mention the classical MoG example mentioned in Bishop [4], section 9.2.1, Fig. 9.7. On a finite dataset, a mixture of Gaussians with two components can already achieve arbitrarily high likelihoods: the first component needs to overfit one datapoint, while the other component should just cover the rest of the data. This way, it is possible to achieve arbitrary likelihoods with a fixed capacity model class, as the data size grows.

This example is, however, different from the manifold overfitting, as the likelihood will only be infinite for one datapoint. But I think it is still worth mentioning.

**References**

[1] _Deep Residual Flow for Out of Distribution Detection_
Ev Zisselman, Aviv Tamar

[2] _Why Normalizing Flows Fail to Detect Out-of-Distribution Data_
Polina Kirichenko, Pavel Izmailov, Andrew Gordon Wilson

[3] _Hybrid Models for Open Set Recognition_
Hongjie Zhang, Ang Li, Jie Guo, Yanwen Guo

[4] _Pattern Recognition and Machine Learning_
Christopher Bishop

---

> ### Author Response · Authors · 2022-06-11
> **Response to reviewer Cev4 (part 1)**
>
> We thank the reviewer for their review, and for finding that our work is interesting, insightful, very well written, and that our method is simple, practical, and has promising empirical results. Please see our points below, which we hope address their concerns:
>
> 1. `The theory in the paper focuses on the case when the data actually lies on a low-dimensional manifold. However, it is not obvious how much of a concern this scenario is in practice. In particular, in practice people often use dequantization which leads to a slightly softer distribution, which doesn't strictly lie on a manifold.`
>
> We agree with the reviewer that this is a relevant point of discussion. On the one hand, as mentioned on the paper, we can interpret Theorem 1 as a justification for dequantization, as an attempt to remove the manifold structure that will result in manifold overfitting. Additionally, whenever small amounts of noise are added to the data, the resulting density will be highly peaked around the manifold. Thus, while approaches based on adding noise do mathematically remove the issue of manifold overfitting, the issue might still happen numerically in practice, at least to a certain degree. We believe our experiments do show, convincingly, that naively adding Gaussian noise to the data (even if tuning the amount of noise), does not outperform our proposed two-step models. On the other hand, we agree with the reviewer that this confirmation does not imply that any approach based on adding noise is bound to fail. We actually hope our work might encourage the community to further think of approaches that better take advantage of dequantization and/or denoising approaches to better prevent manifold overfitting. We will clarify this in the manuscript.
>
> 2. `Generally, it is not obvious from the results in the paper if manifold overfitting happens in generative models in practice. The main evidence is the synthetic experiment in Figure 4, and the experiments showing improved FID scores and OOD detection with the two-step procedure. It would be great to see experiments directly showing that a version of manifold overfitting happens in realistic generative models.`
> `I would be excited to see an experiment showing that manifold overfitting is an issue in practice. [not required]`
>
> Could the reviewer please elaborate on what an experiment that they would find convincing would look like? We believe that the only experiments where manifold overfitting can be confirmed to happen with absolute certainty are simulated examples where the ground truth distribution is known, and the fitted density can be observed to spike incorrectly around the ground truth manifold, e.g. figures 2 and 4. In other, more realistic settings, we believe that the best available confirmation of our theoretical results (or any such results for that matter) are empirical observations which are consistent with the developed theory. In this sense, our theory predicts that two-step models will better match the data-generating distribution than single-step models. Hence, we think of our experiments showing that this prediction indeed holds (using both qualitative comparisons and FID, which albeit as mentioned in the manuscript is imperfect, it is widely used as a performance metric) confirm, as strongly as possible in this realistic setting, that single-step likelihood-based models are indeed experiencing manifold overfitting.
>
> 3. `Methods similar to the proposed two-stage procedure have been considered before, at least in the context of OOD detection (but likely also in generative models more generally). For example [1-3] all consider density modeling in the embedding space, showing strong results.`
>
> We thank the reviewer for pointing out these works, as we agree they are relevant. We will cite all of them in the revised version of our manuscript. We point out some fundamental differences, however: $(i)$ our method is completely unsupervised, whereas Zisselman and Tamar [1] and Zhang et al. [3] require label information, and Kirichenko et al. [2] use a pretrained feature extractor on ImageNet; $(ii)$ these methods are explicitly constructed to improve OOD detection, whereas the focus on our work is to explore the performance on OOD detection of models trained purely to recover the data-generating distribution.
>
> 4. **On the mixture of Gaussians example:**
>
> We thank the reviewer for bringing this to our attention. We agree with the reviewer that, while this phenomenon is not manifold overfitting, it should be mentioned, and we will include this in the revised manuscript.

---

> > ### Author Response · Authors · 2022-06-11
> > **Response to reviewer Cev4 (part 2)**
> >
> > 5. `More controlled experiments would be interesting. In particular, what if the learned manifold in Fig 4 was different from the ground truth (shifted circle or a square)? [not required]`
> >
> > Could the reviewer please elaborate on this? Do they mean forcing $G$ and $g$ to not recover the manifold? As evidenced by the theoretical requirement that $G(g(x))=x$, $\mathbb{P}^*$-almost surely, we should not expect to properly recover the target distribution if the generalized autoencoding step fails drastically.
> >
> > 6. `I would love to see experiments showing that learning a distribution in the low-dimensionality space in the two-step procedure helps with FID scores for state-of-the-art models. As far as I understand, learning a density model in the embedding space should be cheap, so you may even be able to do it on a dataset like ImageNet, with a state-of-the-art GAN model. There, an improvement in FID scores would be quite exciting. [not required]`
> >
> > We actually performed some preliminary experiments using a state-of-the-art GAN, StyleGAN2 [4], trying to invert it, and then learning a second-step model on the learned representations. We attempted to carry out an optimization-based GAN inversion algorithm, but found the reconstruction error to be too large to obtain good performance (for example, the FID obtained from the reconstructions was worse than the FID obtained by samples from the GAN, strongly suggesting that the reconstruction error was too large). While we do not believe these results invalidate the idea, we do believe that achieving state-of-the-art results on ImageNet with a GAN within our framework requires some additional engineering to lower reconstruction error, either through improved or specially-tailored GAN inversion methods, or by learning $G$ and $g$ simultaneously. We thus decided this line of inquiry was better left for future work. Finally, we believe the work of Rombach et al. [5] (which we cite and discuss in our paper) also provides strong empirical evidence that our framework can achieve state-of-the-art empirical performance: in this work the authors use a VAE to obtain low-dimensional representations, and then use a diffusion model on these, and obtain extremely strong empirical performance. We believe that research aiming to further improve empirical performance of these models is highly promising.
> >
> >
> > 7. **Please see our general rebuttal for our answer to comments about setting intrinsic dimension.**
> >
> > Finally, we believe the clarifications above address all the raised concerns, please let us know if this is not the case.
> >
> > [4] Analyzing and Improving the Image Quality of StyleGAN. Karras, Laine, Aittala, Hellsten, Lehtinen and Aila. CVPR 2020.
> >
> > [5] High-Resolution Image Synthesis with Latent Diffusion Models. Rombach, Blattmann, Lorenz, Esser, and Ommer. CVPR 2022 (to appear).

---

> > > ### Comment · Reviewer_Cev4 · 2022-06-28
> > > **Thank you for your response**
> > >
> > > Dear authors, thank you for the detailed response! I agree with your points, and I am happy with the updates you made.

---

### Author Response · Authors · 2022-06-11
**General response to reviews**

We thank all the reviewers for their feedback and the time they spent on our paper. We are glad that the reviewers found our submission well written, interesting, and insightful (Cev4); that the analysis was rigorous, uses minimal assumptions, and explains observed phenomena (UUHX); and that the empirical results were convincing (SQHa, Cev4). We will reply to each reviewer individually, except for shared concerns, which we address below.

However, before addressing these concerns, we would like to point out that none of the reviewers commented nor discussed our theoretical results on density evaluation. Density evaluation for trained two-step models is enabled thanks to Theorem 2, which in turn endows implicit models with density evaluation. To our knowledge, this was previously considered impossible, and the inability to evaluate densities was considered a major drawback of implicit models. We see this as another relevant theoretical contribution of our work and would be thrilled to further discuss this during the review process.

We will update our manuscript towards the end of the discussion phase incorporating all feedback and suggestions brought up in the original reviews and during this phase. We will also further emphasize our contributions regarding density evaluation of implicit models, as we acknowledge that our current manuscript does not give enough weight to this contribution. We now address shared concerns:

1. **On setting the dimension of $\mathbb P\_Z$ (SQHa and CeV4):**

We thank the reviewers for bringing this up, as we believe this should be further emphasized in our manuscript. Indeed, mathematically, if the dimension of $\mathbb{P}_Z$ is chosen larger than $d$, then manifold overfitting could still occur, and if it is set smaller than $d$ then the manifold could not be fully learned (i.e. the assumption that $G(g(x))=x$, $\mathbb{P}^*$-a.s. could not hold). In practice we arbitrarily set this dimension to $20$, both because it is close to the latent dimensions commonly used for the latent spaces of VAEs on such datasets, and because the value is close to those estimated by Pope et al. [1]. Importantly, we did not tune this hyperparameter at all. We discussed in Section 7 that being required to choose $d$ is a limitation of our method that could be resolved by statistically estimating $d$, and that performance can likely be further improved in this way. However, we believe that the strong empirical performance of two-step models, even when not tuning $d$, highlights the relevance of accounting for low-dimensional structure for likelihood-based models.

2. **On broader impact and fairness (SQHa and UUHX):**

We thank the reviewers for their comments, and agree that a more detailed discussion, which we provide below, is warranted. We will include this discussion in the revised manuscript.

Generative modelling has many applications, and while it can have positive impacts on society, it is also possible to apply it inappropriately, or create negative societal impacts through its use [2,3]. When datasets are biased, accurate generative models will inherit those biases [4,5]. Inaccurate generative models may introduce new biases not reflected in the data. Our paper addresses a ubiquitous problem in generative modelling with maximum likelihood estimation -- manifold overfitting -- that causes models to fail to learn the distribution of data correctly. In this sense, correcting manifold overfitting should lead to more accurate generative models, and representations that more closely reflect the data.


[1] The Intrinsic Dimension of Images and Its Impact on Learning. Pope, Zhu, Abdelkader, Goldblum, and Goldstein. ICLR 2021.

[2] The Malicious Use of Artificial Intelligence: Forecasting, Prevention, and Mitigation. Brundage, Avin, Clark, Toner, Eckersley, Garfinkel, Dafoe, Scharre, Zeitzoff, Filar, Anderson, Roff, Allen, Steinhardt, Flynn, O hEigeartaigh, Beard, Belfield, Farquhar, Lyle, Crootof, Lyle, Crootof, Evans, Page, Bryson, Yampolskiy, and Amodei. arXiv preprint 2018.

[3] Dual Use of Artificial-Intelligence-Powered Drug Discovery. Urbina, Lentzos, Invernizzi, and Ekins. Nature Machine Intelligence, 2022.

[4] Image Representations Learned with Unsupervised Pre-Training Contain Human-Like Biases. Steed and Caliskan. Proceedings of the 2021 ACM Conference on Fairness, Accountability, and Transparency.

[5] MaGNET: Uniform Sampling from Deep Generative Network Manifolds Without Retraining. Humayun, Balestriero, and Baraniuk. ICLR 2022.

---

### Author Response · Authors · 2022-06-20
**Manuscript update**

We have updated our manuscript in a way which we believe addresses all the points raised by the reviewers. We kindly remind the reviewers that the discussion period will end later this week, and ask them to please let us know if they still have any lingering concerns in a timely manner so that we may address them before this period ends. For the convenience of the reviewers, we have highlighted in blue all the changes we made to the manuscript, which we will obviously remove for the final version of our paper. Please note that some of the added citations (as well as Table 3 and Figure 6 in the appendix) are not rendered blue, but we have cited all the papers mentioned by the reviewers.

---

### Author Response · Authors · 2022-07-29
**Camera-ready**

We have now uploaded the camera-ready version of our paper, along with a short video explaining our work, and the code to reproduce it. We thank the reviewers and the action editor once again.

---

### Decision · Action_Editors · 2022-07-10

**Recommendation:** Accept as is

**Comment:**

The submission conveys a mathematical insight on manifold overfitting rigorously and provides/explains two-steps solutions to address this issue, backed by convincing experiments. Moreover, the authors have also addressed the reviewers concerns adequately, therefore the reviewers and I have come to to a consensus to accept this submission to TMLR.